# Adaptive Online Mirror Descent for Tchebycheff Scalarization in Multi-Objective Learning

**Meitong Liu**                                                                *meitong4@illinois.edu*
*University of Illinois Urbana-Champaign*
*The University of Hong Kong*

**Xiaoyuan Zhang**                                                          *xzhang2523-c@my.cityu.edu.hk*
*City University of Hong Kong*

**Chulin Xie**                                                                     *chulinx2@illinois.edu*
*University of Illinois Urbana-Champaign*

**Kate Donahue**                                                                *kpd@illinois.edu*
*University of Illinois Urbana-Champaign*

**Han Zhao**                                                                      *hanzhao@illinois.edu*
*University of Illinois Urbana-Champaign*

**Reviewed on OpenReview:** *https://openreview.net/forum?id=MUTRffB3af*

## Abstract

Multi-objective learning (MOL) aims to learn under multiple potentially conflicting objectives and strike a proper balance. While recent preference-guided MOL methods often rely on additional optimization objectives or constraints, we consider the classic Tchebycheff scalarization (TCH) that naturally allows for locating solutions with user-specified trade-offs. Due to its minimax formulation, directly optimizing TCH often leads to training oscillation and stagnation. In light of this limitation, we propose an adaptive online mirror descent algorithm for TCH, called (Ada)OMD-TCH. One of our main ingredients is an adaptive online-to-batch conversion that significantly improves solution optimality over traditional conversion in practice while maintaining the same theoretical convergence guarantees. We show that (Ada)OMD-TCH achieves a convergence rate of $\mathcal{O}(\sqrt{\log m/T})$, where $m$ is the number of objectives and $T$ is the number of rounds, providing a tighter dependency on $m$ in the offline setting compared to existing work. Empirically, we demonstrate on both synthetic problems and federated learning tasks that (Ada)OMD-TCH effectively smooths the training process and yields preference-guided, specific, diverse, and fair solutions.

## 1  Introduction

Multi-objective learning (MOL) is a fundamental problem in various domains, including multi-task learning (MTL) (Caruana, 1997), federated learning (FL) (McMahan et al., 2017b), and drug discovery (Xie et al., 2021). These problems often involve conflicting objectives, such as MTL tasks with different label distributions and FL clients with heterogeneous local data, making it infeasible to find a single solution that optimizes all objectives simultaneously. Consequently, MOL methods identify solutions on the Pareto Front (PF), where any improvement in one objective necessitates a compromise in another (Miettinen, 1998). Moreover, being able to locate specific Pareto optimal (PO) solutions with desired trade-offs is particularly helpful for decision makers with their preferences, such as a uniform preference for fairness or a weighted one to prioritize certain objectives. This motivates the development of *preference-guided* MOL methods (Lin et al., 2019; Mahapatra & Rajan, 2020; Kamani et al., 2021; Momma et al., 2022; Chen et al., 2024b).

Table 1: **Comparison of (Ada)OMD-TCH with previous OMD-based methods.**

| Method | Applied preference | Online-to-batch conversion | OMD instance for $\boldsymbol{\lambda}$ | Convergence rate |
|---|---|---|---|---|
| AFL (Mohri et al., 2019), ALMO (Cortes et al., 2020) | Uniform | Uniform avg. | PGD | $\mathcal{O}(\sqrt{m/T})$ |
| GroupDRO (Sagawa et al., 2020), ExcessMTL (He et al., 2024) | Uniform | Uniform avg. | EG | $\mathcal{O}(m\sqrt{\log m/T})$ |
| OMD-TCH (Ours) AdaOMD-TCH (Ours) | **Diverse** | Uniform avg. **Adaptive avg.** | PGD or EG | PGD: $\mathcal{O}(\sqrt{m/T})$ **EG:** $\mathcal{O}(\sqrt{\log m/T})$ |

MOL methods can be categorized into several lines. One classic line is evolutionary algorithms (Deb et al., 2002; Zhang & Li, 2007; Schaffer, 2014), but they often suffer from slow convergence due to the use of zeroth-order optimization oracles. Recently, gradient-based methods have gained increasing attention for their scalability in deep learning (Chen et al., 2025). *Gradient manipulation methods*, represented by the Multiple Gradient Descent Algorithm (MGDA), aim to find a descent direction by resolving gradient conflicts (Fliege & Svaiter, 2000; Désidéri, 2012). However, they lack the ability to precisely locate a desired PO solution. To overcome this limitation, recent *preference-guided methods* adopt additional optimization objectives or constraints to move iterates closer to the PF and the specified preference simultaneously (Mahapatra & Rajan, 2020; Chen et al., 2024b), which often requires solving subprograms and thus induces more complexity. Another line of methods to tackle MOL is *scalarization techniques*, which reduce multiple objectives to a scalar function (Miettinen, 1998). Linear scalarization has been widely used for its simplicity, yet it is incapable of finding solutions on the non-convex part of the PF (Hu et al., 2024). Moreover, for a given preference, the solution found by LS depends on the shape of the PF (Boyd & Vandenberghe, 2004) and thus does not conform to a strict trade-off. In contrast, *Tchebycheff scalarization* overcomes both drawbacks — it is able to recover the full PF and calibrate the precise quantitative relation among objectives (Ehrgott, 2005), making it a natural choice for preference-guided MOL. Nevertheless, due to its non-smooth nature, direct optimization often suffers from training oscillation and stagnation (Mahapatra & Rajan, 2021).

Given its stated advantages, this paper aims to build upon Tchebycheff scalarization (TCH) to solve preference-guided MOL. To overcome its drawback mentioned above, inspired by the literature on online learning (Nemirovskij & Yudin, 1983; Hazan et al., 2016), we propose an adaptive solver for TCH using online mirror descent (OMD), named (Ada)OMD-TCH, that bypasses the unstable non-smooth formulation. One of our main technical ingredients is a novel adaptive online-to-batch conversion scheme, yielding AdaOMD-TCH, that adaptively discards Pareto-dominated iterates and outputs a reweighted average for offline/batch learning. This scheme is motivated by the poor solution optimality of OMD-TCH using the traditional online-to-batch conversion, which uniformly averages all iterates as the final output.

Theoretically, we prove that (Ada)OMD-TCH converges to the optimal solution of the original TCH problem, i.e., the PO solution with a given trade-off, at a rate of $\mathcal{O}(\sqrt{\log m/T})$ or $\mathcal{O}(\sqrt{m/T})$, depending on the OMD instance, in the stochastic, offline/batch setting, where $m$ is the number of objectives and $T$ is the number of rounds. Our $\mathcal{O}(\sqrt{\log m/T})$ bound reveals a tighter dependency on $m$ for offline learning compared to previous works (Sagawa et al., 2020; He et al., 2024). Empirically, we demonstrate on seven synthetic problems and federated learning tasks that (1) (Ada)OMD-TCH effectively stabilizes the training process as opposed to directly optimizing TCH, (2) AdaOMD-TCH significantly improves solution optimality over OMD-TCH, while retaining the same theoretical convergence properties, and (3) (Ada)OMD-TCH is able to locate the specified and diverse solutions, serving as a competitive preference-guided MOL method.

Beyond the technical contributions, our work contributes to conceptually connecting and unifying different applications in machine learning literature that admit the same problem structure, including agnostic federated learning (AFL) (Mohri et al., 2019), agnostic multi-objective learning (ALMO) (Cortes et al., 2020), group distributionally robust optimization (GroupDRO) (Sagawa et al., 2020), and robust multi-task learning (ExcessMTL) (He et al., 2024), among many others. Despite their distinct application scenarios, the underlying minimax formulation adopted in these works is essentially a special case of Tchebycheff scalarization under a *uniform* preference. By allowing for general weighted preferences, we unify and extend these methods into a

generalized framework, and contribute (1) a novel adaptive online-to-batch conversion scheme that boosts solution optimality without affecting convergence guarantees, (2) a theoretical iteration complexity with optimal dependence on the number of objectives for the offline setting, and (3) empirical evidence of the framework as a competitive preference-guided MOL method, as summarized in Table 1.

## 2    Preliminaries

**Notation.** Given a positive integer $n$, we use $[n]$ for the set $\{1, 2, \ldots, n\}$ and $\Delta_n$ for the $(n-1)$-dimensional simplex $\{\boldsymbol{\alpha} \in \mathbb{R}^n \mid \sum_{i=1}^n \alpha_i = 1, \alpha_i \geq 0, \forall i\}$.

### 2.1    Multi-objective learning

MOL aims to tackle the vector-optimization problem: $\min_{\boldsymbol{\theta} \in \Theta} \mathbf{f}(\boldsymbol{\theta}) = (f_1(\boldsymbol{\theta}), f_2(\boldsymbol{\theta}), \ldots, f_m(\boldsymbol{\theta}))^\top$, where $\Theta \subset \mathbb{R}^d$ is the feasible region and $f_i : \mathbb{R}^d \mapsto \mathbb{R}$ are $m$ differentiable objective functions. Generally, objectives conflict with each other so that no single solution is optimal for every objective. We are instead interested in Pareto optimal (PO) solutions (Miettinen, 1998):

**Definition 2.1** ((Strict) Pareto dominance). For two solutions $\boldsymbol{\theta}_1, \boldsymbol{\theta}_2 \in \Theta$, $\boldsymbol{\theta}_1$ *dominates* $\boldsymbol{\theta}_2$, denoted as $\boldsymbol{\theta}_1 \preceq \boldsymbol{\theta}_2$ (slightly misusing the notation), if $f_i(\boldsymbol{\theta}_1) \leq f_i(\boldsymbol{\theta}_2)$ for all $i \in [m]$ and $f_j(\boldsymbol{\theta}_1) < f_j(\boldsymbol{\theta}_2)$ for some $j \in [m]$. If $f_i(\boldsymbol{\theta}_1) < f_i(\boldsymbol{\theta}_2)$ for all $i \in [m]$, $\boldsymbol{\theta}_1$ strictly dominates $\boldsymbol{\theta}_2$, denoted as $\boldsymbol{\theta}_1 \prec \boldsymbol{\theta}_2$.

**Definition 2.2** ((Weak) Pareto optimality). A solution $\boldsymbol{\theta}^* \in \Theta$ is Pareto optimal if there is no $\boldsymbol{\theta} \in \Theta$ such that $\boldsymbol{\theta} \preceq \boldsymbol{\theta}^*$. A solution $\boldsymbol{\theta}' \in \Theta$ is weakly Pareto optimal if there is no $\boldsymbol{\theta} \in \Theta$ such that $\boldsymbol{\theta} \prec \boldsymbol{\theta}'$.

All (weakly) Pareto optimal solutions form the (weak) Pareto optimal set, whose image in the objective space forms the (weak) Pareto Front (PF). We also define Pareto stationarity:

**Definition 2.3** (Pareto stationarity). A solution $\boldsymbol{\theta}^* \in \Theta$ is Pareto stationary if there exists $\boldsymbol{\alpha} \in \Delta_m$ such that $\sum_{i=1}^m \alpha_i \nabla f_i(\boldsymbol{\theta}) = 0$.

Pareto stationarity is a necessary condition for optimality, and a sufficient condition if objectives are convex and every entry of $\boldsymbol{\alpha}$ is non-zero (Miettinen, 1998).

**Scalarization** is a classic technique for MOL. It finds a particular PO solution by converting the vector-optimization problem into a scalar one given a preference and a scalarization function. Two popular methods are linear scalarization (LS) (Geoffrion, 1967) and Tchebycheff scalarization (TCH) (Bowman Jr, 1976):

$$\text{LS:} \quad \min_{\boldsymbol{\theta} \in \Theta} \text{LS}(\boldsymbol{\theta}; \mathbf{w}) = \min_{\boldsymbol{\theta} \in \Theta} \sum_{i=1}^m w_i f_i(\boldsymbol{\theta}); \qquad \text{TCH:} \quad \min_{\boldsymbol{\theta} \in \Theta} \text{TCH}(\boldsymbol{\theta}; \mathbf{w}) = \min_{\boldsymbol{\theta} \in \Theta} \max_{i \in [m]} w_i f_i(\boldsymbol{\theta}), \tag{1}$$

where $\mathbf{w} \in \Delta_m$ is a user-defined preference vector[1]. It controls the objective trade-off of the specific PO solution found. In particular, LS locates the PO solution where $\mathbf{w}$ is a normal vector of the PF's supporting hyperplane (Boyd & Vandenberghe, 2004), while TCH enjoys the following property:

**Theorem 2.4** (Informal, Ehrgott (2005)). *Suppose $\boldsymbol{\theta}_\mathbf{w}^* = \arg\min_{\boldsymbol{\theta} \in \Theta} \text{TCH}(\boldsymbol{\theta}; \mathbf{w})$, under mild conditions* [2], $w_i f_i(\boldsymbol{\theta}_\mathbf{w}^*) = w_j f_j(\boldsymbol{\theta}_\mathbf{w}^*), \forall i, j \in [m]$.

This quantitative relationship is particularly useful for preference-guided MOL—while the objective trade-off obtained by LS depends on the problem-specific PF's shape, TCH finds the PO solution at the intersection of the preference ray and the PF, offering more precise control. Moreover, LS fails to reach the non-convex regions of the PF (Das & Dennis, 1997; Hu et al., 2024), while TCH can find all (weakly) PO solutions:

---

[1]The complete version of TCH is $\min_{\boldsymbol{\theta} \in \Theta} \max_{i \in [m]} w_i(f_i(\boldsymbol{\theta}) - z_i)$, where $\mathbf{z} \in \mathbb{R}^m$ is a nadir point (i.e., $\mathbf{z} \preceq \boldsymbol{f}(\boldsymbol{\theta}), \forall \boldsymbol{\theta} \in \Theta$). When $\mathbf{f}(\boldsymbol{\theta}) \succeq \mathbf{0}$ for all $\boldsymbol{\theta} \in \Theta$, $\mathbf{z}$ can be set as $\mathbf{0}$ and thus omitted.

[2]① $\exists \boldsymbol{\theta} \in \Theta$, s.t. $w_i f_i(\boldsymbol{\theta}) = w_j f_j(\boldsymbol{\theta}), \forall i, j \in [m]$; ② $\boldsymbol{\theta}_\mathbf{w}^*$ is the only solution to the TCH problem with preference $\mathbf{w}$. When ① is not met, TCH gives the PO solution with the minimum possible maximum weighted objective. When ② is not met, the solution may be weakly PO and the equality in Theorem 2.4 may not hold.

**Theorem 2.5** (Choo & Atkins (1983)). *A feasible solution $\boldsymbol{\theta} \in \Theta$ is weakly Pareto optimal if and only if it is a solution to a Tchebycheff scalarization problem given some $\mathbf{w}$.*

Being able to precisely control objective trade-offs and recover the complete PF, TCH is a natural choice for preference-guided MOL. However, its practicability is limited by training instability and stagnation due to the non-smooth formulation that optimizes only the worst objective at each step (Mahapatra & Rajan, 2020).

## 2.2 Online learning algorithms

**Online learning** refers to scenarios where data arrives in a stream rather than as a fixed dataset (Zinkevich, 2003). The defining characteristic is that the current optimal solution may no longer be optimal after new data is presented. This process can be modeled as a game between a player and an adversary: in each round $t \in \{1, \ldots, T\}$, the player outputs a solution $\mathbf{x}^{(t)} \in \mathcal{X}$ based on past data and the adversary responds with a function $\ell^{(t)} : \mathcal{X} \mapsto \mathbb{R}$ that incurs a loss $\ell^{(t)}(\mathbf{x}^{(t)})$. The goal of the player is to minimize the regret w.r.t. the best solution in hindsight: $\text{Regret}(T) = \sum_{t=1}^{T} \ell^{(t)}(\mathbf{x}^{(t)}) - \min_{\mathbf{x} \in \mathcal{X}} \sum_{t=1}^{T} \ell^{(t)}(\mathbf{x})$. Online algorithms can also be applied to offline/batch learning scenarios with an *online-to-batch conversion* that calculates a deterministic solution from iterates $\{\mathbf{x}^{(t)}\}_{t=1}^{T}$, which conforms to a convergence guarantee transformed from the regret guarantee. The classic conversion is *uniform averaging*: $\bar{\mathbf{x}} = \frac{1}{T} \sum_{t=1}^{T} \mathbf{x}^{(t)}$, which ensures a diminishing optimization error w.r.t. $T$ given a sublinear regret bound (Cesa-Bianchi et al., 2004).

**Online mirror descent** is a fundamental online learning algorithm, commonly favored for its generalizable regret guarantees on decision sets with constrained geometries, as well as practically stable update trajectories (Nemirovskij & Yudin, 1983; Hazan et al., 2016). Its general update rule is:

$$\mathbf{x}^{(t+1)} = \arg\min_{\mathbf{x} \in \mathcal{X}} \langle \nabla \ell^{(t)}(\mathbf{x}^{(t)}), \mathbf{x} \rangle + \frac{1}{\eta} B_\psi(\mathbf{x}; \mathbf{x}^{(t)}), \tag{2}$$

where $\langle \cdot, \cdot \rangle$ denotes the inner product, $\eta$ is the step size, and $B_\psi$ is the Bregman divergence induced by a convex function $\psi : \mathcal{X} \to \mathbb{R}$. With different choices of $\psi$, Equation (2) can be instantiated into different forms. With $\psi$ being the $l$-2 norm and the negative entropy, we have Projected Gradient Descent (PGD) and Exponentiated Gradient (EG), respectively:

$$\text{PGD: } \mathbf{x}^{(t+1)} = \Pi_{\mathcal{X}} \left( \mathbf{x}^{(t)} - \eta \nabla \ell^{(t)}(\mathbf{x}^{(t)}) \right); \qquad \text{EG: } x_i^{(t+1)} = \frac{x_i^{(t)} \exp\left(-\eta \nabla_i \ell^{(t)}(\mathbf{x}^{(t)})\right)}{\sum_{j=1}^{d} x_j^{(t)} \exp\left(-\eta \nabla_j \ell^{(t)}(\mathbf{x}^{(t)})\right)}, \tag{3}$$

where $\Pi_{\mathcal{X}} : \mathbb{R}^d \to \mathcal{X}$ is a projection function to the feasible domain of $\mathbf{x}$.

# 3 (Ada)OMD-TCH

We now present our main methods. Section 3.1 formulates the unified framework for solving TCH via OMD. Section 3.2 introduces the adaptive online-to-batch conversion and its algorithm implementation, AdaOMD-TCH. Section 3.3 provides the convergence analysis and compares it with previous results. Section 3.4 discusses the conceptual differences between (Ada)OMD-TCH and other (preference-guided) MOL methods.

## 3.1 OMD-TCH: a unified framework

**Formulation.** To solve the general TCH problem in Equation (1), following Juditsky et al. (2011), we perform a key equivalent transformation: $\min_{\boldsymbol{\theta} \in \Theta} \max_{i \in [m]} w_i f_i(\boldsymbol{\theta}) = \min_{\boldsymbol{\theta} \in \Theta} \max_{\boldsymbol{\lambda} \in \Delta_m} \sum_{i=1}^{m} \lambda_i \left( w_i f_i(\boldsymbol{\theta}) \right)$. As such, the inner maximization problem is reformulated over a continuous simplex rather than a discrete set, thereby smoothing the one-hot selection. In the following, we use the notation $\mathcal{L}(\boldsymbol{\theta}, \boldsymbol{\lambda}; \mathbf{w}) \coloneqq \sum_{i=1}^{m} \lambda_i \left( w_i f_i(\boldsymbol{\theta}) \right)$.

The transformed problem can be interpreted as a zero-sum game between two players, $\boldsymbol{\theta}$ and $\boldsymbol{\lambda}$, who attempt to minimize and maximize a shared objective $\mathcal{L}(\boldsymbol{\theta}, \boldsymbol{\lambda}; \mathbf{w})$, respectively. This *adversarial* relationship makes optimization for each player an online learning task – in $\boldsymbol{\theta}$'s view, $\boldsymbol{\lambda}$ is the adversary and $\mathcal{L}(\cdot, \boldsymbol{\lambda}^{(t)}; \mathbf{w})$ is the time-varying loss function, and vice versa for player $\boldsymbol{\lambda}$. This dual problem can be solved by applying OMD to $\boldsymbol{\theta}$ and $\boldsymbol{\lambda}$ separately. For $\boldsymbol{\theta}$, setting $\ell_{\boldsymbol{\theta}}^{(t)} = \mathcal{L}(\,\cdot\,, \boldsymbol{\lambda}^{(t)}; \mathbf{w})$, we derive the PGD update from Equation (3):

$$\boldsymbol{\theta}^{(t+1)} = \Pi_\Theta \left( \boldsymbol{\theta}^{(t)} - \eta_{\boldsymbol{\theta}} \sum_{i=1}^{m} \lambda_i^{(t)} w_i \nabla f_i(\boldsymbol{\theta}^{(t)}) \right). \qquad (4)$$

---

**Algorithm 1** OMD-TCH

1: **Input**: Number of rounds $T$, step sizes $\eta_{\boldsymbol{\theta}}$ and $\eta_{\boldsymbol{\lambda}}$, $\boldsymbol{\theta}^{(1)}$, $\boldsymbol{\lambda}^{(1)} = \left( \frac{1}{m}, \cdots, \frac{1}{m} \right)^\top$.
2: **for** round $t = 1, \cdots, T$ **do**
3:     Evaluate $f_i(\boldsymbol{\theta}^{(t)}), \nabla f_i(\boldsymbol{\theta}^{(t)}), \forall i \in [m]$.
4:     Compute $\boldsymbol{\lambda}^{(t+1)}$ by PGD (5) or EG (6).
5:     Compute $\boldsymbol{\theta}^{(t+1)}$ by PGD (4).
6: **end for**
7: **Output**: $\bar{\boldsymbol{\theta}} = \frac{1}{T} \sum_{t=1}^{T} \boldsymbol{\theta}^{(t)}$.

---

When $\Theta$ is unconstrained, PGD reduces to gradient descent. For $\boldsymbol{\lambda} \in \Delta_m$, both PGD and EG are applicable. Setting $\ell_{\boldsymbol{\lambda}}^{(t)} = \mathcal{L}(\boldsymbol{\theta}^{(t)}, \cdot ; \mathbf{w})$, we have from Equation (3):

$$\text{PGD: } \boldsymbol{\lambda}^{(t+1)} = \Pi_{\Delta_m} \left( \boldsymbol{\lambda}^{(t)} + \eta_{\boldsymbol{\lambda}} \mathbf{w} \odot \mathbf{f}(\boldsymbol{\theta}^{(t)}) \right), \qquad (5)$$

$$\text{EG: } \lambda_i^{(t+1)} = \frac{\lambda_i^{(t)} \exp \left( \eta_{\boldsymbol{\lambda}} w_i f_i(\boldsymbol{\theta}^{(t)}) \right)}{\sum_{j=1}^{m} \lambda_j^{(t)} \exp \left( \eta_{\boldsymbol{\lambda}} w_j f_j(\boldsymbol{\theta}^{(t)}) \right)}. \qquad (6)$$

Note that the signs of the gradient terms are flipped for $\boldsymbol{\lambda}$'s maximization goal. See full steps in Algorithm 1.

**Analysis.** The update rule (4) for $\boldsymbol{\theta}$ involves gradients of all objectives, weighted by the a priori preference vector $\mathbf{w}$ and the dynamic $\boldsymbol{\lambda}$. By (5) or (6), $\boldsymbol{\lambda}$ is iteratively updated such that larger weights are assigned to objectives with larger accumulated losses. As such, OMD-TCH smooths out the one-hot update by incorporating all objectives while prioritizing the undertrained ones. As shown later, it converges to the solution of the original TCH problem, i.e., the PO point with the specified trade-off $\mathbf{w}$ as in Theorem 2.4.

**Discussion.** In various scenarios requiring fairness or robustness, many methods adopt a minimax objective and apply different instances of OMD solvers. For example, Mohri et al. (2019) used the PGD instance for $\boldsymbol{\lambda}$ to encourage equal performance across clients in federated learning. Cortes et al. (2020) extended this to multi-objective learning. Sagawa et al. (2020) focused on optimizing the worst group performance for better distributional robustness, and He et al. (2024) tackled multi-task learning, using excess risks as objectives to penalize noise-contaminated tasks, both using EG for $\boldsymbol{\lambda}$. Essentially, these methods solve the special case of TCH with a uniform $\mathbf{w}$, with its validity grounded by Theorem 2.4. However, whether OMD solvers remain effective for diverse trade-offs is left unexplored, which is critical for complete PF traversal or preference-guided MOL. We unify this line by OMD-TCH and provide theoretical and experimental results that support its application as a competitive preference-guided MOL method.

### 3.2 Adaptive online-to-batch conversion

Like the previous methods discussed above, Algorithm 1 adopts the traditional online-to-batch conversion that outputs the uniformly averaged solution $\bar{\boldsymbol{\theta}}$. Although it enjoys a convergence guarantee, we challenge whether the final output should involve every iterate, especially those dominated by new ones as training proceeds. Indeed, OMD-TCH suffers from poor solution optimality, i.e., low overall performance, as observed in our experiments. Motivated by this concern, we propose an adaptive conversion that selectively excludes suboptimal iterates and carefully reweighs the others, retaining the same convergence guarantees and significantly improving solution optimality over the traditional conversion in practice.

**Key idea.** The final output of the adaptive conversion can be expressed as:

$$\tilde{\boldsymbol{\theta}} = \frac{1}{T} \sum_{\boldsymbol{\theta}^{(\tau)} \in \mathcal{P}} \gamma_\tau \boldsymbol{\theta}^{(\tau)},$$

with the following properties:
- $\mathcal{P} = \{\boldsymbol{\theta}^{(\tau)}, \tau \in [T] \mid \nexists t \in [T] \ s.t. \ \boldsymbol{\theta}^{(t)} \preceq \boldsymbol{\theta}^{(\tau)}\}$ is the set of iterates not dominated by any other, which we call trajectory PO iterates.

- $\{\gamma_\tau\}$ is constructed as follows:
    1. Assign each iterate $\boldsymbol{\theta}^{(t)}$ a unit weight of 1.
    2. For each $\boldsymbol{\theta}^{(t)} \notin \mathcal{P}$, distribute its unit weight among $\{\boldsymbol{\theta}^{(\tau)} \in \mathcal{P} \mid \boldsymbol{\theta}^{(\tau)} \preceq \boldsymbol{\theta}^{(t)}\}$, i.e., the trajectory PO iterates that dominate $\boldsymbol{\theta}^{(t)}$.

By this redistribution, we have $\sum_{\boldsymbol{\theta}^{(\tau)} \in \mathcal{P}} \gamma_\tau = T$, which ensures the correct scale of $\tilde{\boldsymbol{\theta}}$.

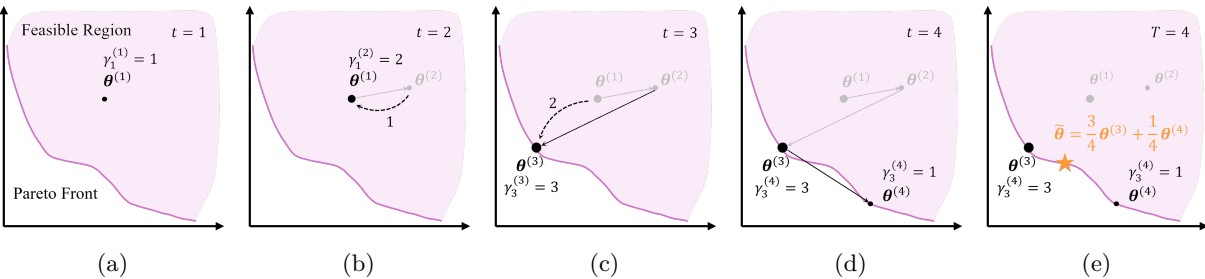

Figure 1: **A sample run of AdaOMD-TCH.** (a) $\boldsymbol{\theta}^{(1)}$ is added to $\mathcal{P}^{(1)}$ with a unit weight of $\gamma_1^{(1)} = 1$. (b) $\boldsymbol{\theta}^{(2)} \succeq \boldsymbol{\theta}^{(1)}$ is discarded with its weight transferred to $\boldsymbol{\theta}^{(1)}$. (c) $\boldsymbol{\theta}^{(3)} \preceq \boldsymbol{\theta}^{(1)}$ is added to $\mathcal{P}^{(3)}$ with its unit weight plus those of $\boldsymbol{\theta}^{(1)}$. $\boldsymbol{\theta}^{(1)}$ is discarded. (d) $\boldsymbol{\theta}^{(4)}$ is a new trajectory optimal iterate and added to $\mathcal{P}^{(4)}$ with $\gamma_4^{(4)} = 1$. (e) The final output $\tilde{\boldsymbol{\theta}}$ is a weighted average of iterates in $\mathcal{P}^{(4)}$.

In short, $\tilde{\boldsymbol{\theta}}$ excludes a large number of insufficiently trained iterates by being *a weighted average of trajectory PO iterates only*, with each one's weight inherited from suboptimal iterates it dominates. The conversion is "adaptive" in the sense that $\mathcal{P}$ and $\{\gamma_\tau\}$ adapt to the actual searching trajectory, rather than being fixed in advance. We show that $\tilde{\boldsymbol{\theta}}$ achieves the same iteration complexity as $\bar{\boldsymbol{\theta}}$ in Section 3.3, and demonstrate its empirical superiority in Section 4.

**Algorithm implementation.** We now introduce AdaOMD-TCH that outputs $\tilde{\boldsymbol{\theta}}$, as summarized in Algorithm 2. The key steps are from lines 4 to 20, where we dynamically maintain a set $\mathcal{P}^{(t)}$, the trajectory PO iterates up to round $t$, and their weights $\boldsymbol{\gamma}^{(t)}$. Specifically, each new candidate $\boldsymbol{\theta}^{(t)}$ is handled by three cases:

- If $\boldsymbol{\theta}^{(t)}$ is dominated by some elements in $\mathcal{P}^{(t-1)}$ ($A \neq \emptyset$, line 6), directly discard it (line 7) and equally split its unit weight to the dominating elements (lines 8 to 10).

- If $\boldsymbol{\theta}^{(t)}$ dominates some elements in $\mathcal{P}^{(t-1)}$ ($B \neq \emptyset$, line 15), discard the suboptimal elements (lines 17, 18) and add $\boldsymbol{\theta}^{(t)}$ to $\mathcal{P}^{(t)}$ (line 12) with its unit weight plus weights inherited from the discarded (lines 13, 16).

- If $\boldsymbol{\theta}^{(t)}$ is not dominated by nor dominating any element in $\mathcal{P}^{(t-1)}$, it is a new temporary trajectory PO iterate; add it to the set with a unit weight of 1 (lines 12, 13).

Figure 1 provides an example illustration. The aforementioned properties can be easily checked: when the algorithm finishes, $\mathcal{P}^{(T)} = \mathcal{P}$; the construction of $\{\gamma_\tau\}$ happens at each step, where we assign a unit weight to the new iterate and reallocate the weights of iterates that become suboptimal.

---

**Algorithm 2** AdaOMD-TCH

1: **Input**: Number of rounds $T$, step sizes $\eta_{\boldsymbol{\theta}}$ and $\eta_{\boldsymbol{\lambda}}$, $\boldsymbol{\theta}^{(1)}$, $\boldsymbol{\lambda}^{(1)} = (\frac{1}{m}, \cdots, \frac{1}{m})^\top$, $\mathcal{P}^{(0)} = \boldsymbol{\gamma}^{(0)} = \emptyset$.
2: **for** round $t = 1, \cdots, T$ **do**
3:      Evaluate $f_i(\boldsymbol{\theta}^{(t)}), \nabla f_i(\boldsymbol{\theta}^{(t)})$, for all $i \in [m]$.
4:      Update $\mathcal{P}^{(t)}$ and $\boldsymbol{\gamma}^{(t)}$: // adaptive conversion
5:      $A \leftarrow \{\boldsymbol{\theta}^{(\tau)} \in \mathcal{P}^{(t-1)} \mid \boldsymbol{\theta}^{(\tau)} \preceq \boldsymbol{\theta}^{(t)}\}$.
6:      **if** $A \neq \emptyset$ **then**
7:          $\mathcal{P}^{(t)} \leftarrow \mathcal{P}^{(t-1)}, \boldsymbol{\gamma}^{(t)} \leftarrow \boldsymbol{\gamma}^{(t-1)}$.
8:          **for** $\boldsymbol{\theta}^{(\tau)} \in A$ **do**
9:              $\gamma_\tau^{(t)} \leftarrow \gamma_\tau^{(t)} + \frac{1}{|A|}$.
10:          **end for**
11:      **else**
12:          $\mathcal{P}^{(t)} \leftarrow \mathcal{P}^{(t-1)} \cup \{\boldsymbol{\theta}^{(t)}\}$.
13:          $\boldsymbol{\gamma}^{(t)} \leftarrow \boldsymbol{\gamma}^{(t-1)} \cup \{\gamma_t^{(t)} = 1\}$.
14:          $B \leftarrow \{\boldsymbol{\theta}^{(\tau)} \in \mathcal{P}^{(t-1)} \mid \boldsymbol{\theta}^{(t)} \preceq \boldsymbol{\theta}^{(\tau)}\}$.
15:          **if** $B \neq \emptyset$ **then**
16:              $\gamma_t^{(t)} \leftarrow \gamma_t^{(t)} + \sum_{\boldsymbol{\theta}^{(\tau)} \in B} \gamma_\tau^{(t)}$.
17:              $\mathcal{P}^{(t)} \leftarrow \mathcal{P}^{(t)} \setminus B$.
18:              $\boldsymbol{\gamma}^{(t)} \leftarrow \boldsymbol{\gamma}^{(t)} \setminus \{\gamma_\tau^{(t)}\}_{\boldsymbol{\theta}^{(\tau)} \in B}$.
19:          **end if**
20:      **end if**
21:      Calculate $\boldsymbol{\lambda}^{(t+1)}$ and $\boldsymbol{\theta}^{(t+1)}$ as in OMD-TCH.
22: **end for**
23: **Output**: $\tilde{\boldsymbol{\theta}} = \frac{1}{T} \sum_{\boldsymbol{\theta}^{(\tau)} \in \mathcal{P}^{(T)}} \gamma_\tau^{(T)} \boldsymbol{\theta}^{(\tau)}$.

---

**Complexity analysis.** The memory complexity of AdaOMD-TCH is determined by the instantaneous size of $\mathcal{P}^{(t)}$. In practice, most elements in the current optimal set are soon dominated and replaced by new iterates as training proceeds. Hence, $\mathcal{P}^{(t)}$ is often sufficiently small, which is also our motivation to dynamically maintain $\mathcal{P}^{(t)}$ and $\boldsymbol{\gamma}^{(t)}$, instead of deriving them after saving all iterates. The time complexity mainly depends on the efficiency of querying, insertion, and deletion on $\mathcal{P}^{(t)}$. With advanced data structures such as k-d trees, such operations can be optimized to a logarithmic complexity w.r.t. the set size on average. In our experiments, we implement naive traversal and already observe minimal overhead, again, due to the small size of $\mathcal{P}^{(t)}$ in practice.

### 3.3 Theoretical convergence guarantees

This section analyzes the convergence rate of (Ada)OMD-TCH for offline/batch learning, where the training dataset for each objective remains fixed. We adopt assumptions commonly used in the convergence analysis for OMD-based methods (Mohri et al., 2019; Cortes et al., 2020; Sagawa et al., 2020; He et al., 2024):

**Assumption 3.1.** For all $i \in [m]$, $f_i(\boldsymbol{\theta})$ is convex in $\boldsymbol{\theta}$ and bounded by $U$: $f_i(\boldsymbol{\theta}) \leq U$, and gradients are bounded by $L$ in the infinity norm: $\|\nabla f_i(\boldsymbol{\theta})\|_\infty \leq L$. The feasible region is bounded by $R_{\boldsymbol{\theta}}$: $\|\boldsymbol{\theta}\|_\infty \leq R_{\boldsymbol{\theta}}$.

Additionally, we consider stochastic feedback and construct the stochastic gradients as follows: in each round $t \in [T]$, for each objective $i \in [m]$, sample a random batch $B_i$ from its training set and compute the stochastic gradients w.r.t. $\boldsymbol{\theta}$ and $\boldsymbol{\lambda}$: $\delta\ell_{\boldsymbol{\theta}}^{(t)} = \sum_{i=1}^m \lambda_i^{(t)} \delta f_i(\boldsymbol{\theta}^{(t)}) = \sum_{i=1}^m \lambda_i^{(t)} \left( \frac{1}{|B_i|} \sum_{(\mathbf{x},y)\in B_i} \nabla f_i(\boldsymbol{\theta}^{(t)}; (\mathbf{x},y)) \right)$, $\delta\ell_{\boldsymbol{\lambda}}^{(t)} = [\dots, \frac{1}{|B_i|} \sum_{(\mathbf{x},y)\in B_i} f_i(\boldsymbol{\theta}^{(t)}; (\mathbf{x},y)), \dots]^T$. Note that we use the $\delta$'s to update $\boldsymbol{\theta}$ and $\boldsymbol{\lambda}$ and still use $\nabla$ to denote the true gradient. By construction, the stochastic gradients are unbiased, and bounded under Assumption 3.1: $\|\delta\ell_{\boldsymbol{\theta}}^{(t)}\|_\infty \leq L$, $\|\delta\ell_{\boldsymbol{\lambda}}^{(t)}\|_\infty \leq U$, for any $t \in [T]$.

We first present the general convergence rate using arbitrary distance-generating functions $\psi_{\boldsymbol{\theta}}$ and $\psi_{\boldsymbol{\lambda}}$.

**Theorem 3.2.** *Suppose Assumption 3.1 holds. With $\psi_{\boldsymbol{\theta}}$, $\psi_{\boldsymbol{\lambda}}$, $\eta_{\boldsymbol{\theta}}$, and $\eta_{\boldsymbol{\lambda}}$ satisfying (1) $\psi_{\boldsymbol{\theta}}$ is $\mu_{\boldsymbol{\theta}}$-strongly convex, (2) $\psi_{\boldsymbol{\lambda}}$ is $\mu_{\boldsymbol{\lambda}}$-strongly convex, (3) $\eta_{\boldsymbol{\theta}} = \sqrt{\frac{4\mu_{\boldsymbol{\theta}} D_{\boldsymbol{\theta}}}{5TC_{\boldsymbol{\theta}} L^2}}$, and (4) $\eta_{\boldsymbol{\lambda}} = \sqrt{\frac{4\mu_{\boldsymbol{\lambda}} D_{\boldsymbol{\lambda}}}{5TC_{\boldsymbol{\lambda}} U^2}}$, both Algorithms 1 and 2 converge as: with probability at least $1 - \gamma$, $0 < \gamma < 1$,*

$$\text{TCH}(\hat{\boldsymbol{\theta}}; \mathbf{w}) - \min_{\boldsymbol{\theta}\in\Theta} \text{TCH}(\boldsymbol{\theta}; \mathbf{w}) \leq \sqrt{\frac{20 D_{\boldsymbol{\theta}} C_{\boldsymbol{\theta}} L^2}{\mu_{\boldsymbol{\theta}} T}} + \sqrt{\frac{20 D_{\boldsymbol{\lambda}} C_{\boldsymbol{\lambda}} U^2}{\mu_{\boldsymbol{\lambda}} T}} + 4(dR_{\boldsymbol{\theta}} L + U)\sqrt{\frac{2}{T}\log\frac{1}{\gamma}}. \tag{7}$$

*On the LHS, $\hat{\boldsymbol{\theta}}$ is $\bar{\boldsymbol{\theta}}$ or $\tilde{\boldsymbol{\theta}}$ output by Algorithm 1 or 2, respectively. On the RHS, $D_{\boldsymbol{\theta}}$, $D_{\boldsymbol{\lambda}}$, $C_{\boldsymbol{\theta}}$, and $C_{\boldsymbol{\lambda}}$ are constants depending on the choices of $\psi_{\boldsymbol{\theta}}$ and $\psi_{\boldsymbol{\lambda}}$ and potentially contain factors of the dimensions of $\boldsymbol{\theta}$ (i.e., $d$) and $\boldsymbol{\lambda}$ (i.e., $m$), respectively. Uncertainty comes from the stochastic gradients.*

**Remark 3.3.** Theorem 3.2 indicates that the outputs of both OMD-TCH and AdaOMD-TCH converge to the solution of the original TCH problem, i.e., a specific PO point with trade-offs controlled by $\mathbf{w}$ following Theorem 2.4. Note that the convergence rate in T, $\mathcal{O}(1/\sqrt{T})$, matches single-objective stochastic gradient descent. The less-informative expectation bound and proof are deferred to Appendix A.1.

To further reveal the dependency on $d$ and $m$, one must unwrap the factors hidden in the constants $D$ and $C$, which depend on the specific choice of OMD instances, i.e., the choice of $\psi$'s. We present two instantiated bounds where we use PGD and EG for $\boldsymbol{\lambda}$, respectively, and PGD for $\boldsymbol{\theta}$ in both cases. We also discuss the $p$-norm instance in Appendix A.2, whose bound is found to be no better than PGD or EG.

**Corollary 3.4.** *Suppose Assumption 3.1 holds. Using PGD for both $\boldsymbol{\theta}$ and $\boldsymbol{\lambda}$, the constants are: $\mu_{\boldsymbol{\theta}} = \mu_{\boldsymbol{\lambda}} = 1$, $D_{\boldsymbol{\theta}} = 2dR_{\boldsymbol{\theta}}^2$, $D_{\boldsymbol{\lambda}} = 2$, $C_{\boldsymbol{\theta}} = d$, $C_{\boldsymbol{\lambda}} = m$. Therefore, under optimal step sizes $\eta_{\boldsymbol{\theta}} = \sqrt{\frac{8R_{\boldsymbol{\theta}}^2}{5TL^2}}$ and $\eta_{\boldsymbol{\lambda}} = \sqrt{\frac{8}{5TmU^2}}$, with probability at least $1 - \gamma$, $0 < \gamma < 1$, both Algorithms 1 and 2 converge as*

$$\text{TCH}(\hat{\boldsymbol{\theta}}; \mathbf{w}) - \min_{\boldsymbol{\theta}\in\Theta} \text{TCH}(\boldsymbol{\theta}; \mathbf{w}) \leq \frac{2\sqrt{10}dR_{\boldsymbol{\theta}} L}{\sqrt{T}} + \frac{2\sqrt{10}\sqrt{m}U}{\sqrt{T}} + 4(dR_{\boldsymbol{\theta}} L + U)\sqrt{\frac{2}{T}\log\frac{1}{\gamma}}. \tag{8}$$

**Corollary 3.5.** *Suppose Assumption 3.1 holds. Using PGD for $\boldsymbol{\theta}$ and EG for $\boldsymbol{\lambda}$, the constants are: $\mu_{\boldsymbol{\theta}} = \mu_{\boldsymbol{\lambda}} = 1$, $D_{\boldsymbol{\theta}} = 2dR_{\boldsymbol{\theta}}^2$, $D_{\boldsymbol{\lambda}} = \log m$, $C_{\boldsymbol{\theta}} = d$, $C_{\boldsymbol{\lambda}} = 1$. Therefore, under optimal step sizes $\eta_{\boldsymbol{\theta}} = \sqrt{\frac{8R_{\boldsymbol{\theta}}^2}{5TL^2}}$ and $\eta_{\boldsymbol{\lambda}} = \sqrt{\frac{4\log m}{5TU^2}}$, with probability at least $1 - \gamma$, $0 < \gamma < 1$, both Algorithms 1 and 2 converge as*

$$\text{TCH}(\hat{\boldsymbol{\theta}}; \mathbf{w}) - \min_{\boldsymbol{\theta}\in\Theta} \text{TCH}(\boldsymbol{\theta}; \mathbf{w}) \leq \frac{2\sqrt{10}dR_{\boldsymbol{\theta}} L}{\sqrt{T}} + \frac{2\sqrt{5}\sqrt{\log m}U}{\sqrt{T}} + 4(dR_{\boldsymbol{\theta}} L + U)\sqrt{\frac{2}{T}\log\frac{1}{\gamma}}. \tag{9}$$

**Remark 3.6.** The factor $d$ in both corollaries can be lifted if we bound the $l$-2 norm of $\boldsymbol{\theta}$ and the gradients, as adopted by most previous works. Since the $l$-2 norm usually scales with dimension, we bound the infinity norm to offer a clearer dependency on $d$. Expectation bounds and proofs are provided in Appendix A.2.

Compared to established bounds with EG for $\boldsymbol{\lambda}$ (Sagawa et al., 2020; He et al., 2024), Corollary 3.5 provides a tighter dependency on the number of objectives $m$, improving from $\mathcal{O}(m\sqrt{\log m})$ to $\mathcal{O}(\sqrt{\log m})$ with no stricter assumptions on objective properties. The improvement comes from a different construction of unbiased stochastic gradients based on a more common and natural sampling strategy for offline MOL problems, i.e., to sample from all, instead of one, objectives in each round. Such full sampling is adopted by most gradient-based MOL methods (Mahapatra & Rajan, 2020; Chen et al., 2024a;b), and is a must for those relying on full gradient information to compute the update direction. As such, our bound reveals optimal iteration complexity better aligned with the MOL setting. We include a detailed discussion in Appendix A.3.

### 3.4 Conceptual comparison

In this section, we compare (Ada)OMD-TCH with (1) other scalarization-based methods, (2) gradient manipulation methods, and (3) other preference-based MOL methods.

**Scalarization-based methods.** We present a connection between (Ada)OMD-TCH with EG for $\boldsymbol{\lambda}$ and smooth Tchebycheff scalarization (STCH) (Lin et al., 2024), which also bypasses the one-hot update in vanilla TCH by leveraging the log-sum-exp technique: $\min_{\boldsymbol{\theta}\in\Theta} \text{STCH}(\boldsymbol{\theta}; \mathbf{w}, \mu) = \min_{\boldsymbol{\theta}\in\Theta} \mu \log\left(\sum_{i=1}^{m} e^{\frac{w_i(f_i(\boldsymbol{\theta})-z_i)}{\mu}}\right)$, where $\mu$ is a scaling constant and $\mathbf{z}$ is a nadir point which we set to 0 for simplicity. Optimizing the STCH objective with gradient descent, the update rule is: $\boldsymbol{\theta}^{(t+1)} = \boldsymbol{\theta}^{(t)} - \eta_{\boldsymbol{\theta}} \sum_{i=1}^{m} \alpha_i^{(t)} w_i \nabla f_i(\boldsymbol{\theta}^{(t)})$, where $\alpha_i^{(t)} = \frac{\exp\left(w_i f_i(\boldsymbol{\theta}^{(t)})/\mu\right)}{\sum_{j=1}^{m} \exp\left(w_j f_j(\boldsymbol{\theta}^{(t)})/\mu\right)}$. Compared to Equations (4) and (6), the only difference lies in $\boldsymbol{\alpha}^{(t)}$ and $\boldsymbol{\lambda}^{(t)}$. While $\boldsymbol{\alpha}^{(t)}$ is solely determined by current losses, $\boldsymbol{\lambda}^{(t)}$ incorporates historic information through $\boldsymbol{\lambda}^{(t-1)}$. Such "buffering" can be viewed as a smoothing technique to make $\boldsymbol{\lambda}$ change less drastically, which can be viewed as a result of OMD's inherent property of regularizing distances between consecutive iterates.

From another perspective, in (Ada)OMD-TCH, by setting $\eta_{\boldsymbol{\lambda}}$ to 0, $\boldsymbol{\lambda}$ becomes a static vector initialized uniformly, and the update for $\boldsymbol{\theta}$ (Equation (4)) is equivalent to linear scalarization with preference $\mathbf{w}$. (Ada)OMD-TCH can, therefore, also be interpreted as a middle ground between LS and TCH, for which we provide an intuitive example on our synthetic experiments in Appendix B.1.

**Gradient manipulation (GM) methods.** Although (Ada)OMD-TCH, as well as STCH, uses a dynamic weight to combine objective gradients, appearing similar in form to GM methods, the two streams are fundamentally different. GM methods aim to mitigate gradient conflicts in each iteration by finding a common descent direction. The Multiple Gradient Descent Algorithm (MGDA) solves for the weights that minimize the norm of the composite gradient (Désidéri, 2012; Sener & Koltun, 2018). Some of its variants regularize the change of weights with momentum updates (Zhou et al., 2022) or mirror descent updates (Fernando et al., 2023; Chen et al., 2024a), along with other conflict-resolving strategies (Liu et al., 2021; Yu et al., 2020). Algorithmically, scalarization-based methods compute the weights based on *objective values*, while GM methods rely on *gradient information*. The major drawback of GM methods is that, although shown to converge to a Pareto stationary solution (Fliege et al., 2019), they do not have control over which specific PO solution is found, potentially due to their focus on local gradients, but not the objective value trade-offs. As a consequence, *even if reweighing objectives with preferences* during optimization, they are less effective in recovering specific and diverse solutions, as shown empirically in Section 4.1.

**Preference-guided methods** aim to locate specific PO solutions satisfying certain preferences, enabling decision makers to select those with the most suitable trade-offs. Most of the recent approaches stem from gradient manipulation methods and thus rely on designed mechanisms to steer the trajectory, such as using the discrepancy between the current and desired trade-offs as an additional optimization objective (Mahapatra & Rajan, 2020; Kamani et al., 2021) or casting preferences as optimization constraints (Lin et al., 2019; Chen et al., 2024b). In contrast, (Ada)OMD-TCH builds on the inherent properties of Tchebycheff scalarization (Theorems 2.4 and 2.5), offering a cleaner formulation for preference-guided learning. Momma et al. (2022) also starts from TCH and aims to avoid the one-hot update. Unlike our approach, they achieve this by deriving several levels of dual problems that lead to a strategy optimizing towards Pareto stationarity and the specified preference simultaneously, similar to those adopting an additional optimization objective. However, theoretical analysis on whether and how fast the method converges to the desired PO solution remains missing, while (Ada)OMD-TCH is guaranteed convergence with a good rate.

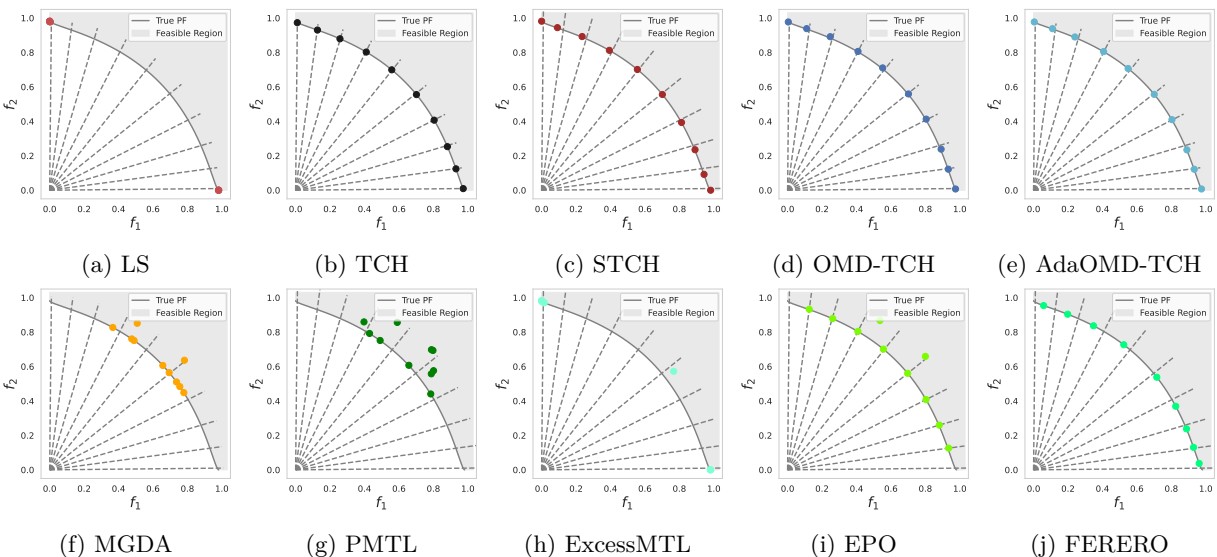

Figure 2: **Solutions found by different methods on VLMOP2.** Each dotted ray denotes the element-wise inverse of a **w**. PGD for **λ** is reported for (Ada)OMD-TCH. Results are averaged over three seeds.

## 4 Experiments

Given the theoretical guarantees for convex objectives in Section 3.3, in this section, we study the performance of (Ada)OMD-TCH in non-convex experiments. We mainly investigate three questions: (1) Can (Ada)OMD-TCH preserve the advantage of TCH in finding specific and diverse PO solutions, serving as a preference-guided MOL method? (2) Can the OMD reformulation mitigate the training oscillation and stagnation problems of vanilla TCH? (3) Is the proposed adaptive online-to-batch conversion effective in improving solution optimality over the traditional conversion? Our experiments provide affirmative evidence for all of them and also demonstrate that AdaOMD-TCH remains highly competitive when compared to other MOL methods.

### 4.1 Synthetic problems

This section focuses on question (1). Synthetic problems with known and visualizable Pareto Fronts provide a suitable environment for testing how well MOL methods locate specific PO solutions under diverse preferences. We verify this ability of (Ada)OMD-TCH on seven non-convex synthetic problems, namely VLMOP2 (Van Veldhuizen & Lamont, 1999) and F1–F6 (Lin et al., 2022). For each problem, we apply ten preference vector **w**'s evenly across the simplex. More detailed setups are deferred to Appendix B.1.

**Methods.** Apart from (Ada)OMD-TCH, we include scalarization methods, namely linear scalarization (LS) and vanilla Tchebycheff scalarization (TCH), as well as the smooth TCH (STCH) (Lin et al., 2024) and ExcessMTL (He et al., 2024), another OMD-based method. We also consider typical gradient manipulation methods, including MGDA, CR-MOGM (Zhou et al., 2022), and Moco (Fernando et al., 2023). Finally, we compare with other preference-guided methods, including PMTL (Lin et al., 2019), EPO (Mahapatra & Rajan, 2020), and FERERO (Chen et al., 2024b). Note that some preference-guided methods discussed earlier (Kamani et al., 2021; Momma et al., 2022) are omitted in the experiments due to a lack of official code for fair comparison. Hyperparameter choices are reported in B.1.

**Results.** Figure 2 illustrates results on VLMOP2. In the figures, each dotted ray represents the element-wise inverse of a preference vector **w**. Ideally, solutions should be scattered across the PF and preferably at the intersections of the rays and the PF, where the objective trade-off strictly follows the corresponding **w**.

As expected, LS only locates the two endpoints on the PF and completely misses the non-convex region. TCH yields diverse solutions on the exact intersections. Note that TCH performs well on these synthetic problems, but degrades drastically when applied to more complex ones, as shown later. Both proposed as

smooth solvers for TCH, solutions of STCH and (Ada)OMD-TCH resemble those found by vanilla TCH, as expected. However, ExcessMTL, although using OMD-based updates as well, does not recover TCH, which we verified is not an artifact of hyperparameters and may be caused by inaccurate estimates of excess risks.

Additionally, MGDA finds arbitrary solutions, even when objectives are reweighed by preferences. Its variants CR-MOGM and Moco show similar results, as in Appendix B.1. This exemplifies that gradient manipulation methods, without additional designs, are insufficient for preference-guided learning. Other preference-guided MOL methods yield less satisfactory solutions compared to STCH and (Ada)OMD-TCH: PMTL does not find the exact PO solutions; EPO performs well for most trade-offs, but misses the two extreme ones; FERERO gives a fairly diverse set, but is less accurate in locating the intersections.

Similar results on the other problems can be found in Appendix B.1. Note that all synthetic problems have non-convex objectives, and VLMOP2 has a non-convex PF. As shown, (Ada)OMD-TCH preserves the property of TCH in finding specific and diverse PO solutions and remains effective in non-convex cases, which complements our theoretical analysis assuming convexity.

## 4.2 Federated learning

To test (Ada)OMD-TCH on real data, we consider federated learning (FL) tasks where clients with heterogeneous local data constitute multiple objectives. First, we address questions (2) and (3) in a 10-client FL setting focused on the fairness-accuracy challenge, where one aims to find the PO solution with a balanced objective trade-off. We demonstrate more stable training using OMD updates and better solution optimality through the adaptive conversion. Next, we employ a 2-client setting to compare (Ada)OMD-TCH against other preference-guided MOL methods under diverse preferences, providing further supporting evidence for question (1). Finally, we benchmark our approach against a wider range of FL methods in the 10-client FL fairness experiments, where AdaOMD-TCH achieves highly competitive results.

### 4.2.1 AdaOMD-TCH gives more stable training and better solution optimality

First, we compare (Ada)OMD-TCH, linear scalarization (LS), and the vanilla Tchebycheff scalarization (TCH), to show that OMD updates effectively stabilize the training process over TCH, and that the adaptive online-to-batch conversion significantly improves solution optimality over the traditional conversion.

**Data, models, and metrics.** Following the settings in Ghosh et al. (2020) and Collins et al. (2021), we simulate three scenarios of data heterogeneity with $m = 10$ clients, namely Rotation and Partial Class with $C = 2$ and 5 classes per client, respectively, using MNIST (Deng, 2012) and CIFAR10 (Krizhevsky & Hinton, 2009). We train a two-layer fully connected neural network with ReLU for MNIST and a ResNet18 (He et al., 2016) model for CIFAR10. The goal is to find the optimal solution with a fair, uniform trade-off. We evaluate the methods by both overall accuracy, for which we use *average test accuracy*, and client-level fairness, for which we use *agnostic loss* (Mohri et al., 2019) and *accuracy parity* (Li et al., 2019). Note that lower fairness metrics are better. Detailed setups, hyperparameters, and random seeds are reported in Appendix B.2.

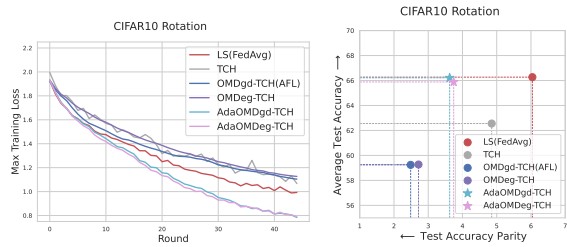

Figure 3: **Examining OMD updates and the adaptive conversion.** "gd" and "eg" denote using PGD and EG, respectively, for $\boldsymbol{\lambda}$. The training curves are plotted on seed=0 to show fluctuations.

**Results.** As shown in Figure 3, TCH suffers from oscillations and slow convergence, with worse results in the Partial Class cases (Appendix B.2). This justifies our motivation for avoiding the one-hot update. OMD-TCH successfully stabilizes the training process and offers a fairer solution than both TCH and LS, but still converges slowly and has an overall accuracy way below LS. Using the adaptive online-to-batch conversion, AdaOMD-TCH preserves only the trajectory PO iterates and thus exhibits significant improvements—faster convergence, higher average accuracy, and minimal impact on fairness. Table 2 presents a more detailed

Table 2: **Comparison between OMD-TCH and AdaOMD-TCH.**

| Method | Rotation | | | Partial Class $C = 2$ | | | Partial Class $C = 5$ | | |
|---|---|---|---|---|---|---|---|---|---|
| | Average Accuracy ↑ | Agnostic Loss ↓ | Accuracy Parity ↓ | Average Accuracy ↑ | Agnostic Loss ↓ | Accuracy Parity ↓ | Average Accuracy ↑ | Agnostic Loss ↓ | Accuracy Parity ↓ |
| OMDgd-TCH | 59.244 | 1.239 | **2.429** | 31.080 | 2.515 | 20.600 | 47.960 | 1.832 | 8.787 |
| AdaOMDgd-TCH | **66.218** | **1.148** | 3.592 | **36.765** | **2.372** | **18.422** | **53.422** | **1.664** | **6.823** |
| OMDeg-TCH | 59.273 | 1.254 | **2.660** | 34.865 | 2.534 | 22.045 | 48.064 | 1.833 | 8.803 |
| AdaOMDeg-TCH | **65.885** | **1.156** | 3.688 | **36.755** | **2.388** | **18.933** | **55.632** | **1.662** | **7.930** |

comparison between OMD-TCH and AdaOMD-TCH on CIFAR, where AdaOMD-TCH outperforms OMD-TCH in almost all cases, especially improving average accuracy to a large extent. The only exception in accuracy parity is possible as the adaptive conversion is mainly designed for better optimality. These results show that (Ada)OMD-TCH stabilizes TCH training, and the adaptive conversion yields significantly better solution optimality over the traditional one.

### 4.2.2 (Ada)OMD-TCH serves as a competitive preference-guided MOL method

Next, we show that (Ada)OMD-TCH remains competitive in finding preference-guided, diverse, and optimal solutions when applied to more complex tasks, complementing results on the synthetic problems.

**Data, metrics, and methods.** To make it easier to quantify the diversity and optimality of the solutions found under multiple preferences, we adopt a 2-client setting where the CIFAR10 dataset is randomly and equally distributed and rotated 0 and 90 degrees, respectively, to create data discrepancy. Following previous work (Chen et al., 2024b), we measure the *hypervolumes* of the solution set over three preference choices of each method, on both objective losses and accuracies. A larger hypervolume indicates a solution set that is both more diverse and more optimal. We compare (Ada)OMD-TCH with methods that are also capable of locating preference-specific PO solutions, including LS, TCH, STCH (Lin et al., 2024), ExcessMTL (He et al., 2024), EPO (Mahapatra & Rajan, 2020), and FERERO (Chen et al., 2024b).

**Results.** As shown in Table 3, OMD-TCH achieves the best results, verifying its application as a general preference-guided MOL method that works beyond the uniform trade-off for fairness and robustness, as used by previous works in Table 1. AdaOMD-TCH achieves slightly lower hypervolumes, which can be explained by its trade-off between optimality and diversity. Specifically, by considering only the non-dominated iterates, AdaOMD-TCH gives better optimality than OMD-TCH in both overall loss ($1.495 < 1.521$) and accuracy ($65.357\% > 62.533\%$), but relatively less diversity, which together leads to lower hypervolumes. Nevertheless, AdaOMD-TCH still yields good results, achieving the second-best hypervolume in accuracy. Moreover, as shown by the 10-client results in Table 2, the improvement in optimality of AdaOMD-TCH is much more significant under a larger client/objective base, making it generally more competitive than OMD-TCH.

### 4.2.3 Comparison with MOL and FL methods on the 10-client fairness benchmark

Finally, we compare (Ada)OMD-TCH with more MOL as well as FL methods under the same 10-client setup as in Section 4.2.1, where one tackles the fairness-accuracy challenge by finding the solution with both higher overall accuracy and better client-level fairness. We show that AdaOMD-TCH, using the adaptive conversion, yields highly competitive performance compared to other methods, especially in fairness.

**Methods.** Apart from (Ada)OMD-TCH, we include all previously mentioned MOL methods: LS, TCH, STCH, ExcessMTL, EPO, and FERERO. Since ExcessMTL adopts a similar OMD framework with EG for

Table 3: **Hypervolumes (↑) of different methods.** PGD for $\boldsymbol{\lambda}$ is reported for (Ada)OMD-TCH.

| Method | LS | TCH | STCH | ExcessMTL | EPO | FERERO | OMD-TCH | AdaOMD-TCH |
|---|---|---|---|---|---|---|---|---|
| Loss | 6.911±0.092 | 6.693±0.863 | 7.583±0.074 | 8.797±0.052 | 7.457±0.208 | 6.041±0.224 | **9.166±0.065** | 7.005±0.106 |
| Accuracy | 0.156±0.001 | 0.125±0.019 | 0.147±0.002 | 0.153±0.001 | 0.141±0.004 | 0.136±0.004 | **0.168±0.001** | 0.159±0.002 |

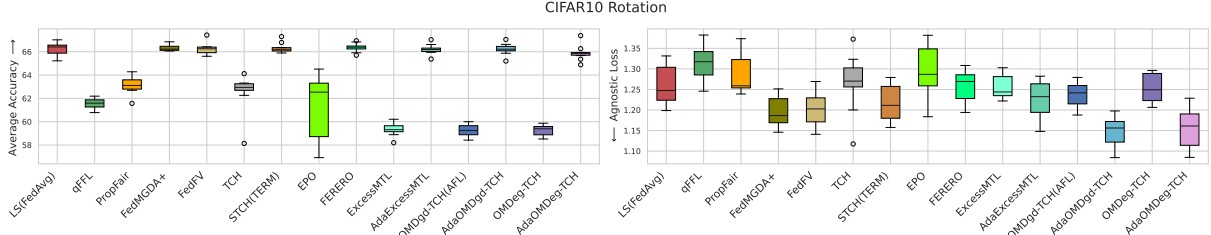

Figure 4: **10-client results for all methods in average accuracy (↑) and agnostic loss (↓).**

**λ**, our adaptive conversion is also applicable, leading to AdaExcessMTL. We also consider fair FL methods: qFFL (Lin et al., 2019), FedFV (Wang et al., 2021), PropFair (Zhang et al., 2022), and FedMGDA+ (Hu et al., 2022). Due to the same nature of learning under multiple objectives, many fair FL methods are equivalent to an MOL method using a uniform preference. In fact, the selected methods also cover FedAvg (McMahan et al., 2017a) (LS), TERM (Li et al., 2020) (STCH), and AFL (Mohri et al., 2019) (OMDgd-TCH).

**Results.** Figure 4 visualizes the average accuracy and agnostic loss of all methods on CIFAR Rotation. AdaOMD-TCH significantly improves solution optimality over OMD-TCH in terms of overall accuracy and achieves competitive results among all methods. It also provides the best fairness in terms of agnostic loss. Similar improvements are observed between ExcessMTL and AdaExcessMTL, which verifies the wide applicability of our adaptive online-to-batch conversion. Compared to other methods, AdaOMD-TCH consistently outperforms qFFL and PropFair across all settings and metrics, as well as EPO and FERERO in most cases. Since OMDgd-TCH encapsulates AFL, the superiority of AdaOMDgd-TCH over OMDgd-TCH also indicates its superiority over AFL. Full results and hyperparameters are in Appendix B.2, where we also report the training time, showing that the adaptive conversion is quite light in computation.

Table 4 offers a finer comparison between AdaOMD-TCH and STCH, both smooth solvers for TCH. As discussed in Section 3.4, STCH and AdaOMDeg-TCH share similar updates, except that the dynamic objective weights in STCH are solely determined by current losses, while those in AdaOMDeg-TCH incorporate historic information. As an attempt to close this gap, we apply the momentum updates proposed in Zhou et al. (2022) to STCH, which also creates a "buffering" effect: $\boldsymbol{\alpha}^{(t)} = \beta_t \boldsymbol{\alpha}^{(t-1)} + (1 - \beta_t)\hat{\boldsymbol{\alpha}}^{(t)}$, where $\hat{\boldsymbol{\alpha}}^{(t)}$ is the original STCH weight and $\beta_t$ is a computable parameter. As shown in Table 4, momentum update can improve STCH, but is still outperformed by AdaOMDeg-TCH in most cases. AdaOMDgd-TCH offers even better results, achieving the best on six out of the nine metrics. Note that STCH w/ momentum, AdaOMDeg-TCH, and AdaOMDgd-TCH all consider past losses for dynamic weights, but in different forms.

### 4.2.4 Ablations and extensions

**Design components.** In Appendix B.2, we restate in detail the roles of the design components in AdaOMD-TCH, namely Tchebycheff scalarization, OMD updates, and the adaptive online-to-batch conversion. We apply the adaptive conversion to LS and TCH, although not rigorously motivated, and examine their performance as a light ablation study. The results further demonstrate the necessity of basing Tchebycheff scalarization and applying OMD updates, as well as the wide applicability of the adaptive conversion.

Table 4: **Comparison between AdaOMD-TCH and STCH.**

| Method | Rotation | | | Partial Class $C = 2$ | | | Partial Class $C = 5$ | | |
|---|---|---|---|---|---|---|---|---|---|
| | Average Accuracy ↑ | Agnostic Loss ↓ | Accuracy Parity ↓ | Average Accuracy ↑ | Agnostic Loss ↓ | Accuracy Parity ↓ | Average Accuracy ↑ | Agnostic Loss ↓ | Accuracy Parity ↓ |
| STCH | 66.320 | 1.219 | 4.997 | 35.380 | 2.386 | 19.139 | **55.328** | **1.622** | 7.311 |
| STCH w/ momentum | **66.422** ↑ | 1.205 ↓ | 4.890 ↓ | 35.105 | 2.380 ↓ | 18.945 ↓ | 54.240 | 1.637 | 7.335 |
| AdaOMDeg-TCH | 65.885 | 1.156 | 3.688 | 36.755 | 2.388 | 18.933 | 55.632 | 1.662 | 7.930 |
| AdaOMDgd-TCH | 66.218 | **1.148** | **3.592** | **36.765** | **2.372** | **18.422** | 53.422 | 1.664 | **6.823** |

**Evaluation data for iterate comparison.** When determining the Pareto dominance among iterates in AdaOMD-TCH, the ideal approach is to evaluate iterates on full training data. Yet in practice, when the data is large and mini-batch updates are adopted, such full evaluations would lead to extra training time. Therefore, in our implementation, we use batch losses as estimates whenever using mini-batch updates. Nevertheless, by comparing results using (1) mini-batch estimates, (2) full training data, and (3) held-out validation sets, respectively, for iterate comparison, we show that the adaptive conversion is robust against different evaluation data. Details are deferred to Appendix B.2.

**Objectives of different scales.** In our FL setting, the client losses are of similar scales, allowing for direct evaluation of preference satisfaction without other artifacts. In broader application scenarios, the objectives may be of different scales, and directly using them for (Ada)OMD-TCH, or any method that involves dynamic weight allocation based on objective values (e.g., TCH and STCH), can cause bias towards objectives of a larger scale and interfere with the original preference. In Appendix B.2, we discuss a practical ratio trick commonly adopted in such cases and show that (Ada)OMD-TCH is compatible with it.

## 5 Conclusion

We propose (Ada)OMD-TCH, an adaptive online mirror descent algorithm for preference-guided multi-objective learning and a unified framework covering multiple previous methods for fairness and robustness. (Ada)OMD-TCH preserves the power of Tchebycheff scalarization in locating specific Pareto optimal solutions and bypasses the one-hot update that leads to training oscillation and stagnation. Our novel adaptive online-to-batch conversion significantly improves solution optimality over traditional conversion while retaining the same convergence guarantees. We prove that (Ada)OMD-TCH converges to the original TCH solution at a rate of $\mathcal{O}(\sqrt{\log m/T})$, revealing an optimal dependency on $m$ for offline learning. We use non-convex synthetic problems and federated learning tasks to show that (1) (Ada)OMD-TCH effectively stabilizes the TCH training process, (2) AdaOMD-TCH significantly improves solution optimality over OMD-TCH, and (3) AdaOMD-TCH is a competitive preference-guided MOL method that finds specific, diverse, and fair solutions.

### Broader Impact Statement

This paper presents work whose goal is to advance the field of multi-objective learning, with potential applications in algorithmic fairness, distributional robustness, and multi-task learning. Given the scope of this research, we do not anticipate immediate ethical concerns or direct societal consequences.

### Acknowledgments

Meitong Liu and Han Zhao are partially supported by an NSF CAREER Award No. 2442290 and a research grant from Meta Platforms, Inc. We thank Dr. Xi Lin and Prof. Qingfu Zhang for their helpful discussion on multi-objective optimization during Han Zhao's visit to the City University of Hong Kong.

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

# A Proofs

## A.1 Theorem 3.2: the corresponding expectation bound and proofs

**Theorem A.1** (Expectation bound for Theorem 3.2). *Suppose Assumption 3.1 holds. With the same choices of $\psi_{\boldsymbol{\theta}}$, $\psi_{\boldsymbol{\lambda}}$, $\eta_{\boldsymbol{\theta}}$, and $\eta_{\boldsymbol{\lambda}}$ as in Theorem 3.2, both Algorithms 1 and 2 converges as:*

$$\mathbb{E}\left[\text{TCH}(\hat{\boldsymbol{\theta}}; \mathbf{w})\right] - \min_{\boldsymbol{\theta} \in \Theta} \text{TCH}(\boldsymbol{\theta}; \mathbf{w}) \leq \sqrt{\frac{20 D_{\boldsymbol{\theta}} C_{\boldsymbol{\theta}} L^2}{\mu_{\boldsymbol{\theta}} T}} + \sqrt{\frac{20 D_{\boldsymbol{\lambda}} C_{\boldsymbol{\lambda}} U^2}{\mu_{\boldsymbol{\lambda}} T}},$$

*where expectation is taken on the stochastic gradients, and all notations are the same as Theorem 3.2.*

We now prove both Theorems 3.2 and A.1, which are based on several lemmas. We first establish the key lemma that fits the proof for $\tilde{\boldsymbol{\theta}}$, output by our adaptive online-to-batch conversion scheme, into the same analysis framework for $\bar{\boldsymbol{\theta}}$, output by the traditional conversion that uses uniform averaging. Recall that $\mathcal{L}(\boldsymbol{\theta}, \boldsymbol{\lambda}; \mathbf{w}) := \sum_{i=1}^{m} \lambda_i (w_i f_i(\boldsymbol{\theta}))$.

**Lemma A.2.** *Suppose Assumption 3.1 holds and $\tilde{\boldsymbol{\theta}} = \frac{1}{T} \sum_{\boldsymbol{\theta}^{(\tau)} \in \mathcal{P}} \gamma_{\tau} \boldsymbol{\theta}^{(\tau)}$ is the output of Algorithm 2 on iterates $\{\boldsymbol{\theta}^{(t)}\}_{t=1}^{T}$ and $\{\boldsymbol{\lambda}^{(t)}\}_{t=1}^{T}$. Then, for any $\boldsymbol{\lambda}, \mathbf{w} \in \Delta_m$,*

$$\mathcal{L}(\tilde{\boldsymbol{\theta}}, \boldsymbol{\lambda}; \mathbf{w}) \leq \frac{1}{T} \sum_{t=1}^{T} \mathcal{L}(\boldsymbol{\theta}^{(t)}, \boldsymbol{\lambda}; \mathbf{w}). \tag{10}$$

*Proof.* First, by Assumption 3.1, $f_i(\boldsymbol{\theta})$ is convex in $\boldsymbol{\theta}$, and hence so is $\mathcal{L}(\boldsymbol{\theta}, \boldsymbol{\lambda}; \mathbf{w})$. By convexity,

$$\mathcal{L}(\tilde{\boldsymbol{\theta}}, \boldsymbol{\lambda}; \mathbf{w}) \leq \frac{1}{T} \sum_{\boldsymbol{\theta}^{(\tau)} \in \mathcal{P}} \gamma_{\tau} \mathcal{L}(\boldsymbol{\theta}^{(\tau)}, \boldsymbol{\lambda}; \mathbf{w}).$$

Now, to prove (10), it suffices to prove:

$$\frac{1}{T} \sum_{\boldsymbol{\theta}^{(\tau)} \in \mathcal{P}} \gamma_{\tau} \mathcal{L}(\boldsymbol{\theta}^{(\tau)}, \boldsymbol{\lambda}; \mathbf{w}) \leq \frac{1}{T} \sum_{t=1}^{T} \mathcal{L}(\boldsymbol{\theta}^{(t)}, \boldsymbol{\lambda}; \mathbf{w}) \tag{11}$$

In fact, this inequality naturally stems from how $\{\gamma_{\tau}\}$ are constructed through weight re-allocation, as stated in the properties of $\tilde{\boldsymbol{\theta}}$. To see this, we first specify some notations that describe the re-allocation:

- Suppose when Algorithm 2 finishes, for each $\boldsymbol{\theta}^{(t)} \notin \mathcal{P}$, its unit weight is re-allocated among elements in $P_t \subset \mathcal{P}$, with each $\boldsymbol{\theta}^{(\tau)} \in P_t$ receiving weight $\beta_{t\tau}$, such that $\sum_{\boldsymbol{\theta}^{(\tau)} \in P_t} \beta_{t\tau} = 1$.
- On the other way round, for each $\boldsymbol{\theta}^{(\tau)} \in \mathcal{P}$, its weight $\gamma_{\tau}$, apart from its own unit weight, is inherited from $S_{\tau} \subset \mathcal{P}^c$, such that $1 + \sum_{\boldsymbol{\theta}^{(t)} \in S_{\tau}} \beta_{t\tau} = \gamma_{\tau}$.

In each step, the weights of some suboptimal iterates are transferred to existing iterates that *dominate* them. By the transitivity of Pareto dominance, when the algorithm finishes, for any $\boldsymbol{\theta}^{(t)} \notin \mathcal{P}$ and $\boldsymbol{\theta}^{(\tau)} \in P_t$, we have $\boldsymbol{\theta}^{(\tau)} \preceq \boldsymbol{\theta}^{(t)}$. Therefore, $f_i(\boldsymbol{\theta}^{(\tau)}) \leq f_i(\boldsymbol{\theta}^{(t)})$ for all $i \in [m]$, and consequently, for any $\boldsymbol{\lambda}, \mathbf{w} \in \Delta_m$, we have:

$$\mathcal{L}(\boldsymbol{\theta}^{(\tau)}, \boldsymbol{\lambda}; \mathbf{w}) \leq \mathcal{L}(\boldsymbol{\theta}^{(t)}, \boldsymbol{\lambda}; \mathbf{w}).$$

With this inequality, we have for each term on the RHS of (11) where $\boldsymbol{\theta}^{(t)} \notin \mathcal{P}$:

$$\begin{aligned}
\mathcal{L}(\boldsymbol{\theta}^{(t)}, \boldsymbol{\lambda}; \mathbf{w}) &= \sum_{\boldsymbol{\theta}^{(\tau)} \in P_t} \beta_{t\tau} \mathcal{L}(\boldsymbol{\theta}^{(t)}, \boldsymbol{\lambda}; \mathbf{w}) \qquad (\because \sum_{\boldsymbol{\theta}^{(\tau)} \in P_t} \beta_{t\tau} = 1) \\
&\geq \sum_{\boldsymbol{\theta}^{(\tau)} \in P_t} \beta_{t\tau} \mathcal{L}(\boldsymbol{\theta}^{(\tau)}, \boldsymbol{\lambda}; \mathbf{w}),
\end{aligned}$$

Finally, we can prove (11) by:

$$
\begin{aligned}
T \times \mathrm{RHS} &= \sum_{\boldsymbol{\theta}^{(\tau)} \in \mathcal{P}} \mathcal{L}(\boldsymbol{\theta}^{(\tau)}, \boldsymbol{\lambda}; \mathbf{w}) + \sum_{\boldsymbol{\theta}^{(t)} \notin \mathcal{P}} \mathcal{L}(\boldsymbol{\theta}^{(t)}, \boldsymbol{\lambda}; \mathbf{w}) \\
&\geq \sum_{\boldsymbol{\theta}^{(\tau)} \in \mathcal{P}} \mathcal{L}(\boldsymbol{\theta}^{(\tau)}, \boldsymbol{\lambda}; \mathbf{w}) + \sum_{\boldsymbol{\theta}^{(t)} \notin \mathcal{P}} \sum_{\boldsymbol{\theta}^{(\tau)} \in P_t} \beta_{t\tau} \mathcal{L}(\boldsymbol{\theta}^{(\tau)}, \boldsymbol{\lambda}; \mathbf{w}) \\
&= \sum_{\boldsymbol{\theta}^{(\tau)} \in \mathcal{P}} \mathcal{L}(\boldsymbol{\theta}^{(\tau)}, \boldsymbol{\lambda}; \mathbf{w}) + \sum_{\boldsymbol{\theta}^{(\tau)} \in \mathcal{P}} \sum_{\boldsymbol{\theta}^{(t)} \in S_\tau} \beta_{t\tau} \mathcal{L}(\boldsymbol{\theta}^{(\tau)}, \boldsymbol{\lambda}; \mathbf{w}) \\
&= \sum_{\boldsymbol{\theta}^{(\tau)} \in \mathcal{P}} \left(1 + \sum_{\boldsymbol{\theta}^{(t)} \in S_\tau} \beta_{t\tau}\right) \mathcal{L}(\boldsymbol{\theta}^{(\tau)}, \boldsymbol{\lambda}; \mathbf{w}) \\
&= \sum_{\boldsymbol{\theta}^{(\tau)} \in \mathcal{P}} \gamma_\tau \mathcal{L}(\boldsymbol{\theta}^{(\tau)}, \boldsymbol{\lambda}; \mathbf{w}) = T \times \mathrm{LHS},
\end{aligned}
$$

which proves (10) and hence the lemma. $\qquad\square$

Lemma A.2 provides an inequality that allows us to continue the proof for $\tilde{\boldsymbol{\theta}}$ the same way as $\bar{\boldsymbol{\theta}}$, which follows the standard analysis of a minimax optimization problem. Before that, we refer to some established results.

**Lemma A.3** (Properties of Bregman divergence, Chen & Teboulle (1993)). *Given a distance generating function $\psi : \mathcal{X} \to \mathbb{R}$ that is $\mu$-strongly convex w.r.t. a norm $\|\cdot\| : \mathcal{X} \to \mathbb{R}$ and continuously differentiable on $\mathrm{int}\,\mathcal{X}$, the Bregman divergence $B_\psi : \mathcal{X} \times \mathrm{int}\,\mathcal{X} \to \mathbb{R}$ induced by $\psi$ is defined as:*

$$
B_\psi(\mathbf{x}; \mathbf{y}) = \psi(\mathbf{x}) - \psi(\mathbf{y}) - \langle \nabla\psi(\mathbf{y}), \mathbf{x} - \mathbf{y} \rangle.
$$

*Moreover, $B_\psi$ has the following properties:*

- *Non-negativity and convexity: $B_\psi \geq 0$ and is convex in the first argument.*
- *Lower bound: For any two points $\mathbf{x} \in \mathcal{X}$ and $\mathbf{y} \in \mathrm{int}\,\mathcal{X}$,*

$$
B_\psi(\mathbf{x}; \mathbf{y}) \geq \frac{\mu}{2} \|\mathbf{x} - \mathbf{y}\|^2.
$$

- *Three-point identity: For any three points $\mathbf{x}, \mathbf{y} \in \mathrm{int}\,\mathcal{X}$ and $\mathbf{z} \in \mathcal{X}$,*

$$
B_\psi(\mathbf{z}; \mathbf{y}) - B_\psi(\mathbf{z}; \mathbf{x}) - B_\psi(\mathbf{x}; \mathbf{y}) = \langle \nabla\psi(\mathbf{y}) - \nabla\psi(\mathbf{x}), \mathbf{x} - \mathbf{z} \rangle.
$$

**Lemma A.4** (Cauchy–Schwarz inequality for dual norms, Boyd & Vandenberghe (2004)). *Suppose $\|\cdot\|_*$ is the dual norm of a norm $\|\cdot\| : \mathcal{X} \to \mathbb{R}$ defined as $\|\mathbf{x}\|_* := \max\limits_{\mathbf{y}, \|\mathbf{y}\| \leq 1} \langle \mathbf{x}, \mathbf{y} \rangle$. Then, for any $\mathbf{x}, \mathbf{y} \in \mathcal{X}$, by definition,*

$$
\langle \mathbf{x}, \mathbf{y} \rangle \leq \|\mathbf{x}\|_* \|\mathbf{y}\|.
$$

**Lemma A.5** (Azuma–Hoeffding inequality on martingales, Azuma (1967)). *A discrete-time stochastic process $\{X_1, X_2, X_3, \ldots\}$ is a martingale w.r.t. a filtration $\{H_1, H_2, H_3, \ldots\}$ if at any time $t$, $\mathbb{E}\left[|X_t|\right] < \infty$ and $\mathbb{E}\left[X_t \mid H_1, \cdots, H_{t-1}\right] = X_{t-1}$. Suppose a martingale satisfies that $|X_t - X_{t-1}| \leq c_t$ almost surely, then, for all positive integers $T$ and all positive reals $\epsilon$,*

$$
P\left(X_T - X_0 \geq \epsilon\right) \leq \exp\left(\frac{-\epsilon^2}{2 \sum_{t=1}^{T} c_t^2}\right).
$$

Based on Lemmas A.3 and A.4, we prove another key lemma:

**Lemma A.6** (Restated, Beck & Teboulle (2003)). *Given arbitrary sequence $\{\boldsymbol{\zeta}^{(1)}, \cdots, \boldsymbol{\zeta}^{(T)}\}$, and sequence $\{\boldsymbol{\nu}^{(1)}, \cdots, \boldsymbol{\nu}^{(T)}\}$ of the same dimension that is defined as:*

$$
\boldsymbol{\nu}^{(t+1)} = \underset{\boldsymbol{\nu} \in V}{\arg\min} \; \langle \boldsymbol{\zeta}^{(t)}, \boldsymbol{\nu} - \boldsymbol{\nu}^{(t)} \rangle + \frac{1}{\eta_{\boldsymbol{\nu}}} B_{\psi_{\boldsymbol{\nu}}}(\boldsymbol{\nu}; \boldsymbol{\nu}^{(t)}), \tag{12}
$$

$$
\text{or } \boldsymbol{\nu}^{(t+1)} = \underset{\boldsymbol{\nu} \in V}{\arg\max} \; \langle \boldsymbol{\zeta}^{(t)}, \boldsymbol{\nu} - \boldsymbol{\nu}^{(t)} \rangle - \frac{1}{\eta_{\boldsymbol{\nu}}} B_{\psi_{\boldsymbol{\nu}}}(\boldsymbol{\nu}; \boldsymbol{\nu}^{(t)}), \tag{13}
$$

where $\eta_{\boldsymbol{\nu}}$ is a constant step size, $B_{\psi_{\boldsymbol{\nu}}}$ is the Bregman divergence induced by $\psi_{\boldsymbol{\nu}} : V \to \mathbb{R}$, and $\psi_{\boldsymbol{\nu}}$ is $\mu_{\boldsymbol{\nu}}$-strongly convex w.r.t. the norm $\|\cdot\|_{\boldsymbol{\nu}}$. Then, for any $\boldsymbol{u} \in V$,

$$\sum_{t=1}^{T} \langle \boldsymbol{\zeta}^{(t)}, \boldsymbol{\nu}^{(t)} - \boldsymbol{u} \rangle \leq \frac{1}{\eta_{\boldsymbol{\nu}}} B_{\psi_{\boldsymbol{\nu}}}(\boldsymbol{u}; \boldsymbol{\nu}^{(1)}) + \frac{\eta_{\boldsymbol{\nu}}}{2\mu_{\boldsymbol{\nu}}} \sum_{t=1}^{T} \|\boldsymbol{\zeta}^{(t)}\|_*^2 , \tag{14}$$

$$\text{or respectively,} \quad \sum_{t=1}^{T} \langle \boldsymbol{\zeta}^{(t)}, \boldsymbol{u} - \boldsymbol{\nu}^{(t)} \rangle \leq \frac{1}{\eta_{\boldsymbol{\nu}}} B_{\psi_{\boldsymbol{\nu}}}(\boldsymbol{u}; \boldsymbol{\nu}^{(1)}) + \frac{\eta_{\boldsymbol{\nu}}}{2\mu_{\boldsymbol{\nu}}} \sum_{t=1}^{T} \|\boldsymbol{\zeta}^{(t)}\|_*^2 , \tag{15}$$

where $\|\cdot\|_*$ is the dual norm of $\|\cdot\|_{\boldsymbol{\nu}}$.

*Proof.* Consider a single term on the LHS of (14),

$$\langle \boldsymbol{\zeta}^{(t)}, \boldsymbol{\nu}^{(t)} - \boldsymbol{u} \rangle = \langle \boldsymbol{\zeta}^{(t)} + \frac{1}{\eta_{\boldsymbol{\nu}}} \nabla \psi_{\boldsymbol{\nu}}(\boldsymbol{\nu}^{(t+1)}) - \frac{1}{\eta_{\boldsymbol{\nu}}} \nabla \psi_{\boldsymbol{\nu}}(\boldsymbol{\nu}^{(t)}), \boldsymbol{\nu}^{(t+1)} - \boldsymbol{u} \rangle \text{ (term A)}$$
$$+ \langle \frac{1}{\eta_{\boldsymbol{\nu}}} \nabla \psi_{\boldsymbol{\nu}}(\boldsymbol{\nu}^{(t)}) - \frac{1}{\eta_{\boldsymbol{\nu}}} \nabla \psi_{\boldsymbol{\nu}}(\boldsymbol{\nu}^{(t+1)}), \boldsymbol{\nu}^{(t+1)} - \boldsymbol{u} \rangle \text{ (term B)}$$
$$+ \langle \boldsymbol{\zeta}^{(t)}, \boldsymbol{\nu}^{(t)} - \boldsymbol{\nu}^{(t+1)} \rangle \text{ (term C).} \tag{16}$$

We bound terms A and B + C, respectively. First, observe rule (12), the entire function to be minimized on the RHS is convex w.r.t. $\boldsymbol{\nu}$. Then, by the optimality condition for $\boldsymbol{\nu}^{(t+1)}$ on this convex function, we have:

$$\langle \nabla_{\boldsymbol{\nu}} \left( \langle \boldsymbol{\zeta}^{(t)}, \boldsymbol{\nu} - \boldsymbol{\nu}^{(t)} \rangle + \frac{1}{\eta_{\boldsymbol{\nu}}} B_{\psi_{\boldsymbol{\nu}}}(\boldsymbol{\nu}; \boldsymbol{\nu}^{(t)}) \right) |_{\boldsymbol{\nu}^{(t+1)}}, \boldsymbol{u} - \boldsymbol{\nu}^{(t+1)} \rangle$$
$$= \langle \boldsymbol{\zeta}^{(t)} + \frac{1}{\eta_{\boldsymbol{\nu}}} \nabla \psi_{\boldsymbol{\nu}}(\boldsymbol{\nu}^{(t+1)}) - \frac{1}{\eta_{\boldsymbol{\nu}}} \nabla \psi_{\boldsymbol{\nu}}(\boldsymbol{\nu}^{(t)}), \boldsymbol{u} - \boldsymbol{\nu}^{(t+1)} \rangle \geq 0, \quad \forall \boldsymbol{u} \in V.$$

Hence, term A $\leq 0$. Next, we apply the three-point identity in Lemma A.3 to term B:

$$\text{B} = \frac{1}{\eta_{\boldsymbol{\nu}}} \langle \nabla \psi_{\boldsymbol{\nu}}(\boldsymbol{\nu}^{(t)}) - \nabla \psi_{\boldsymbol{\nu}}(\boldsymbol{\nu}^{(t+1)}), \boldsymbol{\nu}^{(t+1)} - \boldsymbol{u} \rangle$$
$$= \frac{1}{\eta_{\boldsymbol{\nu}}} \left( B_{\psi_{\boldsymbol{\nu}}}(\boldsymbol{u}; \boldsymbol{\nu}^{(t)}) - B_{\psi_{\boldsymbol{\nu}}}(\boldsymbol{u}; \boldsymbol{\nu}^{(t+1)}) - B_{\psi_{\boldsymbol{\nu}}}(\boldsymbol{\nu}^{(t+1)}; \boldsymbol{\nu}^{(t)}) \right). \tag{17}$$

Then, we have for term C:

$$\text{C} = \frac{1}{\eta_{\boldsymbol{\nu}}} \langle \eta_{\boldsymbol{\nu}} \boldsymbol{\zeta}^{(t)}, \boldsymbol{\nu}^{(t)} - \boldsymbol{\nu}^{(t+1)} \rangle$$
$$\leq \frac{1}{\eta_{\boldsymbol{\nu}}} \|\eta_{\boldsymbol{\nu}} \boldsymbol{\zeta}^{(t)}\|_* \|\boldsymbol{\nu}^{(t)} - \boldsymbol{\nu}^{(t+1)}\| \text{ (by Lemma A.4)}$$
$$\leq \frac{1}{\eta_{\boldsymbol{\nu}}} \left( \frac{1}{2\mu_{\boldsymbol{\nu}}} \|\eta_{\boldsymbol{\nu}} \boldsymbol{\zeta}^{(t)}\|_{\boldsymbol{\nu},*}^2 + \frac{\mu_{\boldsymbol{\nu}}}{2} \|\boldsymbol{\nu}^{(t)} - \boldsymbol{\nu}^{(t+1)}\|_{\boldsymbol{\nu}}^2 \right). \tag{18}$$

Now, adding (17) and (18),

$$\text{B} + \text{C} = \frac{1}{\eta_{\boldsymbol{\nu}}} \left( B_{\psi_{\boldsymbol{\nu}}}(\boldsymbol{u}; \boldsymbol{\nu}^{(t)}) - B_{\psi_{\boldsymbol{\nu}}}(\boldsymbol{u}; \boldsymbol{\nu}^{(t+1)}) \underline{- B_{\psi_{\boldsymbol{\nu}}}(\boldsymbol{\nu}^{(t+1)}, \boldsymbol{\nu}^{(t)})} \right.$$
$$\left. + \frac{1}{2\mu_{\boldsymbol{\nu}}} \|\eta_{\boldsymbol{\nu}} \boldsymbol{\zeta}^{(t)}\|_{\boldsymbol{\nu},*}^2 + \underline{\frac{\mu_{\boldsymbol{\nu}}}{2} \|\boldsymbol{\nu}^{(t)} - \boldsymbol{\nu}^{(t+1)}\|_{\boldsymbol{\nu}}^2} \right)$$
$$\leq \frac{1}{\eta_{\boldsymbol{\nu}}} \left( B_{\psi_{\boldsymbol{\nu}}}(\boldsymbol{u}; \boldsymbol{\nu}^{(t)}) - B_{\psi_{\boldsymbol{\nu}}}(\boldsymbol{u}; \boldsymbol{\nu}^{(t+1)}) + \frac{\eta_{\boldsymbol{\nu}}^2}{2\mu_{\boldsymbol{\nu}}} \|\boldsymbol{\zeta}^{(t)}\|_{\boldsymbol{\nu},*}^2 \right) \text{ (by Lemma A.3)}$$
$$= \frac{1}{\eta_{\boldsymbol{\nu}}} \left( B_{\psi_{\boldsymbol{\nu}}}(\boldsymbol{u}; \boldsymbol{\nu}^{(t)}) - B_{\psi_{\boldsymbol{\nu}}}(\boldsymbol{u}; \boldsymbol{\nu}^{(t+1)}) \right) + \frac{\eta_{\boldsymbol{\nu}}}{2\mu_{\boldsymbol{\nu}}} \|\boldsymbol{\zeta}^{(t)}\|_{\boldsymbol{\nu},*}^2. \tag{19}$$

Therefore, given term A $\leq 0$ and (19), we have from (16):

$$\langle \boldsymbol{\zeta}^{(t)}, \boldsymbol{\nu}^{(t)} - \boldsymbol{u} \rangle = A + (B + C) \leq \frac{1}{\eta_{\boldsymbol{\nu}}} \left( B_{\psi_{\boldsymbol{\nu}}}(\boldsymbol{u}; \boldsymbol{\nu}^{(t)}) - B_{\psi_{\boldsymbol{\nu}}}(\boldsymbol{u}; \boldsymbol{\nu}^{(t+1)}) \right) + \frac{\eta_{\boldsymbol{\nu}}}{2\mu_{\boldsymbol{\nu}}} \|\boldsymbol{\zeta}^{(t)}\|_{\boldsymbol{\nu},*}^2.$$

Finally, by summing up both sides from $t = 1$ to $T$, we have

$$\sum_{t=1}^{T} \langle \boldsymbol{\zeta}^{(t)}, \boldsymbol{\nu}^{(t)} - \boldsymbol{u} \rangle \leq \frac{1}{\eta_{\boldsymbol{\nu}}} \left( B_{\psi_{\boldsymbol{\nu}}}(\boldsymbol{u}; \boldsymbol{\nu}^{(1)}) \underline{- B_{\psi_{\boldsymbol{\nu}}}(\boldsymbol{u}; \boldsymbol{\nu}^{(T+1)})} \right) + \frac{\eta_{\boldsymbol{\nu}}}{2\mu_{\boldsymbol{\nu}}} \sum_{t=1}^{T} \|\boldsymbol{\zeta}^{(t)}\|_{\boldsymbol{\nu},*}^2$$

$$\leq \frac{1}{\eta_{\boldsymbol{\nu}}} B_{\psi_{\boldsymbol{\nu}}}(\boldsymbol{u}; \boldsymbol{\nu}^{(1)}) + \frac{\eta_{\boldsymbol{\nu}}}{2\mu_{\boldsymbol{\nu}}} \sum_{t=1}^{T} \|\boldsymbol{\zeta}^{(t)}\|_{\boldsymbol{\nu},*}^2 \text{ (by non-negativity of } B_{\psi_{\boldsymbol{\nu}}}),$$

which proves (14). The maximization case in Equation (15) can be proved by the same procedure utilizing the concavity of the function to be maximized in rule (13). □

**Now, we are ready to prove Theorems 3.2 and A.1.**

*Proof of Theorems 3.2 and A.1.* Before starting the derivation, we obtain some bounds on the norm of gradients that will be used later. Directly implied by the assumptions, we have:

$$\|\nabla_{\boldsymbol{\theta}} \mathcal{L}(\boldsymbol{\theta}, \boldsymbol{\lambda}; \mathbf{w})\|_{\infty} = \|\sum_{i=1}^{m} \lambda_i w_i \nabla f_i(\boldsymbol{\theta})\|_{\infty} \leq \|\nabla f_i(\boldsymbol{\theta})\|_{\infty} \leq L, \tag{20}$$

$$\|\nabla_{\boldsymbol{\lambda}} \mathcal{L}(\boldsymbol{\theta}, \boldsymbol{\lambda}; \mathbf{w})\|_{\infty} = \|\mathbf{w} \odot \mathbf{f}(\boldsymbol{\theta})\|_{\infty} \leq \|\mathbf{f}(\boldsymbol{\theta})\|_{\infty} \leq U. \tag{21}$$

The same upper bounds also hold for stochastic gradients. Now we begin our derivation. First, we rewrite the LHS of (7) with $\mathcal{L}(\boldsymbol{\theta}, \boldsymbol{\lambda}; \mathbf{w})$:

$$\text{TCH}(\hat{\boldsymbol{\theta}}; \mathbf{w}) - \min_{\boldsymbol{\theta} \in \Theta} \text{TCH}(\boldsymbol{\theta}; \mathbf{w}) = \max_{\boldsymbol{\lambda} \in \Delta_m} \mathcal{L}(\hat{\boldsymbol{\theta}}, \boldsymbol{\lambda}; \mathbf{w}) - \min_{\boldsymbol{\theta}} \max_{\boldsymbol{\lambda} \in \Delta_m} \mathcal{L}(\boldsymbol{\theta}, \boldsymbol{\lambda}; \mathbf{w}).$$

Given that $\mathcal{L}(\boldsymbol{\theta}, \boldsymbol{\lambda}; \mathbf{w})$ is convex in $\boldsymbol{\theta}$ and linear in $\boldsymbol{\lambda}$, we have:

$$\max_{\boldsymbol{\lambda} \in \Delta_m} \mathcal{L}(\hat{\boldsymbol{\theta}}, \boldsymbol{\lambda}; \mathbf{w}) - \min_{\boldsymbol{\theta}} \max_{\boldsymbol{\lambda} \in \Delta_m} \mathcal{L}(\boldsymbol{\theta}, \boldsymbol{\lambda}; \mathbf{w})$$

$$= \max_{\boldsymbol{\lambda} \in \Delta_m} \mathcal{L}(\hat{\boldsymbol{\theta}}, \boldsymbol{\lambda}; \mathbf{w}) - \max_{\boldsymbol{\lambda} \in \Delta_m} \min_{\boldsymbol{\theta}} \mathcal{L}(\boldsymbol{\theta}, \boldsymbol{\lambda}; \mathbf{w}) \text{ (Von Neumann's theorem)}$$

$$\leq \max_{\boldsymbol{\lambda} \in \Delta_m} \mathcal{L}(\hat{\boldsymbol{\theta}}, \boldsymbol{\lambda}; \mathbf{w}) - \min_{\boldsymbol{\theta}} \mathcal{L}(\boldsymbol{\theta}, \bar{\boldsymbol{\lambda}}; \mathbf{w})$$

$$= \mathcal{L}(\hat{\boldsymbol{\theta}}, \boldsymbol{\lambda}^*; \mathbf{w}) - \mathcal{L}(\boldsymbol{\theta}^*, \bar{\boldsymbol{\lambda}}; \mathbf{w}), \tag{22}$$

where $\boldsymbol{\lambda}^* := \arg\max_{\boldsymbol{\lambda} \in \Delta_m} \mathcal{L}(\hat{\boldsymbol{\theta}}, \boldsymbol{\lambda}; \mathbf{w})$, $\boldsymbol{\theta}^* := \arg\min_{\boldsymbol{\theta}} \mathcal{L}(\boldsymbol{\theta}, \bar{\boldsymbol{\lambda}}; \mathbf{w})$, and $\bar{\boldsymbol{\lambda}} = \frac{1}{T} \sum_{t=1}^{T} \boldsymbol{\lambda}^{(t)}$.

Next, by $\mathcal{L}(\boldsymbol{\theta}, \boldsymbol{\lambda}; \mathbf{w})$'s convexity in $\boldsymbol{\theta}$, we have for $\hat{\boldsymbol{\theta}}$ being $\bar{\boldsymbol{\theta}}$ and $\tilde{\boldsymbol{\theta}}$ respectively:

$$\mathcal{L}(\bar{\boldsymbol{\theta}}, \boldsymbol{\lambda}^*; \mathbf{w}) \leq \frac{1}{T} \sum_{t=1}^{T} \mathcal{L}(\boldsymbol{\theta}^{(t)}, \boldsymbol{\lambda}^*; \mathbf{w}), \tag{23}$$

$$\mathcal{L}(\tilde{\boldsymbol{\theta}}, \boldsymbol{\lambda}^*; \mathbf{w}) \leq \frac{1}{T} \sum_{\boldsymbol{\theta}^{(\tau)} \in \mathcal{P}} \gamma_{\tau} \mathcal{L}(\boldsymbol{\theta}^{(t)}, \boldsymbol{\lambda}^*; \mathbf{w}) \leq \frac{1}{T} \sum_{t=1}^{T} \mathcal{L}(\boldsymbol{\theta}^{(t)}, \boldsymbol{\lambda}^*; \mathbf{w}), \tag{24}$$

where the second inequality in (24) is exactly Lemma A.2. With the same RHS in (23) and (24), we are able to continue the proof for both solutions output by the traditional uniform averaging conversion and the new adaptive conversion. Next, by $\mathcal{L}(\boldsymbol{\theta}, \boldsymbol{\lambda}; \mathbf{w})$'s linearity in $\boldsymbol{\lambda}$, we have:

$$\mathcal{L}(\boldsymbol{\theta}^*, \bar{\boldsymbol{\lambda}}; \mathbf{w}) = \frac{1}{T} \sum_{t=1}^{T} \mathcal{L}(\boldsymbol{\theta}^*, \boldsymbol{\lambda}^{(t)}; \mathbf{w}). \tag{25}$$

With (23), (24), and (25), we continue from (22),

$$\mathcal{L}(\hat{\boldsymbol{\theta}}, \boldsymbol{\lambda}^*; \mathbf{w}) - \mathcal{L}(\boldsymbol{\theta}^*, \bar{\boldsymbol{\lambda}}; \mathbf{w}) \le \frac{1}{T} \sum_{t=1}^{T} \mathcal{L}(\boldsymbol{\theta}^{(t)}, \boldsymbol{\lambda}^*; \mathbf{w}) - \frac{1}{T} \sum_{t=1}^{T} \mathcal{L}(\boldsymbol{\theta}^*, \boldsymbol{\lambda}^{(t)}; \mathbf{w})$$

$$= \frac{1}{T} \sum_{t=1}^{T} \left( \underbrace{\mathcal{L}(\boldsymbol{\theta}^{(t)}, \boldsymbol{\lambda}^*; \mathbf{w}) - \mathcal{L}(\boldsymbol{\theta}^{(t)}, \boldsymbol{\lambda}^{(t)}; \mathbf{w})}_{\text{term A}} \right.$$

$$\left. + \underbrace{\mathcal{L}(\boldsymbol{\theta}^{(t)}, \boldsymbol{\lambda}^{(t)}; \mathbf{w}) - \mathcal{L}(\boldsymbol{\theta}^*, \boldsymbol{\lambda}^{(t)}; \mathbf{w})}_{\text{term B}} \right). \tag{26}$$

Now, we bound A and B by linearity and convexity, respectively:

$$\text{A} = \mathcal{L}(\boldsymbol{\theta}^{(t)}, \boldsymbol{\lambda}^*; \mathbf{w}) - \mathcal{L}(\boldsymbol{\theta}^{(t)}, \boldsymbol{\lambda}^{(t)}; \mathbf{w}) = \langle \nabla_{\boldsymbol{\lambda}} \mathcal{L}(\boldsymbol{\theta}^{(t)}, \boldsymbol{\lambda}^{(t)}; \mathbf{w}), \boldsymbol{\lambda}^* - \boldsymbol{\lambda}^{(t)} \rangle,$$

$$\text{B} = \mathcal{L}(\boldsymbol{\theta}^{(t)}, \boldsymbol{\lambda}^{(t)}; \mathbf{w}) - \mathcal{L}(\boldsymbol{\theta}^*, \boldsymbol{\lambda}^{(t)}; \mathbf{w}) \le \langle \nabla_{\boldsymbol{\theta}} \mathcal{L}(\boldsymbol{\theta}^{(t)}, \boldsymbol{\lambda}^{(t)}; \mathbf{w}), \boldsymbol{\theta}^{(t)} - \boldsymbol{\theta}^* \rangle.$$

We can then continue from (26):

$$\frac{1}{T} \sum_{t=1}^{T} (\text{A} + \text{B}) \le \frac{1}{T} \sum_{t=1}^{T} \left( \langle \nabla_{\boldsymbol{\lambda}} \mathcal{L}(\boldsymbol{\theta}^{(t)}, \boldsymbol{\lambda}^{(t)}; \mathbf{w}), \boldsymbol{\lambda}^* - \boldsymbol{\lambda}^{(t)} \rangle + \langle \nabla_{\boldsymbol{\theta}} \mathcal{L}(\boldsymbol{\theta}^{(t)}, \boldsymbol{\lambda}^{(t)}; \mathbf{w}), \boldsymbol{\theta}^{(t)} - \boldsymbol{\theta}^* \rangle \right.$$

$$= \frac{1}{T} \left( \sum_{t=1}^{T} \langle \delta_{\boldsymbol{\lambda}} \mathcal{L}(\boldsymbol{\theta}^{(t)}, \boldsymbol{\lambda}^{(t)}; \mathbf{w}), \boldsymbol{\lambda}^* - \boldsymbol{\lambda}^{(t)} \rangle \ (\text{term D1}) \right.$$

$$+ \sum_{t=1}^{T} \langle \nabla_{\boldsymbol{\lambda}} \mathcal{L}(\boldsymbol{\theta}^{(t)}, \boldsymbol{\lambda}^{(t)}; \mathbf{w}) - \delta_{\boldsymbol{\lambda}} \mathcal{L}(\boldsymbol{\theta}^{(t)}, \boldsymbol{\lambda}^{(t)}; \mathbf{w}), \boldsymbol{\lambda}^* - \boldsymbol{\lambda}^{(t)} \rangle \ (\text{term E1})$$

$$+ \sum_{t=1}^{T} \langle \delta_{\boldsymbol{\theta}} \mathcal{L}(\boldsymbol{\theta}^{(t)}, \boldsymbol{\lambda}^{(t)}; \mathbf{w}), \boldsymbol{\theta}^{(t)} - \boldsymbol{\theta}^* \rangle \ (\text{term D2})$$

$$+ \sum_{t=1}^{T} \langle \nabla_{\boldsymbol{\theta}} \mathcal{L}(\boldsymbol{\theta}^{(t)}, \boldsymbol{\lambda}^{(t)}; \mathbf{w}) - \delta_{\boldsymbol{\theta}} \mathcal{L}(\boldsymbol{\theta}^{(t)}, \boldsymbol{\lambda}^{(t)}; \mathbf{w}), \boldsymbol{\theta}^{(t)} - \boldsymbol{\theta}^* \rangle \ (\text{term E2}). \tag{27}$$

This is the last reduction on the upper bound. After bounding the above four terms separately, we will be able to arrive at the inequalities in the theorem. All of them can be solved by invoking Lemma A.6. For D1 and D2, it is a direct application. For E1 and E2, we need an additional construction step.

**Bounding D1.** Now, for D1, recall the update rule for $\boldsymbol{\lambda}$:

$$\boldsymbol{\lambda}^{(t+1)} = \underset{\boldsymbol{\lambda} \in \Delta_m}{\arg\max} \ \langle \delta_{\boldsymbol{\lambda}} \mathcal{L}(\boldsymbol{\theta}^{(t)}, \boldsymbol{\lambda}^{(t)}; \mathbf{w}), \boldsymbol{\lambda} \rangle - \frac{1}{\eta_{\boldsymbol{\lambda}}} B_{\psi_{\boldsymbol{\lambda}}}(\boldsymbol{\lambda}; \boldsymbol{\lambda}^{(t)}).$$

By Lemma A.6, substituting $\boldsymbol{\zeta}^{(t)} = \delta_{\boldsymbol{\lambda}} \mathcal{L}(\boldsymbol{\theta}^{(t)}, \boldsymbol{\lambda}^{(t)}; \mathbf{w})$, $\boldsymbol{\nu}^{(t)} = \boldsymbol{\lambda}^{(t)}$, and $\boldsymbol{u} = \boldsymbol{\lambda}^*$, we have:

$$\text{D1} = \sum_{t=1}^{T} \langle \delta_{\boldsymbol{\lambda}} \mathcal{L}(\boldsymbol{\theta}^{(t)}, \boldsymbol{\lambda}^{(t)}; \mathbf{w}), \boldsymbol{\lambda}^* - \boldsymbol{\lambda}^{(t)} \rangle$$

$$\le \frac{1}{\eta_{\boldsymbol{\lambda}}} B_{\psi_{\boldsymbol{\lambda}}}(\boldsymbol{\lambda}^*; \boldsymbol{\lambda}^{(1)}) + \frac{\eta_{\boldsymbol{\lambda}}}{2\mu_{\boldsymbol{\lambda}}} \sum_{t=1}^{T} \|\delta_{\boldsymbol{\lambda}} \mathcal{L}(\boldsymbol{\theta}^{(t)}, \boldsymbol{\lambda}^{(t)}; \mathbf{w})\|_{\boldsymbol{\lambda}, *}^2$$

$$\le \frac{1}{\eta_{\boldsymbol{\lambda}}} D_{\boldsymbol{\lambda}} + \frac{\eta_{\boldsymbol{\lambda}}}{2\mu_{\boldsymbol{\lambda}}} T C_{\boldsymbol{\lambda}} U^2, \tag{28}$$

where $\mu_{\boldsymbol{\lambda}}$ and $\| \cdot \|_{\boldsymbol{\lambda}}$ and are the constant and norm on which the strong convexity of $\psi_{\boldsymbol{\lambda}}$ is defined, and $\| \cdot \|_{\boldsymbol{\lambda}, *}$ is the dual norm of $\| \cdot \|_{\boldsymbol{\lambda}}$. The upper bound of the norm stems from (21) at the beginning of the proof, and $C_{\boldsymbol{\lambda}}$, as well as $D_{\boldsymbol{\lambda}}$, are constants depending on the specific choice of $\psi_{\boldsymbol{\lambda}}$.

**Bounding D2.** Similarly, for D2, recall the update rule for $\boldsymbol{\theta}$:

$$\boldsymbol{\theta}^{(t+1)} = \arg\min_{\boldsymbol{\theta}\in\Theta} \langle \delta_{\boldsymbol{\theta}}\mathcal{L}(\boldsymbol{\theta}^{(t)}, \boldsymbol{\lambda}^{(t)}; \mathbf{w}), \boldsymbol{\theta}\rangle + \frac{1}{\eta_{\boldsymbol{\theta}}} B_{\psi_{\boldsymbol{\theta}}}(\boldsymbol{\theta}; \boldsymbol{\theta}^{(t)}).$$

By Lemma A.6, substituting $\boldsymbol{\zeta}^{(t)} = \delta_{\boldsymbol{\theta}}\mathcal{L}(\boldsymbol{\theta}^{(t)}, \boldsymbol{\lambda}^{(t)}; \mathbf{w})$, $\boldsymbol{\nu}^{(t)} = \boldsymbol{\theta}^{(t)}$, and $\boldsymbol{u} = \boldsymbol{\theta}^*$, we have:

$$\text{D2} = \sum_{t=1}^{T}\langle \delta_{\boldsymbol{\theta}}\mathcal{L}(\boldsymbol{\theta}^{(t)}, \boldsymbol{\lambda}^{(t)}; \mathbf{w}), \boldsymbol{\theta}^{(t)} - \boldsymbol{\theta}^*\rangle \leq \frac{1}{\eta_{\boldsymbol{\theta}}} B_{\psi_{\boldsymbol{\theta}}}(\boldsymbol{\theta}^*; \boldsymbol{\theta}^{(1)}) + \frac{\eta_{\boldsymbol{\theta}}}{2\mu_{\boldsymbol{\theta}}} \sum_{t=1}^{T} \|\delta_{\boldsymbol{\theta}}\mathcal{L}(\boldsymbol{\theta}^{(t)}, \boldsymbol{\lambda}^{(t)}; \mathbf{w})\|_{\boldsymbol{\theta},*}^2$$

$$\leq \frac{1}{\eta_{\boldsymbol{\theta}}} D_{\boldsymbol{\theta}} + \frac{\eta_{\boldsymbol{\theta}}}{2\mu_{\boldsymbol{\theta}}} T C_{\boldsymbol{\theta}} L^2. \tag{29}$$

**Bounding E1.** Next, for E1, the first argument of the inner product no longer matches the update rule for $\boldsymbol{\lambda}^{(t)}$ and thus Lemma A.6 cannot be directly applied. Instead, we construct a sequence $\{\boldsymbol{\chi}^{(t)}\}$ such that

$$\boldsymbol{\chi}^{(1)} = \boldsymbol{\lambda}^{(1)},$$

$$\boldsymbol{\chi}^{(t+1)} = \arg\max_{\boldsymbol{\chi}\in\Delta_m} \langle \nabla_{\boldsymbol{\lambda}}\mathcal{L}(\boldsymbol{\theta}^{(t)}, \boldsymbol{\lambda}^{(t)}; \mathbf{w}) - \delta_{\boldsymbol{\lambda}}\mathcal{L}(\boldsymbol{\theta}^{(t)}, \boldsymbol{\lambda}^{(t)}; \mathbf{w}), \boldsymbol{\chi}\rangle - \frac{1}{\eta_{\boldsymbol{\lambda}}} B_{\psi_{\boldsymbol{\lambda}}}(\boldsymbol{\chi}; \boldsymbol{\chi}^{(t)}).$$

Invoke Lemma A.6 on this sequence: substituting $\boldsymbol{\zeta}^{(t)} = \nabla_{\boldsymbol{\lambda}}\mathcal{L}(\boldsymbol{\theta}^{(t)}, \boldsymbol{\lambda}^{(t)}; \mathbf{w}) - \delta_{\boldsymbol{\lambda}}\mathcal{L}(\boldsymbol{\theta}^{(t)}, \boldsymbol{\lambda}^{(t)}; \mathbf{w})$, $\boldsymbol{\nu}^{(t)} = \boldsymbol{\chi}^{(t)}$, and $\boldsymbol{u} = \boldsymbol{\lambda}^*$, we have:

$$\sum_{t=1}^{T}\langle \nabla_{\boldsymbol{\lambda}}\mathcal{L}(\boldsymbol{\theta}^{(t)}, \boldsymbol{\lambda}^{(t)}; \mathbf{w}) - \delta_{\boldsymbol{\lambda}}\mathcal{L}(\boldsymbol{\theta}^{(t)}, \boldsymbol{\lambda}^{(t)}; \mathbf{w}), \boldsymbol{\lambda}^* - \boldsymbol{\chi}^{(t)}\rangle$$

$$\leq \frac{1}{\eta_{\boldsymbol{\lambda}}} B_{\psi_{\boldsymbol{\lambda}}}(\boldsymbol{\lambda}^*; \boldsymbol{\lambda}^{(1)}) + \frac{\eta_{\boldsymbol{\lambda}}}{2\mu_{\boldsymbol{\lambda}}} \sum_{t=1}^{T} \|\nabla_{\boldsymbol{\lambda}}\mathcal{L}(\boldsymbol{\theta}^{(t)}, \boldsymbol{\lambda}^{(t)}; \mathbf{w}) - \delta_{\boldsymbol{\lambda}}\mathcal{L}(\boldsymbol{\theta}^{(t)}, \boldsymbol{\lambda}^{(t)}; \mathbf{w})\|_{\boldsymbol{\lambda},*}^2$$

$$\leq \frac{1}{\eta_{\boldsymbol{\lambda}}} B_{\psi_{\boldsymbol{\lambda}}}(\boldsymbol{\lambda}^*; \boldsymbol{\lambda}^{(1)}) + \frac{\eta_{\boldsymbol{\lambda}}}{2\mu_{\boldsymbol{\lambda}}} \sum_{t=1}^{T} \left( \|\nabla_{\boldsymbol{\lambda}}\mathcal{L}(\boldsymbol{\theta}^{(t)}, \boldsymbol{\lambda}^{(t)}; \mathbf{w})\|_{\boldsymbol{\lambda},*} + \|\delta_{\boldsymbol{\lambda}}\mathcal{L}(\boldsymbol{\theta}^{(t)}, \boldsymbol{\lambda}^{(t)}; \mathbf{w})\|_{\boldsymbol{\lambda},*} \right)^2$$

$$\leq \frac{1}{\eta_{\boldsymbol{\lambda}}} D_{\boldsymbol{\lambda}} + \frac{\eta_{\boldsymbol{\lambda}}}{2\mu_{\boldsymbol{\lambda}}} T 4 C_{\boldsymbol{\lambda}} U^2. \tag{30}$$

Then, we have for E1:

$$\text{E1} = \sum_{t=1}^{T}\langle \nabla_{\boldsymbol{\lambda}}\mathcal{L}(\boldsymbol{\theta}^{(t)}, \boldsymbol{\lambda}^{(t)}; \mathbf{w}) - \delta_{\boldsymbol{\lambda}}\mathcal{L}(\boldsymbol{\theta}^{(t)}, \boldsymbol{\lambda}^{(t)}; \mathbf{w}), \boldsymbol{\lambda}^* - \boldsymbol{\chi}^{(t)}\rangle$$

$$+ \sum_{t=1}^{T}\langle \nabla_{\boldsymbol{\lambda}}\mathcal{L}(\boldsymbol{\theta}^{(t)}, \boldsymbol{\lambda}^{(t)}; \mathbf{w}) - \delta_{\boldsymbol{\lambda}}\mathcal{L}(\boldsymbol{\theta}^{(t)}, \boldsymbol{\lambda}^{(t)}; \mathbf{w}), \boldsymbol{\chi}^{(t)} - \boldsymbol{\lambda}^{(t)}\rangle, \tag{31}$$

where the first term is bounded by (30). Different ways to deal with the second term result in the expectation bound and the high-probability bound, respectively. First, we have by the total law of expectation:

$$\mathbb{E}\left[\sum_{t=1}^{T}\langle \nabla_{\boldsymbol{\lambda}}\mathcal{L}(\boldsymbol{\theta}^{(t)}, \boldsymbol{\lambda}^{(t)}; \mathbf{w}) - \delta_{\boldsymbol{\lambda}}\mathcal{L}(\boldsymbol{\theta}^{(t)}, \boldsymbol{\lambda}^{(t)}; \mathbf{w}), \boldsymbol{\chi}^{(t)} - \boldsymbol{\lambda}^{(t)}\rangle\right]$$

$$= \sum_{t=1}^{T}\mathbb{E}\left[\mathbb{E}\left[\langle \nabla_{\boldsymbol{\lambda}}\mathcal{L}(\boldsymbol{\theta}^{(t)}, \boldsymbol{\lambda}^{(t)}; \mathbf{w}) - \delta_{\boldsymbol{\lambda}}\mathcal{L}(\boldsymbol{\theta}^{(t)}, \boldsymbol{\lambda}^{(t)}; \mathbf{w}), \boldsymbol{\chi}^{(t)} - \boldsymbol{\lambda}^{(t)}\rangle \mid \eta_1, \ldots, \eta_{t-1}\right]\right]$$

$$= \sum_{t=1}^{T}\mathbb{E}\left[\langle \boldsymbol{\chi}^{(t)} - \boldsymbol{\lambda}^{(t)}, \mathbb{E}\left[\nabla_{\boldsymbol{\lambda}}\mathcal{L}(\boldsymbol{\theta}^{(t)}, \boldsymbol{\lambda}^{(t)}; \mathbf{w}) - \delta_{\boldsymbol{\lambda}}\mathcal{L}(\boldsymbol{\theta}^{(t)}, \boldsymbol{\lambda}^{(t)}; \mathbf{w}) \mid \eta_1, \ldots, \eta_{t-1}\right]\rangle\right] \tag{32}$$

$$= \sum_{t=1}^{T}\mathbb{E}\left[\langle \boldsymbol{\chi}^{(t)} - \boldsymbol{\lambda}^{(t)}, \mathbf{0}\rangle\right] = 0, \tag{33}$$

where $\{\eta_t\}$ are the random batch sampled in the past to compute stochastic gradient. Here, (32) is based on the fact that both $\boldsymbol{\chi}^{(t)}$ and $\boldsymbol{\lambda}^{(t)}$ are deterministic given $\delta_{\boldsymbol{\lambda}}\mathcal{L}(\boldsymbol{\theta}^{(t-1)}, \boldsymbol{\lambda}^{(t-1)}; \mathbf{w})$, $\boldsymbol{\theta}^{(t-1)}$, and $\boldsymbol{\lambda}^{(t-1)}$, which are all deterministic given up to the $(t-1)$-th random batch. The next equality (33) is based on the fact that the stochastic gradient is an unbiased estimator of the true gradient.

Next, the key observation to obtain the high-probability bound is that $X_t = \sum_{\tau=1}^{t}\langle\nabla_{\boldsymbol{\lambda}}\mathcal{L}(\boldsymbol{\theta}^{(\tau)}, \boldsymbol{\lambda}^{(\tau)}; \mathbf{w}) - \delta_{\boldsymbol{\lambda}}\mathcal{L}(\boldsymbol{\theta}^{(\tau)}, \boldsymbol{\lambda}^{(\tau)}; \mathbf{w}), \boldsymbol{\chi}^{(\tau)} - \boldsymbol{\lambda}^{(\tau)}\rangle$ is a martingale w.r.t. filtration $\{\eta_t\}$. We can check this by definition:

$$\mathbb{E}\left[|X_t|\right] = \mathbb{E}\left[|\sum_{\tau=1}^{t}\langle\nabla_{\boldsymbol{\lambda}}\mathcal{L}(\boldsymbol{\theta}^{(\tau)}, \boldsymbol{\lambda}^{(\tau)}; \mathbf{w}) - \delta_{\boldsymbol{\lambda}}\mathcal{L}(\boldsymbol{\theta}^{(\tau)}, \boldsymbol{\lambda}^{(\tau)}; \mathbf{w}), \boldsymbol{\chi}^{(\tau)} - \boldsymbol{\lambda}^{(\tau)}\rangle|\right] < \infty$$

$$\mathbb{E}\left[X_t \mid \eta_1, \ldots, \eta_{t-1}\right] = \mathbb{E}\left[\langle\nabla_{\boldsymbol{\lambda}}\mathcal{L}(\boldsymbol{\theta}^{(t)}, \boldsymbol{\lambda}^{(t)}; \mathbf{w}) - \delta_{\boldsymbol{\lambda}}\mathcal{L}(\boldsymbol{\theta}^{(t)}, \boldsymbol{\lambda}^{(t)}; \mathbf{w}), \boldsymbol{\chi}^{(t)} - \boldsymbol{\lambda}^{(t)}\rangle \mid \eta_1, \ldots, \eta_{t-1}\right]$$
$$+ \mathbb{E}\left[X_{t-1} \mid \eta_1, \ldots, \eta_{t-1}\right]$$
$$= 0 + X_{t-1} \text{ (same argument as (32) and (33))} = X_{t-1}.$$

We can now bound the second term by the Azuma-Hoeffding inequality in Lemma A.5. We have the constant $c_t$ from:

$$|X_t - X_{t-1}| = |\langle\nabla_{\boldsymbol{\lambda}}\mathcal{L}(\boldsymbol{\theta}^{(t)}, \boldsymbol{\lambda}^{(t)}; \mathbf{w}) - \delta_{\boldsymbol{\lambda}}\mathcal{L}(\boldsymbol{\theta}^{(t)}, \boldsymbol{\lambda}^{(t)}; \mathbf{w}), \boldsymbol{\chi}^{(t)} - \boldsymbol{\lambda}^{(t)}\rangle|$$
$$\leq \|\nabla_{\boldsymbol{\lambda}}\mathcal{L}(\boldsymbol{\theta}^{(t)}, \boldsymbol{\lambda}^{(t)}; \mathbf{w}) - \delta_{\boldsymbol{\lambda}}\mathcal{L}(\boldsymbol{\theta}^{(t)}, \boldsymbol{\lambda}^{(t)}; \mathbf{w})\|_{\infty}\|\boldsymbol{\chi}^{(t)} - \boldsymbol{\lambda}^{(t)}\|_1 \qquad (34)$$
$$\leq \left(\|\nabla_{\boldsymbol{\lambda}}\mathcal{L}(\boldsymbol{\theta}^{(t)}, \boldsymbol{\lambda}^{(t)}; \mathbf{w})\|_{\infty} + \|\delta_{\boldsymbol{\lambda}}\mathcal{L}(\boldsymbol{\theta}^{(t)}, \boldsymbol{\lambda}^{(t)}; \mathbf{w})\|_{\infty}\right)\left(\|\boldsymbol{\chi}^{(1)}\|_1 + \|\boldsymbol{\lambda}^{(t)}\|_1\right)$$
$$\leq 2U \times 2 = 4U,$$

where (34) is based on Lemma A.4 and that $\|\cdot\|_{\infty}$ is the dual norm of $\|\cdot\|_1$. Then, given that $X_0 = 0$, we have for any positive reals $\epsilon$:

$$P(X_T \leq \epsilon) = P\left(\sum_{t=1}^{T}\langle\nabla_{\boldsymbol{\lambda}}\mathcal{L}(\boldsymbol{\theta}^{(t)}, \boldsymbol{\lambda}^{(t)}; \mathbf{w}) - \delta_{\boldsymbol{\lambda}}\mathcal{L}(\boldsymbol{\theta}^{(t)}, \boldsymbol{\lambda}^{(t)}; \mathbf{w}), \boldsymbol{\chi}^{(t)} - \boldsymbol{\lambda}^{(t)}\rangle \leq \epsilon\right)$$
$$\geq 1 - \exp(\frac{-\epsilon^2}{32TU^2}).$$

Let $\gamma = \exp(\frac{-\epsilon^2}{32TU^2})$, we have $0 < \gamma < 1$ and $\epsilon = 4U\sqrt{2T\log\frac{1}{\gamma}}$. Therefore, we have

$$P\left(\sum_{t=1}^{T}\langle\nabla_{\boldsymbol{\lambda}}\mathcal{L}(\boldsymbol{\theta}^{(t)}, \boldsymbol{\lambda}^{(t)}; \mathbf{w}) - \delta_{\boldsymbol{\lambda}}\mathcal{L}(\boldsymbol{\theta}^{(t)}, \boldsymbol{\lambda}^{(t)}; \mathbf{w}), \boldsymbol{\chi}^{(t)} - \boldsymbol{\lambda}^{(t)}\rangle \leq 4U\sqrt{2T\log\frac{1}{\gamma}}\right) \geq 1 - \gamma. \qquad (35)$$

Combining (30), (31), (33), and (35), we obtain the two types of bounds for E1:

$$\mathbb{E}\left[\text{E1}\right] \leq \frac{D_{\boldsymbol{\lambda}}}{\eta_{\boldsymbol{\lambda}}} + \frac{2\eta_{\boldsymbol{\lambda}}TC_{\boldsymbol{\lambda}}U^2}{\mu_{\boldsymbol{\lambda}}} + 0, \qquad (36)$$

and with probability at least $1 - \gamma$, $0 < \gamma < 1$:

$$\text{E1} \leq \frac{D_{\boldsymbol{\lambda}}}{\eta_{\boldsymbol{\lambda}}} + \frac{2\eta_{\boldsymbol{\lambda}}TC_{\boldsymbol{\lambda}}U^2}{\mu_{\boldsymbol{\lambda}}} + 4U\sqrt{2T\log\frac{1}{\gamma}}. \qquad (37)$$

**Bounding E2.** We simply go through a similar process. We explicitly present the steps for completeness. First, construct a sequence $\{\boldsymbol{v}^{(t)}\}$ such that

$$\boldsymbol{v}^{(1)} = \boldsymbol{\theta}^{(1)}$$

$$\boldsymbol{v}^{(t+1)} = \arg\min_{\boldsymbol{v}\in\Theta} \langle\nabla_{\boldsymbol{\theta}}\mathcal{L}(\boldsymbol{\theta}^{(t)}, \boldsymbol{\lambda}^{(t)}; \mathbf{w}) - \delta_{\boldsymbol{\theta}}\mathcal{L}(\boldsymbol{\theta}^{(t)}, \boldsymbol{\lambda}^{(t)}; \mathbf{w}), \boldsymbol{v}\rangle + \frac{1}{\eta_{\boldsymbol{\theta}}}B_{\psi_{\boldsymbol{\theta}}}(\boldsymbol{v}; \boldsymbol{v}^{(t)}).$$

By Lemma A.6, we have

$$\sum_{t=1}^{T} \langle \nabla_{\boldsymbol{\theta}} \mathcal{L}(\boldsymbol{\theta}^{(t)}, \boldsymbol{\lambda}^{(t)}; \mathbf{w}) - \delta_{\boldsymbol{\theta}} \mathcal{L}(\boldsymbol{\theta}^{(t)}, \boldsymbol{\lambda}^{(t)}; \mathbf{w}), \boldsymbol{v}^{(t)} - \boldsymbol{\theta}^* \rangle$$

$$\leq \frac{1}{\eta_{\boldsymbol{\theta}}} B_{\psi_{\boldsymbol{\theta}}}(\boldsymbol{\theta}^*; \boldsymbol{\theta}^{(1)}) + \frac{\eta_{\boldsymbol{\theta}}}{2\mu_{\boldsymbol{\theta}}} \sum_{t=1}^{T} \|\nabla_{\boldsymbol{\theta}} \mathcal{L}(\boldsymbol{\theta}^{(t)}, \boldsymbol{\lambda}^{(t)}; \mathbf{w}) - \delta_{\boldsymbol{\theta}} \mathcal{L}(\boldsymbol{\theta}^{(t)}, \boldsymbol{\lambda}^{(t)}; \mathbf{w})\|_{\boldsymbol{\theta}, *}^2$$

$$\leq \frac{1}{\eta_{\boldsymbol{\theta}}} B_{\psi_{\boldsymbol{\theta}}}(\boldsymbol{\theta}^*; \boldsymbol{\theta}^{(1)}) + \frac{\eta_{\boldsymbol{\theta}}}{2\mu_{\boldsymbol{\theta}}} \sum_{t=1}^{T} \left( \|\nabla_{\boldsymbol{\theta}} \mathcal{L}(\boldsymbol{\theta}^{(t)}, \boldsymbol{\lambda}^{(t)}; \mathbf{w})\|_{\boldsymbol{\theta}, *} + \|\delta_{\boldsymbol{\theta}} \mathcal{L}(\boldsymbol{\theta}^{(t)}, \boldsymbol{\lambda}^{(t)}; \mathbf{w})\|_{\boldsymbol{\theta}, *} \right)^2$$

$$\leq \frac{1}{\eta_{\boldsymbol{\theta}}} D_{\boldsymbol{\theta}} + \frac{\eta_{\boldsymbol{\theta}}}{2\mu_{\boldsymbol{\theta}}} T 4 C_{\boldsymbol{\theta}} L^2. \tag{38}$$

Then, we have for E2:

$$\text{E2} = \sum_{t=1}^{T} \langle \nabla_{\boldsymbol{\theta}} \mathcal{L}(\boldsymbol{\theta}^{(t)}, \boldsymbol{\lambda}^{(t)}; \mathbf{w}) - \delta_{\boldsymbol{\theta}} \mathcal{L}(\boldsymbol{\theta}^{(t)}, \boldsymbol{\lambda}^{(t)}; \mathbf{w}), \boldsymbol{\theta}^{(t)} - \boldsymbol{v}^{(t)} \rangle$$

$$+ \sum_{t=1}^{T} \langle \nabla_{\boldsymbol{\theta}} \mathcal{L}(\boldsymbol{\theta}^{(t)}, \boldsymbol{\lambda}^{(t)}; \mathbf{w}) - \delta_{\boldsymbol{\theta}} \mathcal{L}(\boldsymbol{\theta}^{(t)}, \boldsymbol{\lambda}^{(t)}; \mathbf{w}), \boldsymbol{v}^{(t)} - \boldsymbol{\theta}^* \rangle,$$

where the second term is bounded by (38). For the first term, we have:

$$\mathbb{E} \left[ \sum_{t=1}^{T} \langle \nabla_{\boldsymbol{\theta}} \mathcal{L}(\boldsymbol{\theta}^{(t)}, \boldsymbol{\lambda}^{(t)}; \mathbf{w}) - \delta_{\boldsymbol{\theta}} \mathcal{L}(\boldsymbol{\theta}^{(t)}, \boldsymbol{\lambda}^{(t)}; \mathbf{w}), \boldsymbol{\theta}^{(t)} - \boldsymbol{v}^{(t)} \rangle \right]$$

$$= \sum_{t=1}^{T} \mathbb{E} \left[ \mathbb{E} \left[ \langle \nabla_{\boldsymbol{\theta}} \mathcal{L}(\boldsymbol{\theta}^{(t)}, \boldsymbol{\lambda}^{(t)}; \mathbf{w}) - \delta_{\boldsymbol{\theta}} \mathcal{L}(\boldsymbol{\theta}^{(t)}, \boldsymbol{\lambda}^{(t)}; \mathbf{w}), \boldsymbol{\theta}^{(t)} - \boldsymbol{v}^{(t)} \rangle \mid \eta_1, \ldots, \eta_{t-1} \right] \right]$$

$$= \sum_{t=1}^{T} \mathbb{E} \left[ \langle \boldsymbol{\theta}^{(t)} - \boldsymbol{v}^{(t)}, \mathbb{E} \left[ \nabla_{\boldsymbol{\theta}} \mathcal{L}(\boldsymbol{\theta}^{(t)}, \boldsymbol{\lambda}^{(t)}; \mathbf{w}) - \delta_{\boldsymbol{\theta}} \mathcal{L}(\boldsymbol{\theta}^{(t)}, \boldsymbol{\lambda}^{(t)}; \mathbf{w}) \mid \eta_1, \ldots, \eta_{t-1} \right] \rangle \right] \tag{39}$$

$$= \sum_{t=1}^{T} \mathbb{E} \left[ \langle \boldsymbol{\theta}^{(t)} - \boldsymbol{v}^{(t)}, \mathbf{0} \rangle \right] = 0, \tag{40}$$

where (39) is based on the fact that both $\boldsymbol{\theta}^{(t)}$ and $\boldsymbol{v}^{(t)}$ are deterministic given $\delta_{\boldsymbol{\theta}} \mathcal{L}(\boldsymbol{\theta}^{(t-1)}, \boldsymbol{\lambda}^{(t-1)}; \mathbf{w}) = \boldsymbol{\lambda}^{(t-1)} \odot \mathbf{w} \odot \delta \mathbf{f}(\boldsymbol{\theta}^{(t-1)})$, $\boldsymbol{\theta}^{(t-1)}$, and $\boldsymbol{\lambda}^{(t-1)}$, which are all deterministic given up to the $(t-1)$-th random batch. And (40) is still based on the fact that the stochastic gradient is an unbiased estimator of the true gradient. With the same argument for (39) and (40), one can check that $Y_t = \sum_{\tau=1}^{t} \langle \nabla_{\boldsymbol{\theta}} \mathcal{L}(\boldsymbol{\theta}^{(\tau)}, \boldsymbol{\lambda}^{(\tau)}; \mathbf{w}) - \delta_{\boldsymbol{\theta}} \mathcal{L}(\boldsymbol{\theta}^{(\tau)}, \boldsymbol{\lambda}^{(\tau)}; \mathbf{w}), \boldsymbol{\theta}^{(\tau)} - \boldsymbol{v}^{(\tau)} \rangle$ is a martingale w.r.t. filtration $\{\eta_t\}$, and we have

$$|Y_t - Y_{t-1}| = |\langle \nabla_{\boldsymbol{\theta}} \mathcal{L}(\boldsymbol{\theta}^{(t)}, \boldsymbol{\lambda}^{(t)}; \mathbf{w}) - \delta_{\boldsymbol{\theta}} \mathcal{L}(\boldsymbol{\theta}^{(t)}, \boldsymbol{\lambda}^{(t)}; \mathbf{w}), \boldsymbol{\theta}^{(t)} - \boldsymbol{v}^{(t)} \rangle|$$

$$\leq \|\nabla_{\boldsymbol{\theta}} \mathcal{L}(\boldsymbol{\theta}^{(t)}, \boldsymbol{\lambda}^{(t)}; \mathbf{w}) - \delta_{\boldsymbol{\theta}} \mathcal{L}(\boldsymbol{\theta}^{(t)}, \boldsymbol{\lambda}^{(t)}; \mathbf{w})\|_{\infty} \|\boldsymbol{\theta}^{(t)} - \boldsymbol{v}^{(t)}\|_1$$

$$\leq \left( \|\nabla_{\boldsymbol{\theta}} \mathcal{L}(\boldsymbol{\theta}^{(t)}, \boldsymbol{\lambda}^{(t)}; \mathbf{w})\|_{\infty} + \|\delta_{\boldsymbol{\theta}} \mathcal{L}(\boldsymbol{\theta}^{(t)}, \boldsymbol{\lambda}^{(t)}; \mathbf{w})\|_{\infty} \right) \left( \|\boldsymbol{\theta}^{(1)}\|_1 + \|\boldsymbol{v}^{(t)}\|_1 \right)$$

$$\leq 2L \times 2dR_{\boldsymbol{\theta}} = 4dR_{\boldsymbol{\theta}} L,$$

where the last inequality is based on $\|\boldsymbol{\theta}\|_1 \leq d\|\boldsymbol{\theta}\|_{\infty} \leq dR_{\boldsymbol{\theta}}$ and the same for $\boldsymbol{v}^{(t)}$. Then, by the Azuma-Hoeffding inequality, we have for $0 < \gamma < 1$:

$$P \left( \sum_{t=1}^{T} \langle \nabla_{\boldsymbol{\theta}} \mathcal{L}(\boldsymbol{\theta}^{(t)}, \boldsymbol{\lambda}^{(t)}; \mathbf{w}) - \delta_{\boldsymbol{\theta}} \mathcal{L}(\boldsymbol{\theta}^{(t)}, \boldsymbol{\lambda}^{(t)}; \mathbf{w}), \boldsymbol{\theta}^{(t)} - \boldsymbol{v}^{(t)} \rangle \leq 4dR_{\boldsymbol{\theta}} L \sqrt{2T \log \frac{1}{\gamma}} \right) \geq 1 - \gamma.$$

Therefore, for E2, we have the two types of bounds as follows:

$$\mathbb{E}[\text{E2}] \leq \frac{D_{\boldsymbol{\theta}}}{\eta_{\boldsymbol{\theta}}} + \frac{2\eta_{\boldsymbol{\theta}} T C_{\boldsymbol{\theta}} L^2}{\mu_{\boldsymbol{\theta}}} + 0, \tag{41}$$

and with probability at least $1 - \gamma$, $0 < \gamma < 1$:

$$\text{E2} \leq \frac{D_{\boldsymbol{\theta}}}{\eta_{\boldsymbol{\theta}}} + \frac{2\eta_{\boldsymbol{\theta}}TC_{\boldsymbol{\theta}}L^2}{\mu_{\boldsymbol{\theta}}} + 4dR_{\boldsymbol{\theta}}L\sqrt{2T\log\frac{1}{\gamma}}. \tag{42}$$

**Finalize.** At last, we are able to assemble the bounds for terms D1, D2, E1, and E2 to bound (27), and thus bound the convergence error. For the expectation bound, with (28) for D1, (29) for D2, (36) for E1, and (41) for E2, we have

$$\mathbb{E}\left[\text{TCH}(\hat{\boldsymbol{\theta}}; \mathbf{w})\right] - \min_{\boldsymbol{\theta} \in \Theta} \text{TCH}(\boldsymbol{\theta}; \mathbf{w}) \leq \frac{1}{T}\left(\underbrace{\frac{2D_{\boldsymbol{\theta}}}{\eta_{\boldsymbol{\theta}}} + \frac{5\eta_{\boldsymbol{\theta}}TC_{\boldsymbol{\theta}}L^2}{2\mu_{\boldsymbol{\theta}}}}_{\text{from D2 and E2}} + \underbrace{\frac{2D_{\boldsymbol{\lambda}}}{\eta_{\boldsymbol{\lambda}}} + \frac{5\eta_{\boldsymbol{\lambda}}TC_{\boldsymbol{\lambda}}U^2}{2\mu_{\boldsymbol{\lambda}}}}_{\text{from D1 and E1}}\right).$$

When $\eta_{\boldsymbol{\theta}} = \sqrt{\frac{4\mu_{\boldsymbol{\theta}}D_{\boldsymbol{\theta}}}{5TC_{\boldsymbol{\theta}}L^2}}$ and $\eta_{\boldsymbol{\lambda}} = \sqrt{\frac{4\mu_{\boldsymbol{\lambda}}D_{\boldsymbol{\lambda}}}{5TC_{\boldsymbol{\lambda}}U^2}}$, we further have:

$$\mathbb{E}\left[\text{TCH}(\hat{\boldsymbol{\theta}}; \mathbf{w})\right] - \min_{\boldsymbol{\theta} \in \Theta} \text{TCH}(\boldsymbol{\theta}; \mathbf{w}) \leq \sqrt{\frac{20D_{\boldsymbol{\theta}}C_{\boldsymbol{\theta}}L^2}{\mu_{\boldsymbol{\theta}}T}} + \sqrt{\frac{20D_{\boldsymbol{\lambda}}C_{\boldsymbol{\lambda}}U^2}{\mu_{\boldsymbol{\lambda}}T}},$$

which proves the expectation bound in Theorem A.1. For the high-probability bound, with (28) for D1, (29) for D2, (37) for E1, and (42) for E2, we have with probability at least $1 - \gamma$, $0 < \gamma < 1$:

$$\text{TCH}(\hat{\boldsymbol{\theta}}; \mathbf{w}) - \min_{\boldsymbol{\theta} \in \Theta} \text{TCH}(\boldsymbol{\theta}; \mathbf{w}) \leq \frac{1}{T}\left(\frac{2D_{\boldsymbol{\theta}}}{\eta_{\boldsymbol{\theta}}} + \frac{5\eta_{\boldsymbol{\theta}}TC_{\boldsymbol{\theta}}L^2}{2\mu_{\boldsymbol{\theta}}} + \frac{3D_{\boldsymbol{\lambda}}}{\eta_{\boldsymbol{\lambda}}} + \frac{9\eta_{\boldsymbol{\lambda}}TC_{\boldsymbol{\lambda}}U^2}{2\mu_{\boldsymbol{\lambda}}}\right.$$
$$\left. + \underbrace{4dR_{\boldsymbol{\theta}}L\sqrt{2T\log\frac{1}{\gamma}}}_{\text{from E2}} + \underbrace{4U\sqrt{2T\log\frac{1}{\gamma}}}_{\text{from E1}}\right).$$

With the same optimal step sizes, we have with probability at least $1 - \gamma$, $0 < \gamma < 1$:

$$\text{TCH}(\hat{\boldsymbol{\theta}}; \mathbf{w}) - \min_{\boldsymbol{\theta} \in \Theta} \text{TCH}(\boldsymbol{\theta}; \mathbf{w}) \leq \sqrt{\frac{20D_{\boldsymbol{\theta}}C_{\boldsymbol{\theta}}L^2}{\mu_{\boldsymbol{\theta}}T}} + \sqrt{\frac{20D_{\boldsymbol{\lambda}}C_{\boldsymbol{\lambda}}U^2}{\mu_{\boldsymbol{\lambda}}T}} + 4(dR_{\boldsymbol{\theta}}L + U)\sqrt{\frac{2}{T}\log\frac{1}{\gamma}},$$

which completes the proof of Theorem 3.2. $\qquad\qquad\square$

## A.2   Corollaries 3.4 and 3.5: expectation bounds, proofs, and analysis on the p-norm instance

**Corollary A.7** (Expectation bound for Corollary 3.4). *Suppose Assumption 3.1 holds. Using PGD for both $\boldsymbol{\theta}$ and $\boldsymbol{\lambda}$, with the same optimal step sizes $\eta_{\boldsymbol{\theta}}$ and $\eta_{\boldsymbol{\lambda}}$ as in Corollary 3.4, both Algorithms 1 and 2 converge as:*

$$\mathbb{E}\left[\text{TCH}(\hat{\boldsymbol{\theta}}; \mathbf{w})\right] - \min_{\boldsymbol{\theta} \in \Theta} \text{TCH}(\boldsymbol{\theta}; \mathbf{w}) \leq \frac{2\sqrt{10}dR_{\boldsymbol{\theta}}L}{\sqrt{T}} + \frac{2\sqrt{10}\sqrt{m}U}{\sqrt{T}}. \tag{43}$$

**Corollary A.8** (Expectation bound for Corollary 3.5). *Suppose Assumption 3.1 holds. Using PGD for $\boldsymbol{\theta}$ and EG for $\boldsymbol{\lambda}$, with the same optimal step sizes $\eta_{\boldsymbol{\theta}}$ and $\eta_{\boldsymbol{\lambda}}$ as in Corollary 3.5, both Algorithms 1 and 2 converge as:*

$$\mathbb{E}\left[\text{TCH}(\hat{\boldsymbol{\theta}}; \mathbf{w})\right] - \min_{\boldsymbol{\theta} \in \Theta} \text{TCH}(\boldsymbol{\theta}; \mathbf{w}) \leq \frac{2\sqrt{10}dR_{\boldsymbol{\theta}}L}{\sqrt{T}} + \frac{2\sqrt{5}\sqrt{\log m}U}{\sqrt{T}}. \tag{44}$$

*Proof of Corollaries 3.4, 3.5, A.7 and A.8.* Given the general bounds in Theorems 3.2 and A.1, the four corollaries can be easily derived by instantiating the constants associated with the specific choices of $\psi$, namely $\mu_{\boldsymbol{\theta}}$, $\mu_{\boldsymbol{\lambda}}$, $D_{\boldsymbol{\theta}}$, $D_{\boldsymbol{\lambda}}$, $C_{\boldsymbol{\theta}}$, and $C_{\boldsymbol{\lambda}}$. Recall their meanings as follows:

- $\psi_{\boldsymbol{\theta}}$ is $\mu_{\boldsymbol{\theta}}$-strongly convex w.r.t. norm $\|\cdot\|_{\boldsymbol{\theta}}$. $\psi_{\boldsymbol{\lambda}}$ is $\mu_{\boldsymbol{\lambda}}$-strongly convex w.r.t. norm $\|\cdot\|_{\boldsymbol{\lambda}}$.

- $B_{\psi_{\boldsymbol{\theta}}}(\boldsymbol{\theta}^*; \boldsymbol{\theta}^{(1)}) \leq D_{\boldsymbol{\theta}}$. $B_{\psi_{\boldsymbol{\lambda}}}(\boldsymbol{\lambda}^*; \boldsymbol{\lambda}^{(1)}) \leq D_{\boldsymbol{\lambda}}$.

- $\|\nabla_{\boldsymbol{\theta}}\mathcal{L}(\boldsymbol{\theta}^{(t)}, \boldsymbol{\lambda}^{(t+1)}; \mathbf{w})\|_{\boldsymbol{\theta},*}^2 \leq C_{\boldsymbol{\theta}}L^2$, $\|\nabla_{\boldsymbol{\lambda}}\mathcal{L}(\boldsymbol{\theta}^{(t)}, \boldsymbol{\lambda}^{(t)}; \mathbf{w})\|_{\boldsymbol{\lambda},*}^2 \leq C_{\boldsymbol{\lambda}}U^2$, with the same inequalities for the corresponding stochastic gradients, where $\|\cdot\|_*$ denotes the dual norm.

We now discuss the cases for Projected Gradient Descent (PGD) and Exponentiated Gradient (EG), respectively, resulting in the corresponding terms in the four theorems. In the following, we use $\mathbf{x}$ to denote either $\boldsymbol{\theta}$ or $\boldsymbol{\lambda}$ if some property holds for both of them and $\dim(\mathbf{x})$ for the dimension of $\mathbf{x}$.

**Projected Gradient Descent.**

- The $l$-2 norm induced $\psi(\mathbf{x}) = \frac{1}{2}\|\mathbf{x}\|_2^2$ is 1-strongly convex w.r.t. $\|\cdot\|_2$. Hence, $\mu_{\mathbf{x}} = 1$.
- For $B_{\psi_{\mathbf{x}}}(\mathbf{x}^*; \mathbf{x}^{(1)})$, we have:

$$B_{\psi_{\mathbf{x}}}(\mathbf{x}^*; \mathbf{x}^{(1)}) = \frac{1}{2}\|\mathbf{x}^* - \mathbf{x}^{(1)}\|_2^2 \leq \frac{1}{2}(\|\mathbf{x}^*\|_2 + \|\mathbf{x}^{(1)}\|_2)^2.$$

Recall that $\|\boldsymbol{\theta}\|_{\infty} \leq R_{\boldsymbol{\theta}}$, hence, for $\boldsymbol{\theta} \in \mathbb{R}^d$, we have $\frac{1}{2}(\|\boldsymbol{\theta}^*\|_2 + \|\boldsymbol{\theta}^{(1)}\|_2)^2 \leq \frac{1}{2}(2\sqrt{d}R_{\boldsymbol{\theta}})^2 = 2dR_{\boldsymbol{\theta}}^2$. For $\boldsymbol{\lambda} \in \Delta_m$, we have $\frac{1}{2}(\|\boldsymbol{\lambda}^*\|_2 + \|\boldsymbol{\lambda}^{(1)}\|_2)^2 \leq \frac{1}{2}(\|\boldsymbol{\lambda}^*\|_1 + \|\boldsymbol{\lambda}^{(1)}\|_1)^2 \leq \frac{1}{2}(2)^2 = 2$. Hence, $D_{\boldsymbol{\theta}} = 2dR_{\boldsymbol{\theta}}^2$ and $D_{\boldsymbol{\lambda}} = 2$.

- For $C_{\mathbf{x}}$, since the dual norm of $l$-2 norm is still $l$-2 norm, we have for any gradient $\nabla_{\mathbf{x}}$,

$$\|\nabla_{\mathbf{x}}\|_{\mathbf{x},*}^2 = \|\nabla_{\mathbf{x}}\|_2^2 \leq \left(\sqrt{\dim(\mathbf{x})\|\nabla_{\mathbf{x}}\|_{\infty}^2}\right)^2 \leq \dim(\mathbf{x})\|\nabla_{\mathbf{x}}\|_{\infty}^2.$$

Given that $\|\nabla_{\boldsymbol{\theta}}\|_{\infty} \leq L$, $\|\nabla_{\boldsymbol{\lambda}}\|_{\infty} \leq U$, we have $\|\nabla_{\boldsymbol{\theta}}\|_{\boldsymbol{\theta},*}^2 \leq dL^2$, $\|\nabla_{\boldsymbol{\lambda}}\|_{\boldsymbol{\lambda},*}^2 \leq mU^2$. The same inequalities hold for stochastic gradients. Hence, $C_{\boldsymbol{\theta}} = d$ and $C_{\boldsymbol{\lambda}} = m$.

Therefore, when applying PGD to $\boldsymbol{\theta}$, by substituting the above constants into expressions in Theorem 3.2, we have the optimal step size $\eta_{\boldsymbol{\theta}} = \sqrt{\frac{4\mu_{\boldsymbol{\theta}}D_{\boldsymbol{\theta}}}{5TC_{\boldsymbol{\theta}}L^2}} = \sqrt{\frac{4\cdot 1\cdot 2dR_{\boldsymbol{\theta}}^2}{5TdL^2}} = \sqrt{\frac{8R_{\boldsymbol{\theta}}^2}{5TL^2}}$, and the first term in (8), (43), (9), and (44):

$$\sqrt{\frac{20D_{\boldsymbol{\theta}}C_{\boldsymbol{\theta}}L^2}{\mu_{\boldsymbol{\theta}}T}} = \sqrt{\frac{20\cdot 2dR_{\boldsymbol{\theta}}^2 \cdot d \cdot L^2}{1\cdot T}} = \frac{2\sqrt{10}dR_{\boldsymbol{\theta}}L}{\sqrt{T}}.$$

Similarly, when applying PGD to $\boldsymbol{\lambda}$, we have the optimal step size $\eta_{\boldsymbol{\lambda}} = \sqrt{\frac{4\mu_{\boldsymbol{\lambda}}D_{\boldsymbol{\lambda}}}{5TC_{\boldsymbol{\lambda}}U^2}} = \sqrt{\frac{4\cdot 1\cdot 2}{5TmU^2}} = \sqrt{\frac{8}{5TmU^2}}$, and the second term in (8) and (43):

$$\sqrt{\frac{20D_{\boldsymbol{\lambda}}C_{\boldsymbol{\lambda}}U^2}{\mu_{\boldsymbol{\lambda}}T}} = \sqrt{\frac{20\cdot 2\cdot m \cdot U^2}{1\cdot T}} = \frac{2\sqrt{10}\sqrt{m}U}{\sqrt{T}}.$$

**Exponentiated Gradient (for $\boldsymbol{\lambda} \in \Delta_m$ only).**

- The negative entropy $\psi(\boldsymbol{\lambda}) = \sum_{i=1}^m \lambda_i \log \lambda_i$ is 1-strongly convex w.r.t. $\|\cdot\|_1$. Hence, $\mu_{\boldsymbol{\lambda}} = 1$.
- For $B_{\psi_{\boldsymbol{\lambda}}}(\boldsymbol{\lambda}^*; \boldsymbol{\lambda}^{(1)})$, with initialization $\boldsymbol{\lambda}^{(1)} = \left[\frac{1}{m}, \cdots, \frac{1}{m}\right]^{\top}$, we have:

$$B_{\psi_{\boldsymbol{\lambda}}}(\boldsymbol{\lambda}^*; \boldsymbol{\lambda}^{(1)}) = \sum_{i=1}^m \lambda_i^* \log \frac{\lambda_i^*}{\lambda_i^{(1)}} = \left(\sum_{i=1}^m \lambda_i^* \log \lambda_i^*\right) + \log m \leq \log m.$$

Hence, $D_{\boldsymbol{\lambda}} = \log m$.

- For $C_{\boldsymbol{\lambda}}$, since the dual norm of $l$-1 norm is the infinity norm, we have for any gradient $\nabla_{\boldsymbol{\lambda}}$, $\|\nabla_{\boldsymbol{\lambda}}\|_{\infty}^2 \leq U^2$. The same holds for the $\delta_{\boldsymbol{\lambda}}$. Hence, $C_{\boldsymbol{\lambda}} = 1$.

Therefore, when applying EG to $\boldsymbol{\lambda}$, we have the optimal step size $\eta_{\boldsymbol{\lambda}} = \sqrt{\frac{4\mu_{\boldsymbol{\lambda}} D_{\boldsymbol{\lambda}}}{5TC_{\boldsymbol{\lambda}} U^2}} = \sqrt{\frac{4\log m}{5TU^2}}$, and the second term in (9) and (44):

$$\sqrt{\frac{20 D_{\boldsymbol{\lambda}} C_{\boldsymbol{\lambda}} U^2}{\mu_{\boldsymbol{\lambda}} T}} = \sqrt{\frac{20 \cdot \log m \cdot 1 \cdot U^2}{1 \cdot T}} = \frac{2\sqrt{5}\sqrt{\log m} U}{\sqrt{T}}.$$

Till now, the proofs for Corollaries 3.4, 3.5, A.7 and A.8 are completed. $\qquad \square$

We also considered the **p-norm algorithm**, another mirror descent instance induced by the *l-p* norm. We analyzed its case and saw that it is no better than PGD or EG, as derived below.

- The *l-p* norm induced $\psi(\mathbf{x}) = \frac{1}{2}\|\mathbf{x}\|_p^2$ is $(p-1)$-strongly convex w.r.t. $\|\cdot\|_p$ for $1 < p \le 2$. Hence, $\mu_{\mathbf{x}} = p - 1$.
- For $B_{\psi_{\mathbf{x}}}(\mathbf{x}^*; \mathbf{x}^{(1)})$, by definition, $B_{\psi_{\mathbf{x}}}(\mathbf{x}^*; \mathbf{x}^{(1)}) = \frac{1}{2}\|\mathbf{x}^*\|_p^2 - \frac{1}{2}\|\mathbf{x}^{(1)}\|_p^2 - \langle \nabla_{\mathbf{x}^{(1)}} \frac{1}{2}\|\mathbf{x}^{(1)}\|_p^2, \mathbf{x}^* - \mathbf{x}^{(1)} \rangle$. To deal with the gradient of the *l-p* norm, consider the gradient w.r.t. the *i*-th element of $\mathbf{x}$, i.e., $x_i$:

$$\nabla_{x_i} \frac{1}{2}\|\mathbf{x}\|_p^2 = \nabla_{x_i} \frac{1}{2} \left( \sum_{j=1}^d |x_j|^p \right)^{2/p} = \left( \sum_{j=1}^d |x_j|^p \right)^{2/p-1} |x_i|^{p-2} x_i. \tag{45}$$

Then, we have $\langle \nabla_{\mathbf{x}} \frac{1}{2}\|\mathbf{x}\|_p^2, \mathbf{x} \rangle = \left( \sum_{j=1}^d |x_j|^p \right)^{2/p-1} \sum_{i=1}^d |x_i|^p = \|\mathbf{x}\|_p^2$, and thus

$$\begin{aligned} B_{\psi_{\mathbf{x}}}(\mathbf{x}^*; \mathbf{x}^{(1)}) &= \frac{1}{2}\|\mathbf{x}^*\|_p^2 - \frac{1}{2}\|\mathbf{x}^{(1)}\|_p^2 - \langle \nabla_{\mathbf{x}^{(1)}} \frac{1}{2}\|\mathbf{x}^{(1)}\|_p^2, \mathbf{x}^* \rangle + \langle \nabla_{\mathbf{x}^{(1)}} \frac{1}{2}\|\mathbf{x}^{(1)}\|_p^2, \mathbf{x}^{(1)} \rangle \\ &= \frac{1}{2}\|\mathbf{x}^*\|_p^2 - \frac{1}{2}\|\mathbf{x}^{(1)}\|_p^2 - \langle \nabla_{\mathbf{x}^{(1)}} \frac{1}{2}\|\mathbf{x}^{(1)}\|_p^2, \mathbf{x}^* \rangle + \|\mathbf{x}^{(1)}\|_p^2 \\ &= \frac{1}{2}\|\mathbf{x}^*\|_p^2 + \frac{1}{2}\|\mathbf{x}^{(1)}\|_p^2 - \langle \nabla_{\mathbf{x}^{(1)}} \frac{1}{2}\|\mathbf{x}^{(1)}\|_p^2, \mathbf{x}^* \rangle. \end{aligned} \tag{46}$$

Next, we upper bound $|\langle \nabla_{\mathbf{x}^{(1)}} \frac{1}{2}\|\mathbf{x}^{(1)}\|_p^2, \mathbf{x}^* \rangle|$. By Lemma A.4, we have $|\langle \nabla_{\mathbf{x}^{(1)}} \frac{1}{2}\|\mathbf{x}^{(1)}\|_p^2, \mathbf{x}^* \rangle| \le \|\nabla_{\mathbf{x}^{(1)}} \frac{1}{2}\|\mathbf{x}^{(1)}\|_p^2\|_q \|\mathbf{x}^*\|_p$, where $\frac{1}{p} + \frac{1}{q} = 1$, i.e., $q + p = pq$. Utilizing Equation (45), we have

$$\begin{aligned} \|\nabla_{\mathbf{x}^{(1)}} \frac{1}{2}\|\mathbf{x}^{(1)}\|_p^2\|_q &= \left( \sum_{j=1}^d |x_j|^p \right)^{2/p-1} \left( \sum_{i=1}^d (|x_i|^{p-1})^q \right)^{1/q} \\ &= \left( \sum_{j=1}^d |x_j|^p \right)^{2/p-1} \left( \sum_{i=1}^d |x_i|^p \right)^{1-1/p} \\ &= \left( \sum_{j=1}^d |x_j|^p \right)^{1/p} = \|\mathbf{x}^{(1)}\|_p. \end{aligned}$$

Therefore, we have $|\langle \nabla_{\mathbf{x}^{(1)}} \frac{1}{2}\|\mathbf{x}^{(1)}\|_p^2, \mathbf{x}^* \rangle| \le \|\mathbf{x}^{(1)}\|_p \|\mathbf{x}^*\|_p \le \max(\|\mathbf{x}^{(1)}\|_p^2, \|\mathbf{x}^*\|_p^2)$.
Finally, continuing from Equation (46), we have

$$B_{\psi_{\mathbf{x}}}(\mathbf{x}^*; \mathbf{x}^{(1)}) \le \frac{1}{2}\|\mathbf{x}^*\|_p^2 + \frac{1}{2}\|\mathbf{x}^{(1)}\|_p^2 + \max(\|\mathbf{x}^*\|_p^2, \|\mathbf{x}^{(1)}\|_p^2) \le 2\max \|\mathbf{x}\|_p^2.$$

For $\boldsymbol{\theta}$, we have $2\max \|\boldsymbol{\theta}\|_p^2 \le 2\left( dR_{\boldsymbol{\theta}}^p \right)^{\frac{2}{p}} = 2d^{\frac{2}{p}} R_{\boldsymbol{\theta}}^2$. For $\boldsymbol{\lambda}$, we have $2\max \|\boldsymbol{\lambda}\|_p^2 \le 2\max \|\boldsymbol{\lambda}\|_1^2 \le 2$. Hence, $D_{\boldsymbol{\theta}} = 2d^{\frac{2}{p}} R_{\boldsymbol{\theta}}^2$ and $D_{\boldsymbol{\lambda}} = 2$.

- For $C_{\mathbf{x}}$, since the dual norm of $\|\cdot\|_p$ is $\|\cdot\|_q$ s.t. $\frac{1}{p} + \frac{1}{q} = 1$, we have for any gradient $\nabla_{\mathbf{x}}$,

$$\|\nabla_{\mathbf{x}}\|_{\mathbf{x},*}^2 = \|\nabla_{\mathbf{x}}\|_q^2 \le (\dim(\mathbf{x})\|\nabla_{\mathbf{x}}\|_\infty^q)^{\frac{2}{q}} \le \dim(\mathbf{x})^{\frac{2}{q}} \|\nabla_{\mathbf{x}}\|_\infty^2.$$

Hence, $C_{\boldsymbol{\theta}} = d^{\frac{2}{q}}$ and $C_{\boldsymbol{\lambda}} = m^{\frac{2}{q}}$.

Therefore, using p-norm, the bounds for $\boldsymbol{\theta}$ and $\boldsymbol{\lambda}$ respectively are:

$$\frac{2\sqrt{10}dR_{\boldsymbol{\theta}}L}{(p-1)\sqrt{T}} \quad \text{and} \quad \frac{2\sqrt{10}m^{\frac{1}{q}}U}{\sqrt{(p-1)T}}.$$

The bound for $\boldsymbol{\theta}$ achieves the minimum value when $p = 2$ (recall that $1 < p \le 2$), in which case p-norm reduces to Projected Gradient Descent. For $\boldsymbol{\lambda}$, we can further upper bound the term by considering:

$$\frac{m^{\frac{1}{q}}}{\sqrt{p-1}} = m^{\frac{1}{q}}\sqrt{q-1} \le m^{\frac{1}{q}}\sqrt{q},$$

whose minimum value is $\sqrt{2e\log m}$, obtained when $q = 2\log m$. Substituting back, the optimal bound for $\boldsymbol{\lambda}$ is $\frac{4\sqrt{5e\log m}U}{\sqrt{T}}$, which is no better than Exponentiated Gradient.

### A.3 Comparison with previous results and discussion on sample complexity

In previous works adopting OMD-based methods with EG for $\boldsymbol{\lambda}$ (Sagawa et al., 2020; He et al., 2024), the convergence rate is derived as $\mathcal{O}(m\sqrt{\log m/T})$. The additional $m$ factor compared to our bound in Corollary 3.5 is induced by their sampling *one* data point in each round. Recall that given the composite loss $\mathcal{L}(\boldsymbol{\theta}, \boldsymbol{\lambda}) = \sum_{i=1}^{m} \lambda_i f_i(\boldsymbol{\theta})$ (the preference weight $\mathbf{w}$ is omitted for simplicity), we want to build unbiased estimators for $\nabla_{\boldsymbol{\lambda}}\mathcal{L}(\boldsymbol{\theta}, \boldsymbol{\lambda}) = (f_1(\boldsymbol{\theta}), \dots, f_m(\boldsymbol{\theta}))^T$. The sampling strategy in these works is: (1) sample an objective index $i \sim [m]$ uniformly, (2) sample one data point $x$ from the dataset of objective $i$, i.e., $x \sim \mathcal{D}_i$, and (3) construct the stochastic gradient $\delta_{\boldsymbol{\lambda}}\mathcal{L}(\boldsymbol{\theta}, \boldsymbol{\lambda})$:

$$(\delta_{\boldsymbol{\lambda}})_i = mf_i(\boldsymbol{\theta}, x) \quad \text{and} \quad (\delta_{\boldsymbol{\lambda}})_j = 0, \quad \forall j \ne i$$

Here, we use $\delta_{\boldsymbol{\lambda}}$ to denote $\delta_{\boldsymbol{\lambda}}\mathcal{L}(\boldsymbol{\theta}, \boldsymbol{\lambda})$ for simplicity. The constant $m$ in the non-zero entry is necessary for this stochastic gradient to be unbiased: for each entry $j$ in $\delta_{\boldsymbol{\lambda}}$,

$$\mathbb{E}\left[(\delta_{\boldsymbol{\lambda}})_j\right] = \frac{m-1}{m} \times 0 + \frac{1}{m} \times m\mathbb{E}_{x \sim D_j}\left[f_j(\boldsymbol{\theta}, x)\right]$$
$$= f_j(\boldsymbol{\theta}) = (\nabla_{\boldsymbol{\lambda}}\mathcal{L}(\boldsymbol{\theta}, \boldsymbol{\lambda}))_j.$$

Due to the constant $m$, the upper bound of $\|\delta_{\boldsymbol{\lambda}}\|_{\infty}$ are amplified by a factor of $m$. This is equivalent to having $U \to mU$ in Corollaries 3.5 and A.8, and hence the additional $m$ in these previous bounds. In contrast, our strategy samples from all objectives at each step and constructs $(\delta_{\boldsymbol{\lambda}})_i = f_i(\boldsymbol{\theta}; B_i)$, where $B_i$ is the random batch sampled from the data of objective $i$. This stochastic gradient is unbiased without a coefficient of $m$, and thus the optimality.

Full sampling is adopted by most MOL methods, as we usually perform offline learning with a pre-collected training set, and many gradient manipulation methods require full gradient information for conflict resolving. Therefore, our analysis provides a more suitable convergence rate for OMD-based methods for MOL.

Finally, while full sampling yields a tighter iteration complexity in the offline setting, we point out that in online/active/continuous learning, where one desires lower sample complexity, there is no optimal strategy between the two sampling methods. Instead, they offer different trade-offs in the sample complexity bound:

$$\text{Single-objective } \mathcal{O}(1) \text{ sampling: } \mathcal{O}\left(\frac{R_{\boldsymbol{\theta}}^2 + m^2\log m + (R_{\boldsymbol{\theta}}^2 + m^2)\log\frac{1}{\gamma}}{\epsilon^2}\right),$$

$$\text{Full } \mathcal{O}(m) \text{ sampling: } \mathcal{O}\left(\frac{mR_{\boldsymbol{\theta}}^2 + m\log m + (mR_{\boldsymbol{\theta}}^2 + m)\log\frac{1}{\gamma}}{\epsilon^2}\right),$$

where $\epsilon$ is the optimization error and constants $d$, $L$, and $U$ are omitted. These sampling complexity bounds can be easily derived from the iteration complexity bounds and the number of samples drawn in each round. As shown, $\mathcal{O}(1)$ sampling avoids multiplication between $m$ and $R_{\boldsymbol{\theta}}$, while $\mathcal{O}(m)$ sampling avoids the quadratic dependency in $m$.

# B    Full experiment details and results

Our code is available at `https://github.com/uiuctml/AdaOMD-TCH`.

## B.1    Synthetic experiments

Experiments are conducted using an open-source MOL library (Zhang et al., 2024) with implemented synthetic problems and some methods. Uncovered methods are added following the corresponding official codes.

**Hyperparameters and random seeds.** All experiments run for 1000 epochs. Those of linear scalarization (LS), vanilla Tchebycheff scalarization (TCH), PMTL, EPO, and FERERO use $\eta_\theta = 0.01$. All experiments of (Ada)OMD-TCH, STCH, ExcessMTL, MGDA, CR-MOGM, and Moco use $\eta_\theta = 0.02$. This is to ensure the same scale for updates, since there is an additional coefficient for the gradient term of $\theta$ from the dynamic weights, which is roughly of scale 0.5 given that the number of objectives is 2. Nevertheless, we did not observe significant differences when using either rate on the same method.

In terms of method-specific hyperparameters, all experiments of (Ada)OMD-TCH, either using PGD or EG for $\lambda$, as well as ExcessMTL, use $\eta_\lambda = 1.0$. For STCH, we use $\mu = 0.01$. The momentum parameter $\alpha_t$ in CR-MOGM is computed as suggested in Corollary 1 of the original work. For Moco, the regularization constant $\rho$ is set as 0.01 and the step size for the dynamic weight, similar to $\eta_\lambda$, is set as 1.0. All results are averages over three random seeds, 0, 19, and 42.

**Full results.** Figures for all methods on all synthetic problems are presented in Figures 6 to 8. (Ada)OMDgd-TCH stands for applying PGD to $\lambda$, and "eg" for EG. Similar results as discussed in the main body are observed across different problems. Additionally, compared to the PGD instance, the EG ones are sometimes less effective in recovering the exact PO solutions found by TCH. We also report hypervolume results in Table 10, where TCH achieves the best hypervolume, and AdaOMDgd-TCH ranks the second on five of the seven problems, complementing our findings in the figures.

**(Ada)OMD-TCH as a middle ground between LS and TCH.** To provide an intuitive illustration for this interpretation in Section 3.3, we apply OMDgd-TCH on VLMOP2 with different $\eta_\lambda$. As shown in Figure 5, when $\eta_\lambda$ reduces to near 0, the results resemble LS, and increasing it recovers the results of TCH.

## B.2    Experiments on fair federated learning

**Data settings.** The detailed approach to simulate client data heterogeneity is as follows:

- Rotation: Both datasets are equally and randomly split into 10 subsets for 10 clients. For MNIST, 7 subsets are not rotated, 2 subsets are rotated by 90 degrees, and 1 subset is rotated by 180 degrees. For CIFAR10, 7 subsets are unchanged, and 3 subsets are rotated by 180 degrees. For the 2-client preference-guided experiment, data are equally and randomly split into two subsets, each rotated by 0 and 90 degrees.

- Partial Class with $C = 2$ or 5: Both datasets are first equally and randomly split into 10 subsets. For each client, $C$ classes are randomly chosen, and the images in the corresponding subset of these classes are assigned to the client.

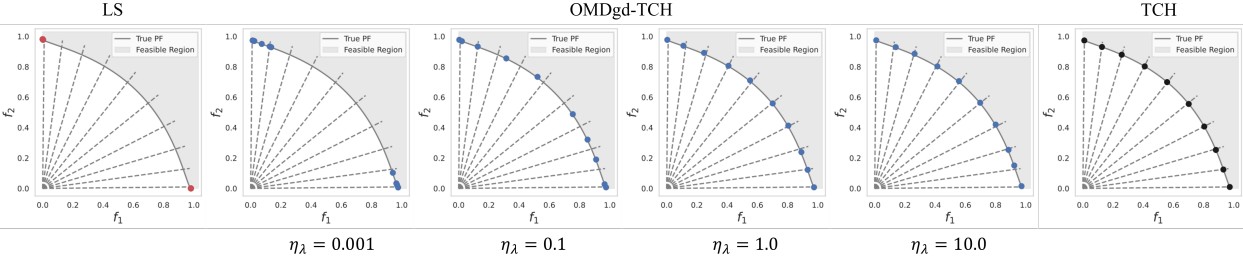

Figure 5: **An example of (Ada)OMD-TCH lying between LS and TCH.**

Table 5: **Method-specific hyperparameter values.**

| Method | Hyperparam | Attempted Values | Best Value | | | | | |
| --- | --- | --- | --- | --- | --- | --- | --- | --- |
| | | | MNIST Rotation | MNIST Partial Class C=2 | MNIST Partial Class C=5 | CIFAR10 Rotation | CIFAR10 Partial Class C=2 | CIFAR10 Partial Class C=5 |
| qFFL | $q$ | 0.1, 0.5, 1.0, 5.0, 10.0 | 0.1 | 0.1 | 0.1 | 0.1 | 0.1 | 0.1 |
| PropFair | $M$ | 2.0, 3.0, 4.0, 5.0 | 2.0 | 2.0 | 2.0 | 3.0 | 4.0 | 2.0 |
| FedMGDA+ | $\epsilon$ | 0.05, 0.1, 0.5, 1.0 | 0.5 | 0.5 | 0.05 | 0.1 | 0.1 | 0.1 |
| FedFV | $\alpha$ | 0.1, 0.2, 0.5 | 0.2 | 0.2 | 0.2 | 0.2 | 0.1 | 0.2 |
| | $\tau$ | 0, 1, 3, 10 | 1 | 1 | 1 | 1 | 0 | 1 |
| STCH(TERM) | $\mu$ | 0.01, 0.03, 0.1, 0.3, 1.0, 3.0, 10.0 | 0.01 | 0.01 | 0.01 | 0.01 | 0.03 | 0.03 |
| ExcessMTL | $0.1\eta_\lambda$ | | 1.0 | 1.0 | 1.0 | 1.0 | 0.1 | 0.1 |
| AdaExcessMTL | $0.1\eta_\lambda$ | | 0.3 | 1.0 | 1.0 | 0.1 | 0.03 | 0.001 |
| OMDgd-TCH(AFL) | $0.1\eta_\lambda$ | 0.001, 0.003, 0.01, 0.03, 0.1, 0.3, 1.0 | 0.03 | 0.01 | 0.01 | 1.0 | 0.01 | 0.01 |
| AdaOMDgd-TCH | $0.1\eta_\lambda$ | | 0.03 | 1.0 | 1.0 | 0.03 | 0.003 | 0.3 |
| OMDeg-TCH | $0.1\eta_\lambda$ | | 0.3 | 0.3 | 0.1 | 1.0 | 0.03 | 0.1 |
| AdaOMDeg-TCH | $0.1\eta_\lambda$ | | 0.1 | 0.3 | 0.03 | 0.3 | 0.03 | 0.03 |

**Hyperparameters and random seeds.** For all MNIST experiments, the hidden layer of the fully connected neural network is of size $h = 200$. We go through $T = 300$ communication rounds, with each client executing $\tau = 10$ epochs of full-batch local updates in each round. For all CIFAR10 experiments, we train the ResNet18 model for $T = 45$ communication rounds, with each client doing $\tau = 1$ epoch of local update with a batch size of 100 in each round.

All methods' step sizes for model parameters are set as $\eta_{\boldsymbol{\theta}} = 0.1$. For other method-specific hyperparameters, we try values reported in their original paper and select the best ones that yield the smallest worst client training loss, in order to match our methods for a fair comparison. The attempted values and the best ones are listed in Table 5. Here, $\eta_{\boldsymbol{\lambda}}$ for (Ada)OMD-TCH is reported as $0.1\eta_{\boldsymbol{\lambda}}$, which is what we used in practical codes. This is because under a balanced tradeoff, $w_i = 1/m = 0.1$ for all objectives/clients and can be integrated into $\eta_{\boldsymbol{\lambda}}$. For the 2-client preference-guided experiment, we directly use the best hyperparameter of the corresponding method in the CIFAR10 Rotation setting.

All evaluation results are averaged over 10 random seeds: 0, 25, 37, 42, 53, 81, 119, 1010, 1201, 2003.

**Machines and training time.** We use a Linux machine with two AMD EPYC 7352 24-Core Processors and eight NVIDIA RTX A6000 GPUs. Table 6 reports the training time per FL round of different methods averaged over 10 seeds. Comparing AdaOMD-TCH and OMD-TCH, AdaExcessMTL and ExcessMTL, we see that the adaptive conversion does not impose any significant overhead. As discussed in Section 3.2, this is mainly because of the small size of $\mathcal{P}^{(t)}$ in practice. Additionally, since OMD-TCH requires strictly no more computation than ExcessMTL, the discrepancy between them is likely due to scheduling deviation of the servers. Therefore, most methods with small differences can be considered similar in training time.

Table 6: **Training time per round.**

| Method | MNIST | | | CIFAR10 | | |
| --- | --- | --- | --- | --- | --- | --- |
| | Rotation | Partial Class C=2 | Partial Class C=5 | Rotation | Partial Class C=2 | Partial Class C=5 |
| LS(FedAvg) | 0.338 | 0.206 | 0.236 | 9.748 | 2.142 | 6.168 |
| qFFL | 0.591 | 0.181 | 0.482 | 7.823 | 2.123 | 5.450 |
| PropFair | 0.733 | 0.559 | 0.620 | 8.033 | 2.059 | 5.247 |
| FedMGDA+ | 1.360 | 0.535 | 0.276 | 7.920 | 2.118 | 3.272 |
| FedFV | 0.341 | 0.233 | 0.607 | 7.741 | 2.438 | 5.937 |
| TCH | 0.329 | 0.227 | 0.266 | 8.191 | 1.480 | 3.170 |
| STCH(TERM) | 0.590 | 0.546 | 0.441 | 7.939 | 2.537 | 5.118 |
| EPO | 0.377 | 0.138 | 0.180 | 8.656 | 1.738 | 4.241 |
| FERERO | 0.462 | 0.227 | 0.263 | 8.736 | 2.000 | 4.492 |
| ExcessMTL | 0.344 | 0.122 | 0.197 | 8.576 | 1.562 | 4.327 |
| AdaExcessMTL | 0.442 | 0.142 | 0.175 | 7.881 | 1.389 | 4.239 |
| OMDgd-TCH(AFL) | 0.365 | 0.206 | 0.266 | 8.042 | 2.420 | 5.273 |
| AdaOMDgd-TCH | 0.591 | 0.226 | 0.278 | 7.900 | 3.022 | 5.527 |
| OMDeg-TCH | 0.348 | 0.209 | 0.270 | 8.313 | 2.342 | 6.017 |
| AdaOMDeg-TCH | 0.564 | 0.460 | 0.569 | 7.995 | 2.431 | 5.508 |

Table 7: **Comparison between LS, AdaLS, TCH, AdaTCH, and AdaOMDgd-TCH.**

(a) Average accuracy ↑

| Method | LS | AdaLS | TCH | AdaTCH | AdaOMD-TCH |
|---|---|---|---|---|---|
| Rotation | 66.269 | *66.169* ↓ | 62.556 | *65.257* ↑ | 66.218 |
| Partial Class $C = 2$ | 38.925 | *38.095* ↓ | 20.935 | *21.150* ↑ | 36.765 |
| Partial Class $C = 5$ | 55.626 | *55.540* ↓ | 36.242 | *47.702* ↑ | 53.422 |

(b) Accuracy parity ↓

| Method | LS | AdaLS | TCH | AdaTCH | AdaOMD-TCH |
|---|---|---|---|---|---|
| Rotation | 6.012 | *6.283* ↑ | 4.727 | *1.920* ↓ | 3.592 |
| Partial Class $C = 2$ | 21.801 | *22.658* ↑ | 27.199 | *16.345* ↓ | 18.422 |
| Partial Class $C = 5$ | 8.610 | *8.565* ↓ | 16.141 | *7.204* ↓ | 6.823 |

**Full results.** Figure 9 includes the training and test curves of the maximum client loss for LS, TCH, and (Ada)OMD-TCH, complementing Figure 3 in the main text. Figure 10 includes the scatter plots of average test accuracy and test accuracy parity, also complementing Figure 3. Figure 11 visualizes the benchmark across all methods, and Table 11 lists the corresponding numerical results.

**Design components.** The motivations for the three components of AdaOMD-TCH are as follows:

- *Tchebycheff scalarization* ensures (1) recovering the non-convex PF and (2) locating solutions with specific trade-offs (Theorem 2.4), which are important for preference-guided MOL and many methods fail to satisfy, such as LS and gradient manipulation methods without additional design.

- *OMD updates* smooth out the one-hot update of TCH to avoid training oscillation and stagnation.

- *The adaptive conversion* considers only the trajectory PO iterates to improve solution optimality over traditional conversion while retaining the theoretical convergence guarantees.

Although the adaptive conversion is designed for online algorithms such as OMD and thus does not apply, in principle, to LS and TCH, we still investigate the performance of AdaLS and AdaTCH to further justify the use of Tchebycheff scalarization and OMD updates.

Table 7 presents the comparison on CIFAR over 10 seeds. In terms of accuracy, the adaptive conversion does not help for LS. AdaTCH improves over TCH, but is still outperformed by AdaOMD-TCH by a large amount, not addressing the unstable and slow one-hot update as effectively as OMD. Nevertheless, this improvement indicates that our proposed adaptive scheme is more broadly applicable.

In terms of fairness, AdaLS is hardly improved and is still outperformed by most TCH-based methods. This is because the adaptive conversion does not change the property of LS, i.e., LS does not aim for equal (weighted) losses like TCH. The fairness of AdaTCH is improved, but its key drawback is still the suboptimal accuracy.

In summary, AdaTCH < AdaOMD-TCH in average accuracy demonstrates the necessity of applying OMD updates, and AdaLS > most TCH methods in accuracy parity further confirms the advantage of Tchebycheff scalarization. Applying the adaptive conversion to a broader range of methods, such as TCH, may be an interesting direction, which we leave for future studies.

**Evaluation data for iterate comparison.** To demonstrate the robustness of the adaptive conversion against different evaluation data, we conduct an ablation on the CIFAR rotation setting, where the objective values, i.e., client losses, for dominance comparison use:

- mini-batch estimates (mini): This corresponds to the current results in Table 11d.

- full training loss (full): Each client is evaluated on its full local training data.

- held-out validation estimates (val): Since CIFAR10 does not feature a validation set, we split the test set in half, using one for validation and the other for testing.

Table 8: **Ablation on different evaluation data for iterate comparison.**

| Method | AdaOMDgd-TCH | | | AdaOMDeg-TCH | | |
|---|---|---|---|---|---|---|
| | mini | full | val | mini | full | val |
| Avg. Accuracy ↑ | 66.218 | 66.061 | 66.386 | 65.885 | 65.913 | 66.260 |
| Agnostic Loss ↓ | 1.148 | 1.128 | 1.185 | 1.156 | 1.130 | 1.171 |
| Accuracy Parity ↓ | 3.592 | 3.503 | 4.022 | 3.688 | 3.372 | 3.819 |

Table 9: **Compatibility of (Ada)OMD-TCH with using objective ratios for weight updates.**

| Method | LS | TCH | STCH | OMDgd-TCH | AdaOMDgd-TCH | OMDeg-TCH | AdaOMDeg-TCH |
|---|---|---|---|---|---|---|---|
| Agnostic Ratio ↓ | 0.656 | 0.648 | 0.627 | 0.647 | **0.465** | 0.651 | 0.465 |
| Ratio Parity ↓ | 0.087 | 0.066 | 0.068 | **0.024** | 0.043 | 0.029 | 0.042 |

We directly use the reported hyperparameters and seeds. As shown in Table 8, comparing the full and mini columns, one sees that using full training loss indeed improves agnostic loss, which is the goal of Tchebycheff scalarization, and accuracy parity. Using the validation set slightly worsens the two but increases the average accuracy. Most importantly, switching to either evaluation barely changes the relative performance of AdaOMD-TCH compared to other methods, e.g., they still achieve the best agnostic loss. This validates that the adaptive conversion is robust against different dominance comparison data.

**Objectives of different scales.** One practical trick to resolve the bias induced by scale differences is to, when updating the dynamic weights, replace $f_i(\theta^{(t)})$ with the ratio $f_i(\theta^{(t)})/f_i(\theta^{(0)}) \in (0, 1]$, where $f_i(\theta^{(0)})$ is the initial value of the $i$-th objective. Thereby, the weight allocation process examines not the raw objectives, but how much each has improved, aiming for a preference-specified degree of training across objectives. This trick has been adopted by previous works to normalize excess risks (He et al., 2024; Chen et al., 2018).

To demonstrate the compatibility of our methods with this trick, we apply it to TCH, STCH, and (Ada)OMD-TCH on the CIFAR rotation setting. By using the ratios, we evaluate the agnostic ratio, i.e., the worst test loss ratio across clients, and the ratio parity, i.e., the standard deviation of test loss ratios. We use the current hyperparameters and seeds. As shown in Table 9, the results are consistent with our current findings. Specifically, we still have (1) TCH is better than LS, (2) OMD-TCH is often better than TCH, and (3) AdaOMD-TCH greatly improves optimality over OMD-TCH, as reflected by the smaller agnostic ratio, i.e., the performance of the least-trained objective, with a slight increase in ratio parity, which is still better than other methods. This demonstrates the compatibility of our methods with the ratio trick and, thus, potentially with different scales of objectives.

Table 10: **Hypervolumes of different methods on synthetic problems.**

| Problem | VLMOP2 | F1 | F2 | F3 | F4 | F5 | F6 |
|---|---|---|---|---|---|---|---|
| LS | 0.042±0.001 | 0.938±0.065 | 1.005±0.005 | 0.967±0.018 | 1.002±0.013 | 1.004±0.019 | 0.989±0.038 |
| TCH | **0.295±0.000** | **1.011±0.009** | **1.021±0.009** | **1.015±0.003** | **1.024±0.016** | **1.027±0.003** | **1.023±0.006** |
| STCH | 0.291±0.000 | 0.986±0.007 | 0.995±0.007 | 0.966±0.004 | 1.004±0.014 | 1.003±0.007 | 1.014±0.007 |
| OMDgd-TCH | 0.289±0.001 | 0.970±0.007 | 0.990±0.003 | 0.950±0.003 | 0.973±0.003 | 0.983±0.010 | 1.003±0.004 |
| AdaOMDgd-TCH | 0.292±0.000 | 0.992±0.005 | 1.008±0.004 | 0.970±0.003 | 1.006±0.013 | 1.014±0.010 | 1.015±0.004 |
| OMDeg-TCH | 0.269±0.006 | 0.960±0.010 | 0.992±0.004 | 0.926±0.024 | 0.942±0.013 | 0.962±0.016 | 0.977±0.018 |
| AdaOMDeg-TCH | 0.270±0.006 | 0.981±0.009 | 1.006±0.004 | 0.944±0.021 | 0.979±0.015 | 0.984±0.026 | 0.990±0.014 |
| ExcessMTL | 0.119±0.011 | 0.931±0.022 | 0.965±0.026 | 0.986±0.012 | 0.994±0.005 | 0.993±0.018 | 0.989±0.016 |
| EPO | 0.283±0.003 | 0.968±0.003 | 0.953±0.007 | 0.966±0.003 | 0.957±0.014 | 0.965±0.024 | 0.952±0.020 |
| FERERO | 0.289±0.000 | 1.010±0.008 | 1.001±0.001 | 0.993±0.005 | 0.996±0.009 | 1.002±0.014 | 0.995±0.009 |
| MGDA | 0.232±0.006 | 0.963±0.015 | 0.970±0.015 | 0.981±0.013 | 0.970±0.014 | 0.956±0.018 | 0.994±0.010 |
| CR-MOGM | 0.231±0.006 | 0.963±0.015 | 0.971±0.014 | 0.982±0.013 | 0.970±0.014 | 0.958±0.017 | 0.993±0.010 |
| Moco | 0.255±0.011 | 0.956±0.023 | 0.977±0.014 | 0.979±0.021 | 0.992±0.034 | 0.981±0.024 | 0.989±0.035 |
| PMTL | 0.208±0.028 | 0.883±0.021 | 0.892±0.022 | 0.884±0.004 | 0.887±0.019 | 0.880±0.043 | 0.884±0.011 |

Table 11: **Numerical results of all methods.**

(a) MNIST Rotation

| Method | LS(FedAvg) | qFFL | PropFair | FedMGDA+ | FedFV | TCH | STCH | EPO | FERERO | ExcessMTL | AdaExcessMTL | OMDgd-TCH(AFL) | AdaOMDgd-TCH | OMDeg-TCH | AdaOMDeg-TCH |
|---|---|---|---|---|---|---|---|---|---|---|---|---|---|---|---|
| Avg. Accuracy | 92.450±0.234 | 91.896±0.226 | 90.622±0.194 | 92.416±0.210 | **94.501±0.179** | 92.579±0.230 | 92.977±0.230 | 92.604±0.302 | 92.443±0.232 | 88.566±0.279 | 92.520±0.235 | 88.597±0.241 | 92.593±0.236 | 88.664±0.220 | 92.583±0.185 |
| Agnostic Loss | 0.639±0.028 | 0.675±0.027 | 0.742±0.025 | 0.322±0.013 | **0.302±0.027** | 0.342±0.027 | 0.409±0.027 | 0.350±0.029 | 0.640±0.028 | 0.503±0.013 | 0.326±0.013 | 0.514±0.024 | 0.341±0.019 | 0.518±0.026 | 0.334±0.019 |
| Accuracy Parity | 4.796±0.272 | 5.021±0.262 | 5.454±0.152 | 1.153±0.168 | 1.646±0.335 | 1.397±0.292 | 2.385±0.247 | 1.467±0.288 | 4.807±0.271 | 1.637±0.106 | **1.117±0.148** | 1.658±0.257 | 1.296±0.211 | 1.680±0.238 | 1.201±0.227 |

(b) MNIST Partial Class C=2

| Method | LS(FedAvg) | qFFL | PropFair | FedMGDA+ | FedFV | TCH | STCH | EPO | FERERO | ExcessMTL | AdaExcessMTL | OMDgd-TCH(AFL) | AdaOMDgd-TCH | OMDeg-TCH | AdaOMDeg-TCH |
|---|---|---|---|---|---|---|---|---|---|---|---|---|---|---|---|
| Avg. Accuracy | 92.330±1.429 | 90.264±1.714 | 91.274±1.505 | 92.236±1.405 | **93.675±1.331** | 93.250±1.468 | 92.592±1.558 | 91.866±2.986 | 92.301±1.467 | 91.354±1.368 | 92.435±1.514 | 90.965±1.502 | 92.387±1.616 | 90.133±1.850 | 92.209±1.580 |
| Agnostic Loss | 0.534±0.106 | 0.659±0.118 | 0.573±0.105 | **0.386±0.055** | 0.397±0.094 | 0.409±0.067 | 0.477±0.098 | 0.517±0.152 | 0.545±0.111 | 0.489±0.047 | 0.404±0.066 | 0.549±0.092 | 0.424±0.099 | 0.569±0.111 | 0.435±0.105 |
| Accuracy Parity | 4.574±1.469 | 6.025±1.696 | 4.987±1.620 | **2.629±0.489** | 3.318±0.868 | 3.685±0.982 | 3.989±1.201 | 4.273±1.772 | 4.700±1.500 | 3.530±0.673 | 2.753±0.523 | 4.373±1.027 | 3.153±0.817 | 4.563±1.095 | 3.084±0.865 |

(c) MNIST Partial Class C=5

| Method | LS(FedAvg) | qFFL | PropFair | FedMGDA+ | FedFV | TCH | STCH | EPO | FERERO | ExcessMTL | AdaExcessMTL | OMDgd-TCH(AFL) | AdaOMDgd-TCH | OMDeg-TCH | AdaOMDeg-TCH |
|---|---|---|---|---|---|---|---|---|---|---|---|---|---|---|---|
| Avg. Accuracy | 93.907±0.418 | 93.008±0.514 | 92.865±0.462 | 93.944±0.480 | **95.017±0.515** | 94.690±0.317 | 94.140±0.405 | 93.402±0.942 | 93.891±0.421 | 92.424±0.460 | 94.063±0.444 | 92.256±0.501 | 94.239±0.440 | 92.261±0.505 | 94.021±0.432 |
| Agnostic Loss | 0.282±0.034 | 0.320±0.034 | 0.328±0.030 | 0.267±0.027 | **0.231±0.032** | 0.250±0.045 | 0.272±0.032 | 0.304±0.047 | 0.283±0.034 | 0.342±0.022 | 0.264±0.031 | 0.349±0.025 | 0.263±0.033 | 0.352±0.028 | 0.275±0.032 |
| Accuracy Parity | 1.559±0.378 | 1.752±0.431 | 1.727±0.424 | **1.175±0.244** | 1.287±0.306 | 1.518±0.375 | 1.360±0.335 | 1.910±0.484 | 1.564±0.403 | 1.578±0.443 | 1.205±0.268 | 1.733±0.445 | 1.193±0.280 | 1.799±0.482 | 1.351±0.371 |

(d) CIFAR10 Rotation

| Method | LS(FedAvg) | qFFL | PropFair | FedMGDA+ | FedFV | TCH | STCH | EPO | FERERO | ExcessMTL | AdaExcessMTL | OMDgd-TCH(AFL) | AdaOMDgd-TCH | OMDeg-TCH | AdaOMDeg-TCH |
|---|---|---|---|---|---|---|---|---|---|---|---|---|---|---|---|
| Avg. Accuracy | 66.269±0.541 | 61.514±0.450 | 63.108±0.705 | 66.321±0.264 | 66.242±0.475 | 66.320±0.403 |  | 61.380±2.680 | **66.368±0.355** | 59.322±0.520 | 66.211±0.412 | 59.244±0.489 | 66.218±0.467 | 59.273±0.443 | 65.885±0.610 |
| Agnostic Loss | 1.260±0.044 | 1.314±0.041 | 1.286±0.044 | 1.197±0.035 | 1.201±0.038 | 1.266±0.066 | 1.219±0.043 | 1.293±0.062 | 1.259±0.040 | 1.256±0.028 | 1.227±0.041 | 1.239±0.029 | **1.148±0.037** | 1.254±0.033 | 1.156±0.045 |
| Accuracy Parity | 6.012±0.596 | 5.451±0.514 | 5.312±0.385 | 4.632±0.430 | 4.668±0.581 | 4.727±1.099 | 4.997±0.712 | 4.731±0.928 | 5.922±0.665 | 2.795±0.332 | 5.615±0.745 | **2.429±0.531** | 3.592±0.487 | 2.660±0.549 | 3.688±0.646 |

(e) CIFAR10 Partial Class C=2

| Method | LS(FedAvg) | qFFL | PropFair | FedMGDA+ | FedFV | TCH | STCH | EPO | FERERO | ExcessMTL | AdaExcessMTL | OMDgd-TCH(AFL) | AdaOMDgd-TCH | OMDeg-TCH | AdaOMDeg-TCH |
|---|---|---|---|---|---|---|---|---|---|---|---|---|---|---|---|
| Avg. Accuracy | 38.925±2.325 | 34.995±2.658 | 35.505±2.107 | 38.560±2.974 | **39.015±3.170** | 20.935±6.355 | 35.380±3.949 | 20.510±7.571 | 38.700±3.236 | 35.240±2.402 | 38.330±2.893 | 31.080±4.816 | 36.765±3.475 | 34.865±2.422 | 36.755±3.373 |
| Agnostic Loss | 2.477±0.256 | 2.584±0.255 | 2.486±0.215 | **2.148±0.089** | 2.394±0.197 | 4.378±0.387 | 2.386±0.228 | 4.364±0.641 | 2.468±0.260 | 2.526±0.294 | 2.434±0.277 | 2.515±0.269 | 2.372±0.239 | 2.534±0.244 | 2.388±0.259 |
| Accuracy Parity | 21.801±5.071 | 23.085±5.014 | 22.642±4.907 | **14.227±2.428** |  | 19.289±4.173 | 19.139±6.063 | 24.116±4.796 | 21.922±4.676 | 22.256±5.647 | 21.518±5.055 | 20.600±6.157 | 18.422±5.435 | 22.045±5.490 | 18.933±5.478 |

(f) CIFAR10 Partial Class C=5

| Method | LS(FedAvg) | qFFL | PropFair | FedMGDA+ | FedFV | TCH | STCH | EPO | FERERO | ExcessMTL | AdaExcessMTL | OMDgd-TCH(AFL) | AdaOMDgd-TCH | OMDeg-TCH | AdaOMDeg-TCH |
|---|---|---|---|---|---|---|---|---|---|---|---|---|---|---|---|
| Avg. Accuracy | 55.626±1.606 | 49.316±1.859 | 53.794±1.530 | **56.174±1.339** | 55.746±1.448 | 36.242±4.324 | 55.328±1.656 | 37.458±2.333 | 55.662±1.277 | 48.142±1.862 | 55.766±1.753 | 47.960±1.874 | 53.422±1.402 | 48.064±1.952 | 55.632±1.524 |
| Agnostic Loss | 1.705±0.125 | 1.856±0.147 | 1.740±0.137 | **1.596±0.143** | 1.662±0.098 | 3.748±0.279 | 1.622±0.048 | 3.138±0.421 | 1.729±0.157 | 1.870±0.136 | 1.705±0.170 | 1.832±0.080 | 1.664±0.065 | 1.833±0.089 | 1.662±0.074 |
| Accuracy Parity | 8.610±1.751 | 9.614±2.169 | 9.037±1.697 | 6.862±1.842 | 7.609±1.284 | 16.141±2.355 | 7.311±1.253 | 14.602±1.629 | 8.854±2.064 | 9.715±2.075 | 8.354±2.072 | 8.787±1.413 | **6.823±1.504** | 8.803±1.133 | 7.930±1.185 |

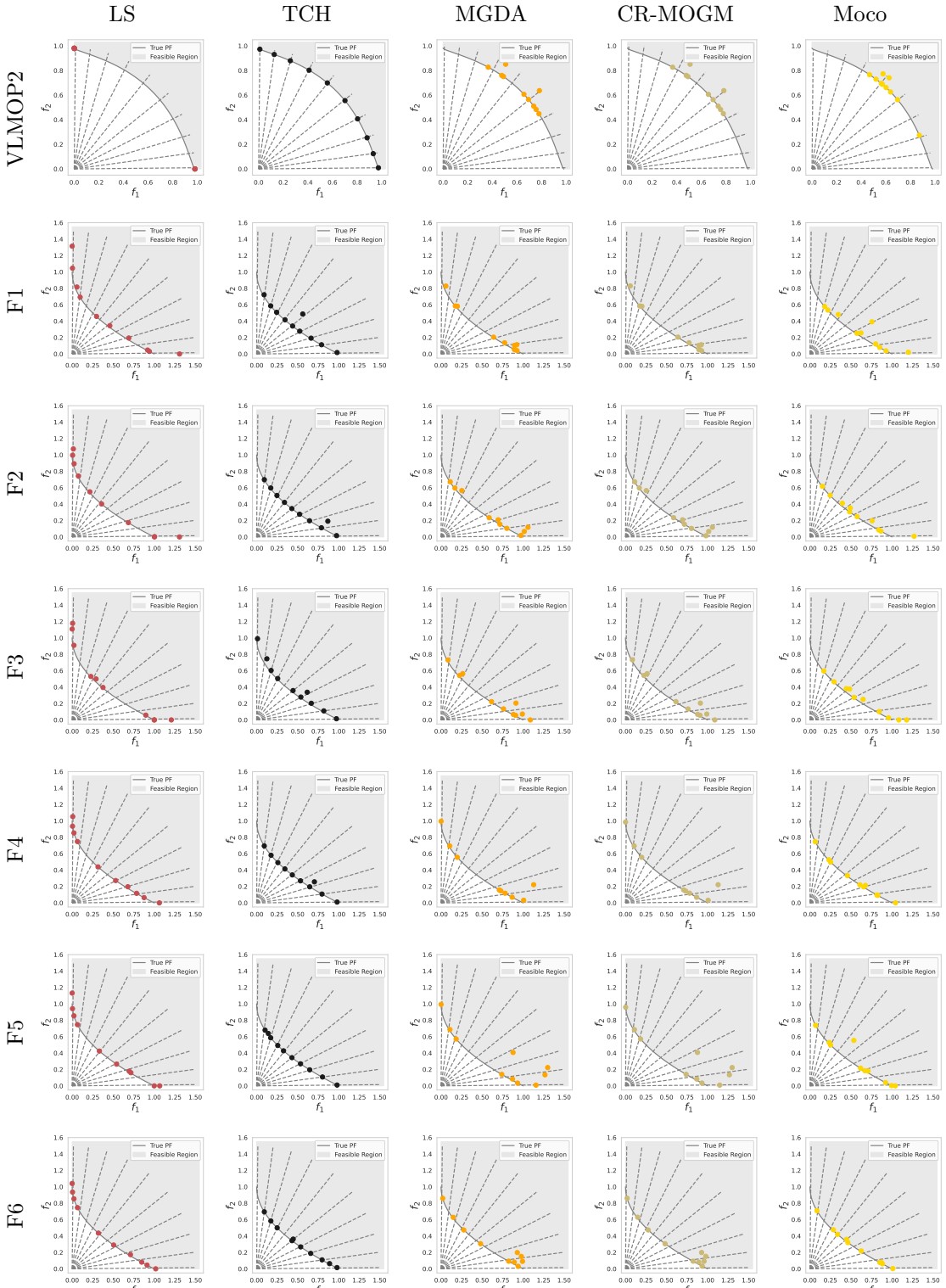

Figure 6: **Full results of LS, TCH, and gradient manipulation methods.**

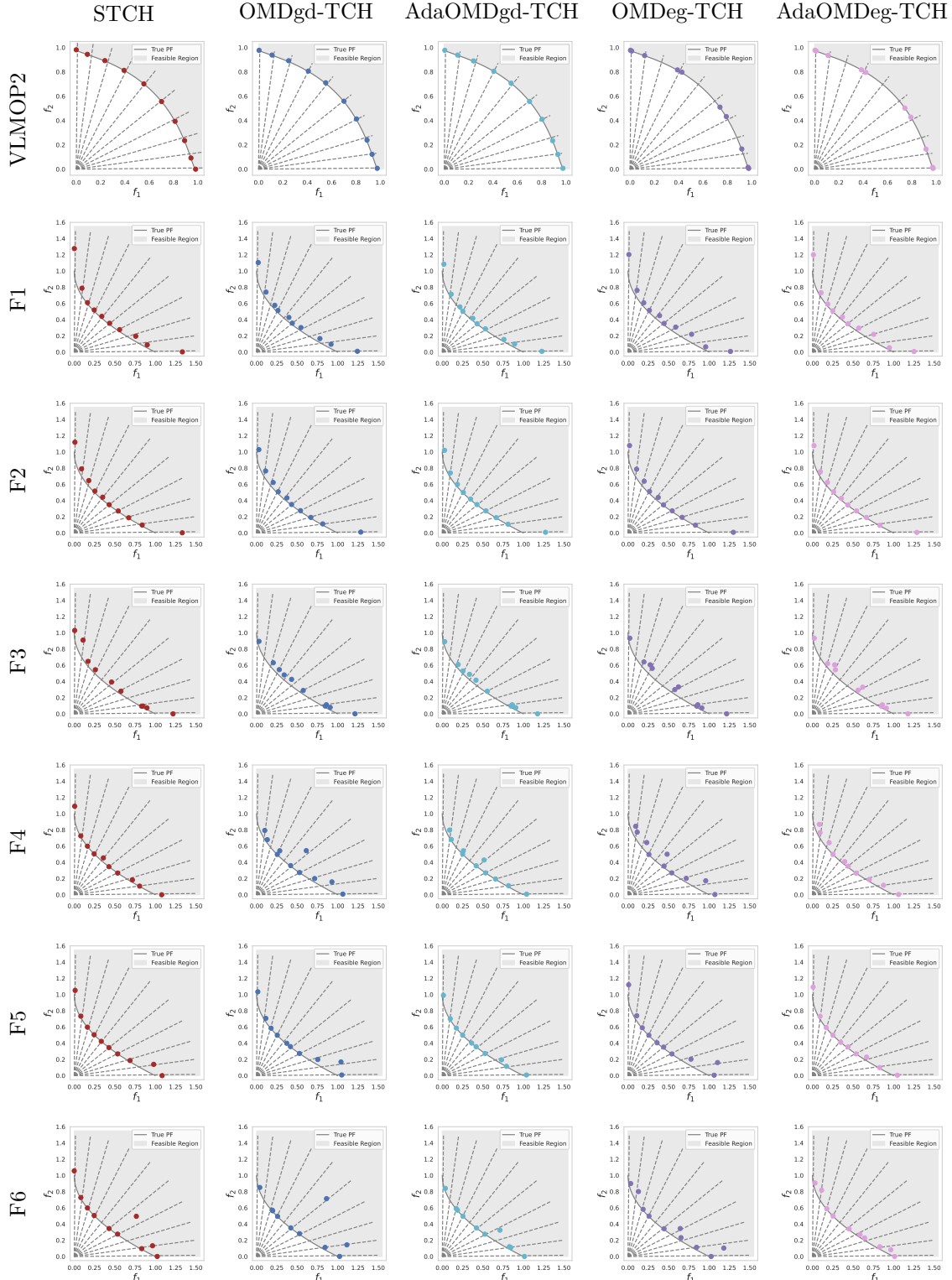

Figure 7: **Full results of STCH and (Ada)OMD-TCH.**

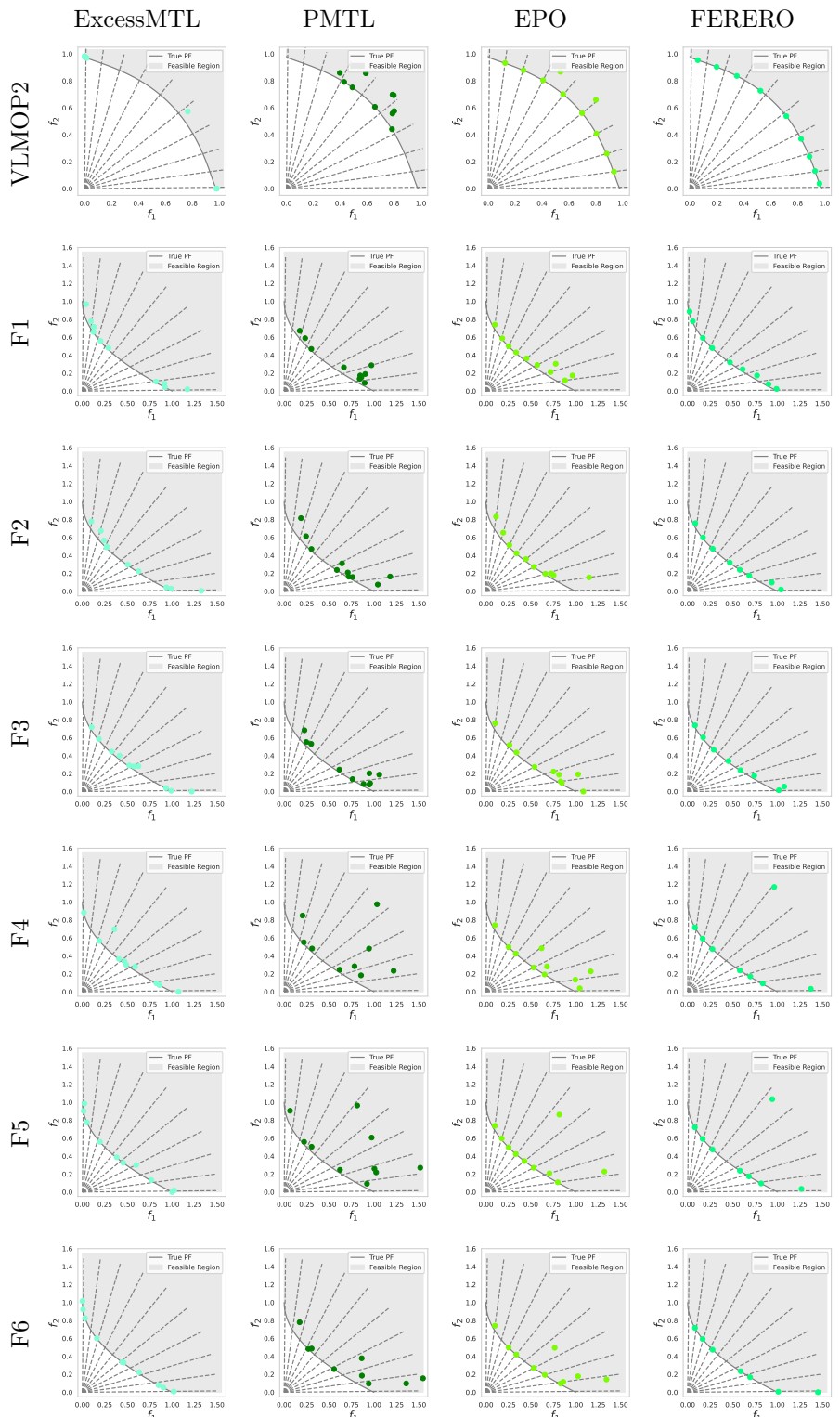

Figure 8: **Full results of ExcessMTL and preference-guided MOL methods.**

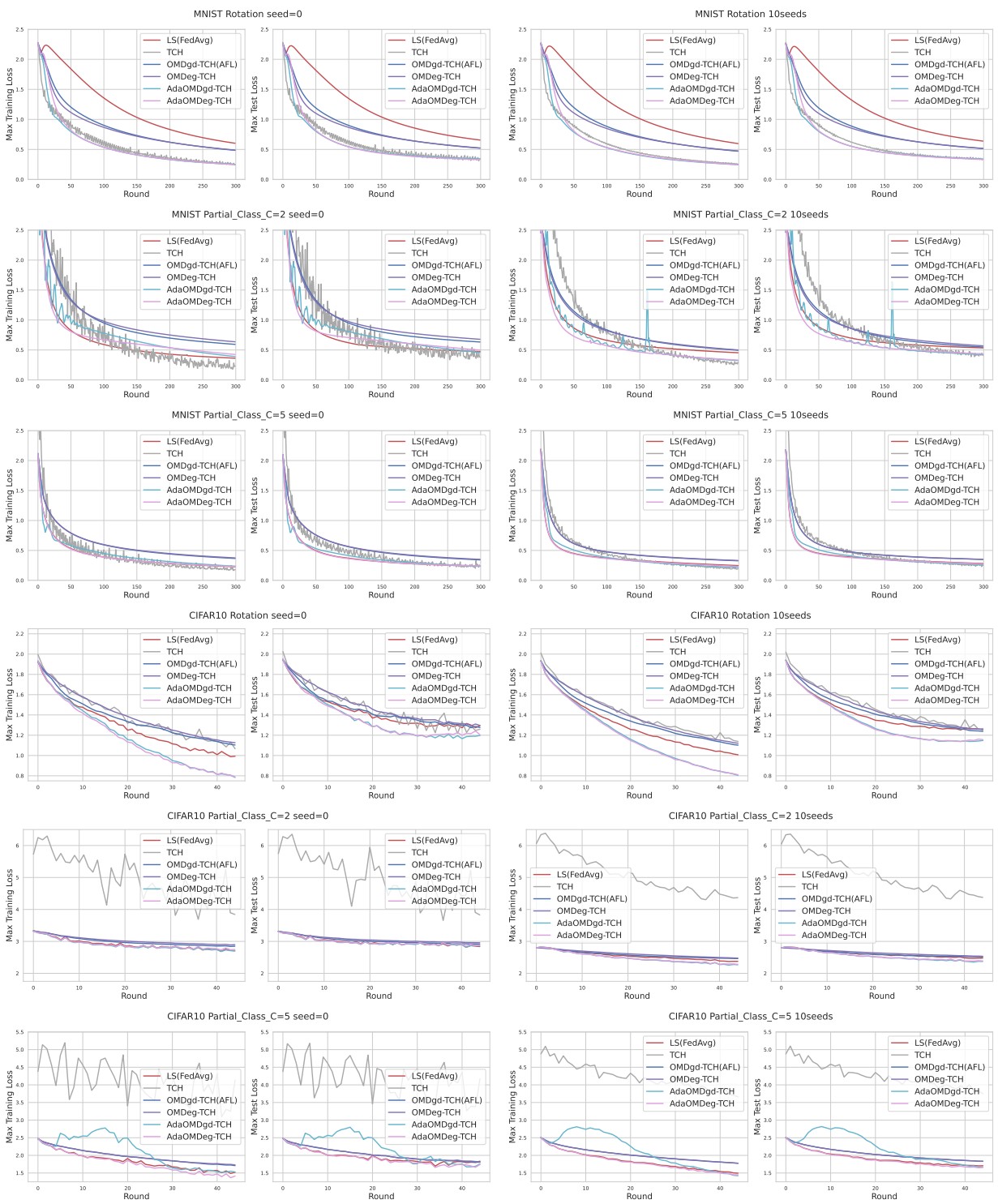

Figure 9: **Training and test curves of the worst client loss.**

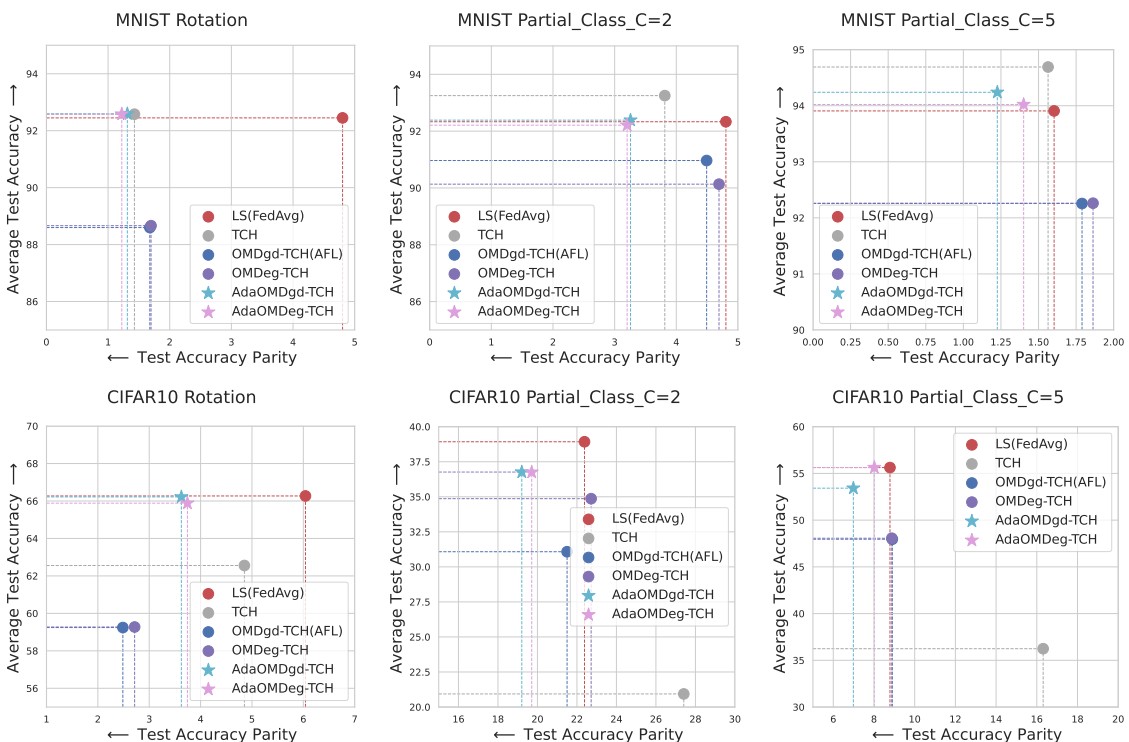

Figure 10: **Scatter plots of average accuracy and accuracy parity.**

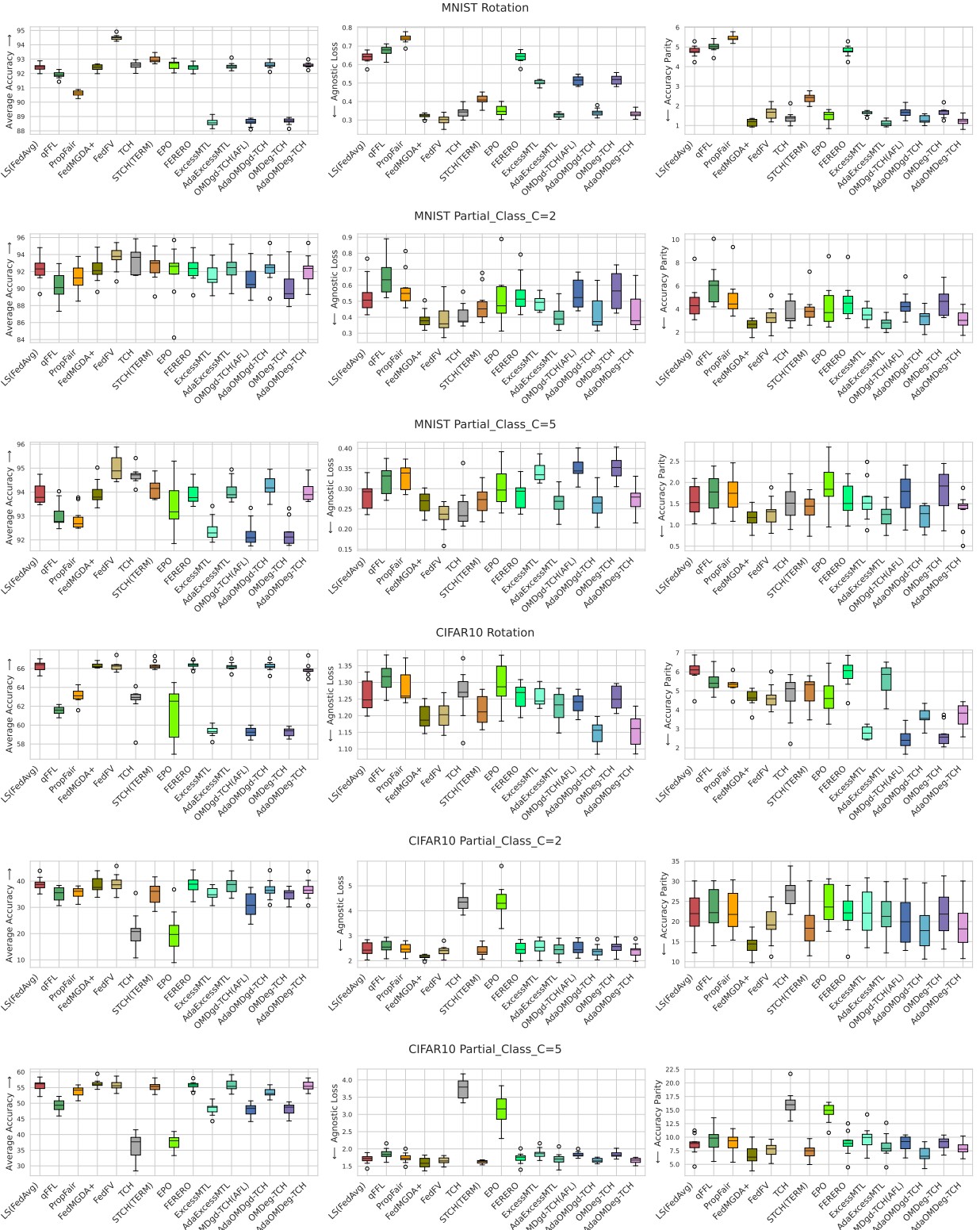

Figure 11: **Boxplots of all methods on three metrics.**

