# OpenReview forum: "Adaptive Online Mirror Descent for Tchebycheff Scalarization in Multi-Objective Learning"
_TMLR — Accepted by TMLR_

### Review · Reviewer_ZACz · 2025-12-18

**Summary Of Contributions:**

The paper develops methods for multi-objective learning, where there are multiple objective functions present. In particular, the paper focuses on solving the Tchebycheff scalarization version of the problem, which is formulated as a minmax problem over a (potentially non-uniformly) weighted average of the $m$ different objective functions. The paper modifies this minmax formulation by replacing the max over functions with a max over a simplex weighting of the functions. They then propose solving the problem with the minimax version of mirror descent. They also propose a new weighting scheme, different from releasing the average iterate, which they show has experimental benefit.

**Audience:**

No

**Audience Explanation:**

As far as the papers interest to the community, I am a bit more concerned. The proposed algorithm is simply mirror descent, and I currently don't see how this provides new insights to the community. The AdaOMD algorithm is a bit more novel, but the idea of releasing only a combination of the best iterates is also not really new. My thinking is that the most potentially interesting part for many people will be the experiments, but this is these do not seem to be the focus of the paper.

**Claims And Evidence:**

Yes

**Claims Explanation:**

I don't see any major correctness or presentational issues. A more minor comment is the following.

- The authors claim that in Remark 3.6 that the factor d in Corollary 3.5 could be lifted. Could the authors elaborate on how this is done? A direct approach using the l2 analysis of mirror descent/gradient descent would still leave a $\sqrt{d}$ factor because the width of the constraint set has l_2 radius $\sqrt{d}$.

**Requested Changes:**

It would be great if the authors could address the following concerns:

- Near the end of the introduction, the authors claim that previous works do handle non-uniform weighting, and claim they generalize these methods. Could the authors elaborate on why non-uniform weighting is difficult? It seems to be we can always push the weights into the objective function, say $f_i'(\theta) = w_i f_i(\theta)$. Now run an algorithm for uniform weighting using the $f_i'$ functions. Is there a reason this doesn't work?

- While not entirely clear, the impression the current writing gives is that converting the problem $\min \max \{w_i f_i(\theta)}$ into a smooth problem which uses a simplex weighting is a novel step. However, reformulating the non-smooth problem into a smooth minimax problem is a classic use of minmax optimization.

- The usage of ``online to batch conversion'' is a bit confusing to me. While not strictly incorrect, the algorithm the authors use would normally just be called mirror descent (not online mirror descent). Calling this an online to batch conversion feels like it adds extra steps. Could the authors clarify if this conversion is adding anything, over existing analysis? For example, see [1] Section 6.5 and 5.2.

[1]:  Convex Optimization: Algorithms and Complexity, Sebastion Bubeck

- Overall, can the authors provide more justification as to why their results would be of interest to the community? In particular, could the authors argue that their results are more than a direct application of classic techniques (as I've described in previous points) to the Tchebycheff scalarization problem?

---

> ### Author Response · Authors · 2026-02-09
>
> We sincerely thank the reviewer for their dedicated time and effort! In the following, we respond to their questions in the order we believe is most important to demonstrate the contribution of our work. We hope our answers convincingly justify that our results are of interest to the TMLR community. Please bear with us the slightly verbose explanations as we try our best to be clear.
>
> **The adaptive online-to-batch conversion**
>
> Let us first clarify the "online-to-batch conversion".
>
> > - The usage of ``online to batch conversion'' is a bit confusing to me. While not strictly incorrect, the algorithm the authors use would normally just be called mirror descent (not online mirror descent). Calling this an online to batch conversion feels like it adds extra steps. Could the authors clarify if this conversion is adding anything, over existing analysis? For example, see [1] Section 6.5 and 5.2.
> >
> > [1]: Convex Optimization: Algorithms and Complexity, Sebastion Bubeck
>
> "Online mirror descent" and "mirror descent" are essentially the same family of algorithms focusing on different learning scenarios. Fundamentally, they share the same general update rule (Eq. (2) in our paper) and derive the same set of instances, e.g., PGD, EG, p-norm. What may seem different is the form of their theoretical guarantees, the former for online problems and the latter for offline/batch problems:
>
> - online learning-sublinear regret: for any comparator $x^\*$, one has $\sum_{t=1}^T \ell(x^{(t)}) - \sum_{t=1}^T \ell(x^*) = \mathcal O(\sqrt{T})$;
> - offline/batch learning-convergence rate: let $\bar x = \frac{1}{T}\sum_{t=1}^T x^{(t)}$ and $x^\*$ be the optimal solution, one has $\ell(\bar x) - \ell(x^*) = \mathcal O(1/\sqrt{T})$, which is the one-player version of the convergence guarantees in **[1] Sections 6.5 and 5.2**.
>
> Crucially, the convergence rate is essentially derived from a sublinear regret guarantee (or an inequality of the same form, if not called regret). The key step is to take $\bar x = \frac{1}{T}\sum_{t=1}^T x^{(t)}$, converting the (online) iterates to a deterministic solution, on which convergence can be established. This process, especially the averaging step, is usually referred to as an "online-to-batch conversion". This relationship is specified in **Section 2.2**; we restate it here to show the connection to the referenced text. Thereby, we answer your questions:
>
> - Is the "conversion" adding any unnecessary steps? **No, it is not introducing any redundancy.** It is inherently required to establish convergence guarantees for (online) mirror descent on offline problems, such as those in [1] Sections 6.5 and 5.2.
> - Why do we use "online mirror descent" instead of "mirror descent"? Because we want to emphasize the "online-to-batch conversion", since we are proposing a new conversion: instead of the uniform average $\bar x = \frac{1}{T}\sum_{t=1}^T x^{(t)}$, we take an adaptive weighted average, which still leads to convergence guarantees but significantly improves solution optimality.
>
> > The AdaOMD algorithm is a bit more novel, but the idea of releasing only a combination of the best iterates is also not really new.
>
> We respectfully disagree. In all previous work that apply OMD to minimax problems, the default uniform average is adopted, as listed in Table 1. To the best of our knowledge, we are the first to propose a new online-to-batch conversion. Technically, our new conversion is not simply an arbitrary combination of best iterates; the weights are carefully allocated to ensure convergence. Moreover, we not only propose the theoretical formation of the new conversion, but also develop an algorithm that adaptively computes the weights as training proceeds, which is computationally light and thus practically useful. As shown by our experiments, our adaptive conversion significantly improves solution optimality over the traditional conversion. **Therefore, we strongly believe that our contribution of the adaptive conversion is of interest to multiple communities, such as online learning, multi-objective learning, federated learning, and algorithmic fairness.**

---

> ### Author Response · Authors · 2026-02-09
>
> **The non-uniform weighting**
>
> > - Near the end of the introduction, the authors claim that previous works do (not) handle non-uniform weighting, and claim they generalize these methods. Could the authors elaborate on why non-uniform weighting is difficult? It seems to be we can always push the weights into the objective function, say $f_i'(\theta) = w_if_i(\theta)$. Now run an algorithm for uniform weighting using the functions. Is there a reason this doesn't work?
>
> Thanks for asking for clarification! Indeed, for any multi-objective learning algorithm, we can always run it with $f_i'(\theta) = w_if_i(\theta)$ to somehow specify our preference. Yet the critical question is: can the algorithm actually find the preference-specific solution? Importantly, this cannot be answered by the algorithm's performance when using uniform weights. For example, MGDA-type MOL methods are very competitive on uniformly weighted problems, but they cannot locate the preference-specific solution even after reweighting the objectives. Please see **Figure 2(f)** for an intuitive illustration. Given such examples, when prior work using OMD for minimax problems only explored uniform weights, one cannot conclude whether OMD-TCH can be used for preference-guided MOL tasks.
>
> Essentially, as pointed out in **Section 3.4**, the reason why MGDA-type methods cannot handle non-uniform weights is that they are only guaranteed to converge to an arbitrary PO solution, so pushing the weights does not make a difference. In contrast, OMD-TCH is guaranteed to locate a specific PO solution, which can be controlled by the weights. Therefore, the success of OMD-TCH for non-uniform weights stems from its inherent properties, instead of additional designs. This is why **we did not claim the uniform-to-non-uniform weighting as a technical contribution** ("*Beyond technical contributions...*", as in the introduction).
>
> Instead, we emphasize that using OMD-TCH for non-uniform weights, and thus for general preference-guided MOL, is a "conceptually" new message. We convey this message by formalizing the theoretical convergence with embedded weights and conducting experiments on diverse preferences. This adds to the **preference-guided MOL line**, e.g., [1], where even TCH is not considered. Hence, while handling non-uniform weights is not technically challenging, we believe **our message that "(Ada)OMD-TCH can serve as a competitive preference-guided MOL method" is of important interest to the field**.
>
> [1] Chen et al. FERERO: A flexible framework for preference-guided multi-objective learning. 2024.
>
> **Other questions**
>
> > - While not entirely clear, the impression the current writing gives is that converting the problem into a smooth problem which uses a simplex weighting is a novel step. However, reformulating the non-smooth problem into a smooth minimax problem is a classic use of minmax optimization.
>
> Thank you for pointing this out. The transformation is indeed not new, as the methods we acknowledge and compare in Table 1 all implicitly perform it. Nevertheless, since it has not been applied to Tchebycheff scalarization, our writing aims to explain it as clearly as possible to the MOL audience. We agree that using a citation is better and have added the following source to our manuscript: "*..., following [1], we perform a key equivalent transformation: ...*"
>
> [1] Juditsky, Nemirovski, and Tauvel. Solving variational inequalities with Stochastic Mirror-Prox algorithm. 2011.
>
> > - The authors claim that in Remark 3.6 that the factor d in Corollary 3.5 could be lifted. Could the authors elaborate on how this is done? A direct approach using the l2 analysis of mirror descent/gradient descent would still leave a $\sqrt{d}$ factor because the width of the constraint set has l_2 radius $\sqrt{d}$.
>
> Kindly note that in our claim, by "*if we bound the $l$-2 norm of $\theta$*", we also lift the dependency on $d$ of the constraint set radius.

---

> > ### Comment · Reviewer_ZACz · 2026-02-11
> >
> > Thank you for the thoughtful rebuttal. I have read it and will take it into consideration when making my final recommendation.

---

> > > ### Author Response · Authors · 2026-02-11
> > >
> > > Thank you very much for the acknowledgment! Please let us know if you have any follow-up questions or concerns.

---

> ### Author Response · Authors · 2026-02-09
>
> **Summarization**
>
> > - Overall, can the authors provide more justification as to why their results would be of interest to the community? In particular, could the authors argue that their results are more than a direct application of classic techniques (as I've described in previous points) to the Tchebycheff scalarization problem?
>
> Combining the answers above, we summarize our findings that are of interest to the community:
>
> - We unify the framework of OMD-TCH. We convey the important conceptual message that it serves as a competitive preference-guided MOL method, supported by formalized theoretical convergence and systematic experiments.
> - We propose a technically novel adaptive online-to-batch conversion that differs significantly from prior work. We establish its theoretical convergence and demonstrate its empirical optimality.
> - Other insights:
>   - We present a theoretical iteration complexity with optimal dependence on the number of objectives $m$ for the offline setting, as in Table 1.
>   - We connect and compare our methods with other MOL methods, offering a comprehensive overview of their properties.
>
> **As such, we strongly believe our work is worth examining by the TMLR audience.**
>
> Again, thank you very much for your dedication to the reviewing process. We hope our response addresses your concerns, and please feel free to ask us if you have any questions!

---

### Review · Reviewer_ULkm · 2025-12-23

**Summary Of Contributions:**

The paper proposes AdaOMD-TCH, an algorithm for multi-objective learning via Tchebycheff scalarization (TCH). To overcome the issue of non-smoothness when optimizing TCH, AdaOMD-TCH reformulates the problem as a zero-sum game between the solution $\theta$ and the preference vector $\lambda$. Thanks to the transformation, the inner TCH maximization becomes continuous and can be solved by Online Mirror Descent (OMD), hence the name OMD-TCH. The paper further proposes an adaptive version, AdaOMD-TCH, that applies online-to-batch and reweighs the solutions.  The paper presents a theoretical convergence analysis for the non-adaptive version. Empirically, both versions are benchmarked, which shows that the adaptive version is better, and both are competitive against the baseline approaches.

**Audience:**

Yes

**Audience Explanation:**

I believe researchers with online learning interest (especially preference-guided multi-objective learning) will appreciate this work.

**Claims And Evidence:**

Yes

**Claims Explanation:**

-	The paper presents a decent overview of the settings and related literature.
-	The methodology is sufficiently clear and sound. There seems to be no major issue.
-	Although there is no convergence analysis for the adaptive version, the analysis for the standard version is sound.
-	The empirical performance seems to be promising.
-	However, there are some issues I would like the authors to address, please see the requested changes.

**Requested Changes:**

-	Add some explanation on how OMD can be used for online learning problems (e.g., what makes it suitable for online settings) and why it should be used (e.g., why not other algorithms).
-	Regarding the transformation in Section 3.1. Is this transformation a common technique in the field, or is it proposed by the authors? It will be great if there are more explanations for this key transformation.
-	Regarding the adaptive version AdaOMD-TCH, it seems high-performance solutions found in early rounds (e.g., in round 1) will have much larger weights compared to high-performance ones found later (e.g., in round 100). Will this cause bias towards the early solutions and make the algorithm stuck?
-	The experiment results are hard to follow because the authors use different formats to present the same metrics. For example, Figure 3 uses line plots, Table 2 is table format, and Figure 4 uses box plots – all share the same purpose of showing the accuracy and loss metrics. If there are no other purposes for this, I highly suggest converting to a single format, preferably Figure 3 (left), since it shows the performance across rounds – it would demonstrate clearly how the proposed method competes against the baselines across iterations.
-	Furthermore, about the experimental results, I think Figure 2 serves very little purpose (as it only shows one random run) and should be accompanied by another metric. I suggest using hypervolume (like in Table 3) to further quantify the performance. The hypervolume, as the authors also agreed, is a robust metric to quantify the performance of multi-objective problems. In fact, I think it would be best if the authors could use **line plots of hypervolume vs rounds** as the main results for all experiments, followed by tables that show the final results of each objective.

---

> ### Author Response · Authors · 2026-02-09
>
> We are sincerely grateful to the reviewer for their detailed examination of our work. We appreciate their recognition of our decent literature overview, clear and sound methodology, theoretical analysis, and promising empirical performance. We hope the following addresses their requested changes.
>
> > The paper presents a theoretical convergence analysis for the non-adaptive version.
> >
> > - Although there is no convergence analysis for the adaptive version, the analysis for the standard version is sound.
>
> We would like to first correct this misunderstanding. In fact, our theoretical analysis in Section 3.3 applies to **both the non-adaptive and adaptive** versions: kindly note the "*both Algorithms 1 and 2 converge as*" in Theorem 3.2 and the "*(with) $\hat \theta$ as $\bar \theta$ or $\tilde \theta$*" in Corollaries 3.4 and 3.5. To make it clear, we have unified all statements to "*both Algorithms 1 and 2 converge as*".
>
> > - Add some explanation on how OMD can be used for online learning problems (e.g., what makes it suitable for online settings) and why it should be used (e.g., why not other algorithms).
>
> Thank you for this suggestion! We have added the following explanation to the introduction to OMD in Section 2.2, after the overview of online learning: "*(Online mirror descent is a fundamental online learning algorithm,) commonly favored for its generalizable regret guarantees on decision sets with constrained geometries, as well as practically stable update trajectories. (Its general update rule is...)*"
>
> > - Regarding the transformation in Section 3.1. Is this transformation a common technique in the field, or is it proposed by the authors? It will be great if there are more explanations for this key transformation.
>
> Thank you for raising this question, and let us clarify the picture. The transformation of $\min_{\theta} \max_{i \in [m]} f_i(\theta)$ to $\min_{\theta} \max_{\lambda \in \Delta_m} \sum_{i=1}^m \lambda_i f_i(\theta)$ is indeed not new - its first elaboration dates back to Juditsky, Nemirovski, and Tauvel, 2011 [1]. It has then been applied to many minimax problems with one-hot selections like $i \in [m]$ so that they can be solved with OMD - precisely those listed in Table 1 and discussed at the end of Section 3.1 ("*In various scenarios requiring fairness or robustness, many methods adopt a minimax objective and apply different instances of OMD solvers...*"). Nevertheless, this technique has not been applied to Tchebycheff scalarization, and our work differs from previous applications by exploring diverse preferences, presenting a better convergence rate, and proposing the novel adaptive conversion, as discussed at the end of introduction and Section 3.1. Finally, regarding the transformation, we agree that a clarifying citation is necessary and have added the said source to our manuscript.
>
> [1] Juditsky, Nemirovski, and Tauvel. Solving variational inequalities with Stochastic Mirror-Prox algorithm. 2011.
>
> > - Regarding the adaptive version AdaOMD-TCH, it seems high-performance solutions found in early rounds (e.g., in round 1) will have much larger weights compared to high-performance ones found later (e.g., in round 100). Will this cause bias towards the early solutions and make the algorithm stuck?
>
> Thanks for asking! We guess you are considering such a special case: (1) a trajectory PO solution is found in round 1, denoted as $\theta^{(1)}$; (2) many non-PO solutions are encountered in the following rounds, and their weights are re-assigned to $\theta^{(1)}$; (3) another trajectory PO solution is found in round 100, denoted as $\theta^{(100)}$; (4) it seems that $\theta^{(1)}$ gets to absorb more weights from $\theta^{(2) \sim (99)}$ than $\theta^{(100)}$. This impression is partly correct, but we emphasize that it actually has little impact on the algorithm's performance. The key principle is that $\theta^{(1)}$ only inherits weights from the iterates it dominates, but not from those only dominated by $\theta^{(100)}$. Therefore, if $\theta^{(100)}$ is far from $\theta^{(1)}$ and the iterates dominated by each of them are nearly disjoint, $\theta^{(1)}$ can't "steal" too much weight from $\theta^{(100)}$; conversely, if the iterates dominated by $\theta^{(1)}$ and $\theta^{(100)}$ overlap a lot, it indicates that they are close, and slightly biasing weights between them won't make too much difference. Moreover, since OMD updates inherently control the distance between adjacent iterates, for two close high-performance solutions, instead of $\theta^{(1)}$ and $\theta^{(100)}$, they are more likely to be $\theta^{(95)}$ and $\theta^{(100)}$, where the number of intermediate iterates is very small, making the bias negligible. To conclude, the raised concern is **handled by the design of the adaptive conversion and the OMD updates**. Our experiments also demonstrate non-affected performance in practice.

---

> ### Author Response · Authors · 2026-02-09
>
> > - The experiment results are hard to follow because the authors use different formats to present the same metrics. For example, Figure 3 uses line plots, Table 2 is table format, and Figure 4 uses box plots – all share the same purpose of showing the accuracy and loss metrics. If there are no other purposes for this, I highly suggest converting to a single format, preferably Figure 3 (left), since it shows the performance across rounds, ....
> > - Furthermore, about the experimental results, I think Figure 2 serves very little purpose (as it only shows one random run) and should be accompanied by another metric. I suggest using hypervolume (like in Table 3) to further quantify the performance. ... In fact, I think it would be best if the authors could use **line plots of hypervolume vs rounds** as the main results for all experiments, followed by tables that show the final results of each objective.
>
> Thank you for the suggestions. We kindly argue that the formats are indeed chosen to **serve different purposes**. Below, we explain why each figure/table is presented that way and respond to the suggested changes.
>
> - Figure 3, Table 2, and Figure 4: Yes, they all report results for the 10-client fair federated learning experiment, but are designed to **support different claims**. To show the unstable training of TCH and the smoothing effects of OMD updates, Figure 3 plots the training curves. To quantitatively demonstrate the optimality of AdaOMD-TCH over OMD-TCH, Table 2 presents final numerical metrics. To clearly reflect relative performance across 15 methods, Figure 4 visualizes box plots instead of using raw numbers.
>
>   Regarding the use of line plots of hypervolume vs. rounds, our response is two-fold. First, hypervolume does not apply to the fair FL experiments, as in this setting we aim for **one** fair solution, instead of multiple diverse solutions. While hypervolume also measures optimality, it does not reflect fairness, e.g., a loss of (0.4, 0.6) may have the same hypervolume as (0.2, 0.7) using (1.0, 1.0) as the reference point, but (0.4, 0.6) is clearly more fair. In contrast, our current metrics (avg. accuracy, agnostic loss, accuracy parity) account for both optimality and fairness. Second, plotting training curves for Figure 3 is possible because only 6 methods are considered; however, the figure will be very messy if all 15 methods are included, with too many overlapping and crossing curves. This is why we use box plots and tables to report the results.
>
> - Figure 2: We first clarify that the results in Figure 2 are **averaged over 3 seeds**, and we have updated its caption to note this. Besides examining solution diversity, another purpose here is to see how well each method finds the **exact** PO solutions, i.e., the intersections of the PF and the preference rays, which is an important property of our methods. Since synthetic problems have known Pareto Fronts, we believe it is best to visualize the solutions and intersections as in Figure 2 to reflect the exactness. As a complement, we follow the suggestion and **add hypervolume results**. Please check the end of Appendix B.1 and Table x, where TCH achieves the best hypervolume, and AdaOMDgd-TCH ranks second on five of the seven problems, which are **consistent with our current findings**. Similarly, we skip plotting curves due to the large number of methods (14) considered.
>
> Again, thank you very much for your time and effort. We hope our responses address the issues, and please feel free to ask us if you have any questions!

---

### Review · Reviewer_QyVB · 2026-01-27

**Summary Of Contributions:**

This paper studies preference-guided multi-objective optimization via Tchebycheff scalarization (TCH). It reformulates TCH as a min–max game over a simplex variable and solves it with the online mirror descent (OMD) method. The primary contribution is an adaptive online-to-batch conversion that outputs a weighted average of only the trajectory Pareto-optima, with weight redistribution from dominated iterates. The paper provides convex stochastic convergence bounds and conducts experiments on synthetic MOL and federated settings. The results demonstrate the effectiveness of the proposed method.

Overall, the presentation is clean, and the idea of adaptive conversion is straightforward yet highly effective. But I have some concerns.

1. Alg.2 requires the comparison among different $\theta$ to determine the Pareto dominance. Ideally, the objective values should be evaluated with full data. But in practice, these values are estimated from minibatches. I would like some clarification on the implementation details.
2. The main convergence theorem assumes convexity, but in practice the experiment settings are nonconvex. I would like to see further discussion on this gap, or some convergence analysis in a nonconvex setting.

**Audience:**

Yes

**Audience Explanation:**

The convergence analysis here assumes convexity, and it is consistent with the compared baselines in Table 1. The proposed methods in this paper yield better convergence rates.

**Broader Impact Concerns:**

This paper tackles multi-objective learning, which is a common problem in the real world. I have no additional concerns.

**Claims And Evidence:**

Yes

**Claims Explanation:**

The presentation of this paper is clean, and the theoretical and experimental results demonstrate the effectiveness of the proposed method.

**Requested Changes:**

I would like to see the following adjustments:

1. Ablation where dominance comparisons are conducted using (i) full training loss, (ii) a held-out validation estimate, and (iii) minibatch estimates. As mentioned before, there are some inconsistencies between the method and implementation.
2. Does this method apply to the varying scales of losses? I mean, from the experiments, the loss scales are similar and comparable. But in practice, loss scales can differ across tasks or clients. What is the empirical performance then?

---

> ### Author Response · Authors · 2026-02-09
>
> We extend sincere gratitude to the reviewer for their time and effort. We are glad they appreciate our work on the "straightforward yet highly effective" adaptive conversion, clean presentation, and better convergence rates compared to baselines in Table 1. We hope to address their remaining concerns as follows.
>
> > 1. Alg.2 requires the comparison among different $\theta$ to determine the Pareto dominance. Ideally, the objective values should be evaluated with full data. But in practice, these values are estimated from minibatches. I would like some clarification on the implementation details.
> >
> > 1. Ablation where dominance comparisons are conducted using (i) full training loss, (ii) a held-out validation estimate, and (iii) minibatch estimates. As mentioned before, there are some inconsistencies between the method and implementation.
>
> Thank you for this insightful question! Indeed, when comparing iterates for the adaptive conversion, the most accurate way is to evaluate each objective on its full training data. Yet in practice, when the data is large and minibatch updates are adopted, such full evaluations would lead to extra training time. Therefore, in our implementation, we use batch losses as estimates whenever using minibatch updates, which applies to the federated learning experiments on CIFAR. Still, this yields the current results that support our findings and claims.
>
> Following your suggestion, we conduct an ablation on the fair federated learning CIFAR rotation setting, where the objective values for dominance comparison use
>
> - minibatch estimates: This corresponds to the current results in Table 8(d).
> - full training loss: Each objective/client is evaluated on its full local training data.
> - held-out validation estimates: Since CIFAR10 does not feature a validation set, we split the test set in half, using one for validation and the other for testing. We did not split the training set as altering training data would make the current hyperparameters, e.g., number of epochs, unfit.
>
> We directly use the current hyperparameters. The results are averaged over 10 seeds (gd for AdaOMDgd-TCH, eg for AdaOMDeg-TCH):
>
> |                              | gd-mini (current) | gd-full | gd-val | eg-mini (current) | eg-full | eg-val |
> | ---------------------------- | ----------------- | ------- | ------ | ----------------- | ------- | ------ |
> | Avg. Accuracy $\uparrow$     | 66.218            | 66.061  | 66.386 | 65.885            | 65.913  | 66.260 |
> | Agnostic Loss $\downarrow$   | 1.148             | 1.128   | 1.185  | 1.156             | 1.130   | 1.171  |
> | Accuracy Parity $\downarrow$ | 3.592             | 3.503   | 4.022  | 3.688             | 3.372   | 3.819  |
>
> Comparing the full and mini columns, one sees that using full training loss indeed improves agnostic loss, which is the goal of Tchebycheff scalarization, and accuracy parity. Using the validation set slightly worsens the two but increases the average accuracy. Most importantly, **switching to either evaluation barely changes the relative performance of AdaOMD-TCH compared to other methods in Table 8(d)**, e.g., they still achieve the best agnostic loss. This validates that our methods are **robust to different dominance comparison strategies**.
>
> The above ablation study has been added to our manuscript. Please see the end of Appendix B.2.
>
> > 2. The main convergence theorem assumes convexity, but in practice the experiment settings are nonconvex. I would like to see further discussion on this gap, or some convergence analysis in a nonconvex setting.
>
> Let us clarify the gap. Overall, we aim for our convex theoretical analysis and non-convex experiments to complement each other. In the convex setting, we use rigorous proofs to ensure convergence to a specific Pareto optimal solution, making convex experiments less urgent. In the non-convex setting, proving exact PO solution convergence is very challenging, yet we still hope to verify such properties, so we resort to practical experiments. Indeed, these experiments validate that our methods converge to specific PO solutions, with applications in finding diverse/fair solutions, even in the non-convex case.
>
> To make this clear, we have added the following sentence at the beginning of Section 4: "*Given the theoretical guarantees for convex objectives in Section 3.3, in this section, we study the performance of (Ada)OMD-TCH in non-convex experiments.*"

---

> ### Author Response · Authors · 2026-02-09
>
> > 2. Does this method apply to the varying scales of losses? I mean, from the experiments, the loss scales are similar and comparable. But in practice, loss scales can differ across tasks or clients. What is the empirical performance then?
>
> Thank you for raising this interesting question. When the objectives are of different scales, directly using them for our methods, or any method that involves dynamic weight allocation based on objective values (e.g., TCH), can cause bias towards objectives of a larger scale, interfering with the original preference. This is actually one of the reasons we chose the current setup, where the similar loss scales enable us to best investigate fairness/trade-off performance without other artifacts.
>
> One practical trick to resolve the scaling problem is to, when updating the dynamic weights, replace $f_i(\theta^{(t)})$ with the ratio $f_i(\theta^{(t)}) / f_i(\theta^{(0)}) \in (0,1]$, where $f_i(\theta^{(0)})$ is the $i$-th loss in the first training round. Thereby, the weight allocation process examines not the raw objectives, but how much each has improved, aiming for a balanced/preference-specified *degree of training* across objectives. This trick has been adopted by previous works to normalize excess risks [1,2].
>
> For the empirical performance, due to the extensive work required to adapt to other datasets with objectives of different scales, we would like to leave it for future exploration. Nevertheless, we apply the ratio trick to TCH, STCH, and (Ada)OMD-TCH on our fair federated learning CIFAR rotation setting to obtain a rough impression. By using the ratios, we evaluate the agnostic ratio, i.e., the worst test loss ratio across objectives, and the ratio parity, i.e., the standard deviation across test loss ratios. We use the current hyperparameters and average results over 10 seeds:
>
> |                             | LS    | TCH   | STCH  | OMDgd-TCH | AdaOMDgd-TCH | OMDeg-TCH | AdaOMDeg-TCH |
> | --------------------------- | ----- | ----- | ----- | --------- | ------------ | --------- | ------------ |
> | Agnostic ratio $\downarrow$ | 0.656 | 0.648 | 0.627 | 0.647     | **0.465**    | 0.651     | 0.465        |
> | Ratio parity $\downarrow$   | 0.087 | 0.066 | 0.068 | **0.024** | 0.043        | 0.029     | 0.042        |
>
> These results are **consistent with our current findings**. Specifically, we still have (1) TCH is better than LS, (2) OMD-TCH is often better than TCH, (3) AdaOMD-TCH greatly improves optimality over OMD-TCH as reflected by the smaller agnostic ratio, i.e., the performance of the least-trained objective, with a slight increase in ratio parity, which is still better than other methods. This demonstrates **the compatibility of our methods with the ratio trick and, thus, potentially with different scales of objectives**.
>
> The above ablation study has been added to our manuscript. Please see the end of Appendix B.2.
>
> [1] He et al. Robust multi-task learning with excess risks. 2024.
>
> [2] Chen et al. Gradnorm: Gradient normalization for adaptive loss balancing in deep multitask networks. 2018.
>
> Again, thank you very much for your devotion to reviewing our work. We hope our responses address your concerns, and please feel free to ask us if you have any questions!

---

### Author Response · Authors · 2026-02-09
**Updated manuscript**

We sincerely thank all reviewers for their dedicated time and detailed examination. We have carefully addressed their concerns in the individual responses. Please note that we have updated the manuscript to incorporate several changes:

- an ablation study on different evaluation data for iterate comparison in Appendix B.2, as requested by Reviewer QyVB
- a demonstration of the compatibility of our methods with the ratio trick for different scales of objectives in Appendix B.2, as suggested by Reviewer QyVB
- hypervolume metrics for synthetic experiments in Appendix B.1, as requested by Reviewer ULkm
- several changes in the main body for clarifications

The changes are highlighted in red. Please let us know if there are any additional questions!

---

### Decision · Action_Editor_QhXS · 2026-03-08

**Recommendation:** Accept as is

**Audience:**

Yes

**Audience Explanation:**

This paper should be of interest to audience in multi-objective optimization, learning from multiple distributions in general, optimization, and beyond.

**Claims And Evidence:**

Yes

**Claims Explanation:**

The claims made in the submission are supported by enough evidence. During the rebuttal stage, the reviewers mostly addressed reviewers' concerns around novelty, the gap between theoretical analysis and practical implementation, and presentation of the work. In particular, they performed additional evaluation to compare iterates for the adaptive conversion and show results on objectives with different scales, and clarified implications of theoretical results.

---

> ### Author Response · Authors · 2026-03-19
>
> We highly appreciate all reviewers and the AE for their dedication to evaluating our work! We have uploaded the camera-ready version with all previously discussed changes.